



# A framework to regionalize conceptual model parameters for global hydrological modeling

Wenyan Qi[1], Jie Chen[1, 2], Lu Li[3], Chong-yu Xu[4], Jingjing Li[1], Yiheng Xiang[1], Shaobo Zhang[1]

[1]State Key Laboratory of Water Resources & Hydropower Engineering Science, Wuhan University, 299 Bayi Road, Wuchang Distinct, Wuhan, Hubei, 430072, China

[2]Hubei Provincial Key Lab of Water System Science for Sponge City Construction, Wuhan University, Wuhan, China

[3]NORCE Norwegian Research Centre, Bjerknes Centre for Climate Research, Bergen, Norway

[4]Department of Geosciences, University of Oslo, P.O. Box 1047, Blindern, 0316 Oslo, Norway

*Correspondence to*: jiechen (jiechen@whu.edu.cn)

**Abstract.** To provide an accurate estimate of global water resources and help to formulate water allocation policies, global hydrological models (GHMs) have been developed. However, it is difficult to obtain parameter values for GHMs, which results in large uncertainty in estimation of the global water balance components. In this study, a framework is developed for building GHMs based on parameter regionalization of catchment scale conceptual hydrological models. That is, using appropriate global scale regionalization scheme (GSRS) and conceptual hydrological models to simulate runoff at the grid scale globally and the Network Response Routing (NRF) method to converge the grid runoff to catchment streamflow. To achieve this, five regionalization methods (i.e. the global mean method, the spatial proximity method, the physical similarity method, the physical similarity method considering distance, and the regression method) are first tested for four conceptual hydrological models over thousands medium-sized catchments (2500-50000 km$^2$) around the world to find the appropriate global scale regionalization scheme. The selected GSRS is then used to regionalize conceptual model parameters for global land grids with 0.5$^o$×0.5$^o$ resolution on latitude and longitude. The results show that: (1) Spatial proximity method with the Inverse Distance Weighting (IDW) method and the output average option (SPI-OUT) offers the best regionalization solution, and the greatest gains of the SPI-OUT method were achieved with mean distance between the donor catchments and the target catchment is no more than 1500 km. (2) It was found the Kling-Gupta efficiency (KGE) value of 0.5 is a good threshold value to select donor catchments. And (3) Four different GHMs established based on framework were able to produce reliable streamflow simulations. Overall, the proposal framework can be used with any conceptual hydrological model for estimating global water resources, even though uncertainty exists in terms of using difference conceptual models.

## 1 Introduction

Water resource is one of the most important natural resources that can significantly influence the social and economic development for a region and a country (Parajka et al., 2007). The management of water resources should be based on a fully understanding of the spatial and temporal variation of regional water resources. In particular, problems caused by climate



change, increasing water demand due to growing world population, water conflicts in multinational river basins and virtual water trade all reflect the requirements of continental and global-scale hydrological simulations (Döll et al., 2003; GRDC, 2005; Oki and Kanae, 2006; Widén-Nilsson et al., 2007). Hydrological models are main tools for simulating runoff at multiple spatial and temporal scales (Arnell, 2003; Xia et al., 2003; Oki and Kanae, 2006; Elin et al., 2007) and have achieved a great

progress during the past few decades (Korzoun et al., 1974; Vörösmarty et al. 1989; Widén-Nilsson et al., 2007; Oleson et al., 2010; Beck et al., 2016; Sood and Smakhtin, 2015). The most commonly used continental or global scale hydrological models are the land surface scheme (LSS) which is a component of climate models that simulate the energy balance at soil, atmosphere and vegetation interfaces (Haddeland et al., 2011; Bierkens, 2015). However, global climate models have large biases in global runoff simulations (Yang and Dickinson, 1996; Sellers et al. 1986; Sood and Smakhtin, 2015). Hence, global hydrological

models (GHMs) are developed to simulate (sub-) surface water fluxes and storages. Some of the widely used GHMs include Variable Infiltration Capacity model (VIC, Liang et al. 1994), Water Balance Model–Water Transport Model (WBM-WTM, Vörösmarty et al. 1989) and PCRaster GLOBal Water Balance model (PCRGLOBWB, van Beek and Bierkens 2008; van Beek et al., 2012; http://www.globalhydrology.nl/models/pcr-globwb-1-0/).

The majority of GHMs applied at the continental to global scale tends to rely on a priori parameterizations based on expert

opinion, hydrologic theory, field data, case studies, or data sets of questionable quality (Beck et al., 2016). For example, the parameter values of the WBM-WTM are tuned by an adjustment factor, rather than calibration (Vörösmarty et al. 1989). The parameter values of the Macro Probability Distribution Model (Macro-PDM) are set based on literature review or previous model applications and 6 out of 13 parameters are globally uniform (Arnell et al., 1999, 2003). The Community Land Model (CLM) (Oleson et al., 2010) consider the base flow recession constant (k) as a fixed value, even though it has been recognized

that k varies spatially (Hall, 1968; Beck et al., 2013b). The k value of the PCRGLOBWB was determined based on the drainage theory and hydrogeologic data, however, many studies have found that there is a weak link between k and current hydrogeologic data sets (van Beek and Bierkens 2008; Peña-Arancibia et al., 2010; Beck et al., 2013b). Therefore, it is unlikely that the current global-scale hydrological models have reached their potential in streamflow simulations. Considering the restriction of the priori parameterizations, some GHMs have been developed on the basis of model parameter regionalization.

For example, the Water and Snow Balance Modeling System Macroscale (WASMOD-M) transferred the calibrated parameter sets to grid cells by searching for the most commonly occurring parameter set within a rectangular window and found that regionalized parameters produced better streamflow estimates than spatially uniform parameters (Widén-Nilsson et al., 2007). Beck et al. (2016) transferred the calibrated parameter sets of Hydrologiska Byråns Vattenbalansavdelning (HBV) model from the selected (674 out of 1787) donor catchments to 0.5° grid cells with the most similar climatic and physiographic characteristics and found that HBV with regionalized parameters outperformed nine state-of-the-art macroscale models.

Additionally, some studies have focused on regionalization of macro-scale hydrologic models and illustrated the effectiveness of regionalization method in macro-scale runoff simulation. For example, Troy et al. (2006) interpolated the model parameters





of calibrated grid cells to the uncalibrated grid cells across the continental United States and found that this approach was efficient for large-scale applications. Livneh and Lettenmaier (2013) tested regression model which linking ''zonally

representative'' parameters to catchment descriptors across the continental United States and found that this approach resulted in skillful model performance. Various regionalization methods were proposed in the past few decades(Abdulla et al., 1997; Hundecha et al., 2004; Pokhrel et al., 2008; Jin et al., 2009; Luis et al., 2010; He et al., 2011; Razavi and Coulibaly, 2013). The widely used regionalization methods in literatures include regression-based approaches (RE), distance/attributes-based approaches (spatial proximity and physical similarity) and global mean method (GM) (Jin et al., 2009; He et al., 2011; Razavi

and Coulibaly, 2013). Other techniques for regionalization include clustering methods and hydrologic classification (Merz and Bloschl, 2004; Luis et al., 2010; Livneh and Lettenmaier, 2013). Numerous studies have been made to compare regionalization approaches in different regions (e.g., Oudin et al., 2008; Li et al., 2010; Yang et al., 2018, 2019, 2020). However, there is still no clear conclusion on the best-performed regionalization method. In addition, none of these GHMs developed on model parameter regionalization has compared the widely used regionalization approach and investigated the

optimal approaches at the global scale, instead, they all used one specific scheme to calculate model parameters for GHMs. More over, most of these studies only established one global hydrological model and the uncertainty of GHMs is not taken into account by using various hydrological models with different structures and concepts.

Therefore, in order to complement existing global water-balance models to reduce runoff estimation uncertainty and provide valuable spatial and temporal estimates of global water resources, a framework for building GHMs is proposed by combining

four conceptual hydrological models and five regionalization methods over 2277 medium-sized catchments with drainage area between 2500 and 50000 km$^2$ around the world. Specifically, the objectives of this study are to:

(1) identify an appropriate parameter regionalization method and the optimal global scale regionalization scheme (GSRS) for building GHMs;

(2) build four GHMs by regionalizing model parameters from watershed scale to grid scale using the framework;

(3) validate the performance of GHMs over 2277 catchments around the world widely;

(4) simulate global water resources.

**2. Material and methods**

**2.1 Meteorological data**

The meteorological data used in this study are daily precipitation, air temperature and potential evaporation. The potential

evaporation data at the global scale were obtained from the Global Land Evaporation Amsterdam Model (GLEAM v3) potential evaporation data set (1980-2015) at the 0.5 °resolution, which is calculated by using the Priestley and Taylor equation based on observations of surface net radiation and near-surface air temperature (Martens et al., 2017; Miralles et al., 2011). The





daily temperature data were obtained from the European Centre for Medium range Weather Forecasts (ECMWF)–Interim

Reanalysis (ERA-Interim) at the 0.5° resolution (1979–2019, Dee et al., 2011;

https://apps.ecmwf.int/datasets/data/interim-full-daily/levtype=sfc/). The precipitation data were obtained from

the Global Precipitation Climatology Centre (GPCC) V.2018 precipitation data set (1982-2016) (Fuchs et al.,2009), which is

based on gauged precipitation data provided by national meteorological and hydrological services, regional and global data

collections as well as the World Meteorological Organization (WMO) GTS-data (GPCC, http://gpcc.dwd.de).

In addition, thirteen catchment descriptors were selected for hydrological model parameter regionalization, which have been

commonly used in some other studies (e.g. Koren et al., 2010; Yang et al., 2018; Oudin et al., 2008; Merz and Blöschl, 2004)

(Table 1).We assumes that these catchment descriptors independent from each other and a well-behaved relationship exists in

the catchment descriptors and model parameters.

**Table 1 The statistical information of catchment descriptors used in regionalization methods**

|  | Mean | Median | Minimum | Maximum |
|---|---|---|---|---|
| Climate index |  |  |  |  |
| Aridity index | 0.85 | 0.80 | 0.04 | 3.33 |
| Mean annual potential evaporation | 1169 | 1089 | 301 | 3060 |
| Terrain characteristics |  |  |  |  |
| Mean slope (°) | 2.49 | 1.78 | 0.01 | 20.95 |
| Mean elevation (m) | 645 | 545 | 1.94 | 4719 |
| Area (km$^2$) | 12016 | 7486 | 2500 | 50000 |
| Land use |  |  |  |  |
| Forest (%) | 41.67 | 41.01 | 0.00 | 98.79 |
| Water body (%) | 2.60 | 0.07 | 0.00 | 69.73 |
| Built-up land (%) | 1.49 | 0.31 | 0.00 | 62.35 |
| Total cultivated land (%) | 16.98 | 6.41 | 0.00 | 96.20 |
| Soil index |  |  |  |  |
| Topsoil Clay Fraction (% wt.) | 16.83 | 15.97 | 0.00 | 73.26 |
| Subsoil Clay Fraction (% wt.) | 18.38 | 17.35 | 0.00 | 78.22 |
| Water holding capacity | 11.82 | 3.74 | 0.00 | 50.00 |
| Soil thickness(mm) | 42.36 | 43.83 | 0.00 | 100.00 |

Data Citation: Harmonized World Soil Database (version 1.1); GlobCover Land Cover Maps,

http://due.esrin.esa.int/page_globcover.php; Global Aridity and PET Database, http://www.cgiar-csi.org.



### 2.2 Observed streamflow data

The observed daily streamflow data were obtained from the Global Runoff Data Centre (GRDC; http://www.bafg.de/GRDC/). The GRDC dataset aims at helping earth scientists to analyze global climate trends and to assess the environmental impacts and risks. It comprises river discharge data for more than 9500 stations from 161 countries. Basin boundaries and flow path were

taken from HYDRO1K (USGS 1996a). Continent boundaries were taken from STN-30p (Vörösmarty et al., 2000).

There are some uncertainties which will impact the results. For example, small catchments might result in less reliable results since the model resolution is 0.5 degree. Besides, almost all large rivers are regulated (Vörösmarty et al., 2004; Nilsson et al., 2005) while it is difficult to obtain reliable time-series data on regulation. The following three criteria were used to choose catchments from our analysis:

(1) The streamflow record length was required to be at least 5 years (not necessarily consecutive) during the 1982–2015 period.

(2) The catchment size is over 2,500 $km^2$.

(3) The upper limit of catchment size was set as 50,000 $km^2$ in order to minimize the effects of regulation.

In total, 2277 catchments were selected, and Fig. 1 shows the spatial distribution of these catchments. The majority of the

catchments are located in North America, Europe, Southeast Asia and central South America. Only a few catchments are located in the Middle East, North Africa, the central Australian and the Russian Far East and Siberia regions.

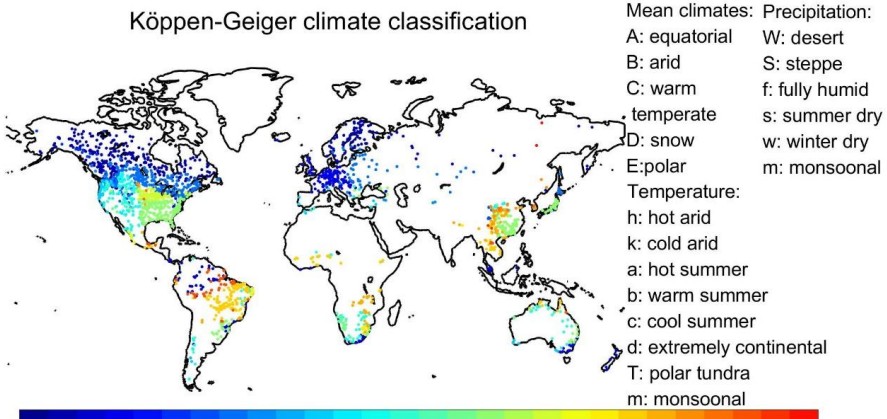

**Figure 1: Location of the catchments used in this study**

### 2.3 Hydrological models

Four conceptual hydrological models were used to simulate runoff at the daily time step. These hydrological models were chosen because of the proven effectiveness around the world and the successful application in regionalization studies. The model structures of these hydrological models differ from each other and the number of the model parameters varies from six to twenty-one.

### 2.3.1 The Génie Rural à 4 paramètres Journalier model (GR4J)

GR4J has four free parameters and is constructed based on the unit hydrograph principles (Perrin et al., 2003). GR4J is one of the most commonly used hydrological models in the world, because of its simpleness and good performance (Li et al., 2014; Oudin et al., 2008; Zhang et al., 2014; 2016). Since there is no snow module in the original GR4J model, the snow module-CemaNeige (Valéry et al., 2010) was incorporated into GR4J in this study. CemaNeige allows to estimate the snowmelt and simulate the snowpack evolution by using 2 parameters, and the coupling of GR4J and CemaNeige has been

tested in some other studies (e.g. Valéry et al., 2010; Coron et al., 2014; Hublart et al., 2015). Therefore, there are six model parameters in total for GR4J in this study.

### 2.3.2 The simple lumped conceptual daily rainfall-runoff model (SIMHYD)

SIMHYD is a simplified version of the HYDROLOG model (Chiew et al., 1980), which contains three storages for interception loss, soil moisture and groundwater and routing process (Chiew et al., 2002). The SIMHYD model has been

applied in ungauged catchments over different climate regions (e.g. Zhang and Chiew, 2009). The same as GR4J, the snow module (CemaNeige, Valéry et al., 2010; Coron et al., 2014; Hublart et al., 2015) was incorporated into the original version for snowmelt simulations, so there are eleven parameters in total for SIMHYD in this application.

### 2.3.3 The Xinanjiang model (XAJ)

XAJ was developed by Zhao et al. (1980; 1992) for the prediction inflow of Xinanjiang Reservoir. It consists of

evapotranspiration, runoff production, runoff separation, and flow routing. It is the most commonly used hydrological model in China, and has been tested in various climate regions and many regionalization studies (e.g. Zhang and Chiew, 2009; Li and Zhang, 2017; Yang et al., 2020). The same snow module (CemaNeige, Valéry et al., 2010; Coron et al., 2014; Hublart et al., 2015) was coupled with the original XAJ model for snowmelt simulation. Therefore, the total number of parameters becomes seventeen.

### 2.3.4 The Hydrological Model of École de technologie supérieure model (HMETS)

HMETS was developed by École de technologie supérieure and contains two reservoirs for the saturated and vadose zones (Chen et al., 2011; Martel et al., 2017). HMETS has been used in diverse climate regions in Canada and China (e.g. Chen et al., 2018; Shen et al., 2018). It has twenty-one parameters and can simulate the main hydrological processes (i.e. snow accumulation, snowmelt and refreezing, infiltration and flow routing and evapotranspiration).

### 2.4 Model calibration and evaluation criteria

For each catchment, the record of observed streamflow data was split into a calibration period (consisting of the first 70% of the record) and a validation period (consisting of the remaining 30% of the record). In addition, the shuffled complex evolution



method optimization algorithm (SCE-UA, Duan et al., 1992; 1993) was used to optimize the hydrological model parameters, in which, the objective function was chosen to be the KGE (Gupta et al., 2009). KGE has been introduced as an improvement of

the widely used Nash-Sutcliffe efficiency, which considers different types of  model errors, namely the error in the mean, the variability, and the dynamics. The KGE was calculated by using Eq. (1):

$$\mathrm{KGE} = 1 - \sqrt{(R-1)^2 + \left(\frac{\overline{Q}_{\mathrm{sim}}}{\overline{Q}_{\mathrm{obs}}} - 1\right)^2 + \left(\frac{\mathrm{CV}_{\mathrm{sim}}}{\mathrm{CV}_{\mathrm{obs}}} - 1\right)^2}$$

(1)

where $\overline{Q}_{\mathrm{obs}}$ is the mean of observed runoff and $\overline{Q}_{\mathrm{sim}}$ is the mean of simulated runoff. R is the Pearson correlation coefficient between observed and simulated runoffs. $\mathrm{CV}_{\mathrm{sim}}$ and $\mathrm{CV}_{\mathrm{obs}}$ represent the coefficient of variations of observed and

simulated streamflows. KGE value ranges from  $-\infty$ to 1 and the larger the KGE value, the better the simulation.

**2.5 Regionalization methods**

The regionalization methods used in this study are the most commonly used regression-based approaches, distance/attributes-based approaches and global mean method (Jin et al., 2009; He et al., 2011; Razavi and Coulibaly, 2013). The regression-based method assumes that a well-behaved relationship exists in the observable catchment characteristics and

model parameters (Burn and Boorman, 1993). Parameters for an ungauged catchment are derived by using the relationship between catchment descriptors and model parameters for the donor catchments. In this study, all catchment descriptors were assumed to be related to model parameters (Arsenault and Brissette, 2014; Yang et al., 2018, 2019).

The distance/attributes-based methods assume that the parameter sets of hydrological models on gauged catchments can be transferred to nearby or physically similar ungauged catchments following different procedures. The key of these methods is to

find the closest or the most similar donor (gauged) catchments to the ungauged catchments. The distance/attributes-based methods usually include: spatial proximity (SP) method, physical similarity (PS) method and physical similarity method considering distance (PSD) (Yang et al., 2018, 2020).

The SP method assumes that nearby catchments should have similar behavior for climate and catchment conditions (features) varying uniformly in space. The Euclidean distance was used to calculate the distance $D_{ud}$ between the donor and ungauged

catchments in this study. The $D_{ud}$ was calculated by using Eq. (2):

$$D_{ud} = \sqrt{(x_u - x_d)^2 + (y_u - y_d)^2}$$

(2)

where, u and d represent the ungauged and donor catchments, respectively; $x_u, x_d$ and $y_u, y_d$ are catchment positions of the ungauged and donor catchments under the Lambert Azimuthal Equal Area projection system, respectively.

The PS method is based on the assumption that catchments with similar attributes show similar hydrological behaviors. The

core of PS method is the selection of the physical similarity metric (Samaniego et al., 2010). Many studies have focused on the selection of proper similarity index between donor and ungauged catchments and the proper catchments attributes for





similarity index calculation (Burn and Boorman 1993; Luis et al., 2010; Samaniego et al., 2010).The similarity index in this study was calculated following Eq. (3) (Burn and Boorman 1993):

$$SI_{ud} = \sum_{i=1}^{k} \frac{|CD_{d,i} - CD_{u,i}|}{\Delta CD_i}$$ (3)

where, $CD$ is the catchment descriptor; u and d represent the ungauged and donor catchments, respectively; $k$ is the total number of catchment descriptors and $\Delta CD_i$ represents the range of $i^{th}$ catchment descriptor.

Considering the limitation of the two regionalization methods mentioned above, some studies (e.g. Samuel et al., 2011; Viviroli and Seibert, 2015; Yang et al., 2018) integrated SP with PS to improve the regionalization ability. In present study, the PSD method in which the distance was considered as one of the catchment descriptors was used and then the similarity index

was calculated.

For distance/attributes-based methods, there are two different averaging options to transfer the model parameter sets from donor catchments: (i) parameter average option, which transfers the averaged model parameters from donor catchments to ungauged catchments; and (ii) output average option, which averages runoff simulations calculated by using individual parameter sets from donor catchments to ungauged catchments (Oudin et al., 2008; Yang et al., 2018). In addition, there are

two different weighting approaches used to combine the model parameters or model outputs: (i) Arithmetic Mean (AM) method; and (ii) Inverse Distance Weighted (IDW) method (Parajka et al. 2007; Yang et al. 2018).

The GM method is a relatively simple regionalization method. Generally, the arithmetic mean of the parameters of all gauged catchments is directly applied to the ungauged catchments. All regionalization methods used in this study are summarized in Table 2.

**2.6 The performance of regionalization methods under different efficiency thresholds**

In order to find the suitable donor catchments and the optimal regionalization scheme for GHMs, the performance of regionalization methods under different thresholds of model efficiency was tested. A threshold of model efficiency (all, >0, >0.5, >0.6, >0.7, >0.8, >0.9) was determined for the calibration period. In other words, when the efficiencies of the catchments were below a threshold, these catchments were not used as donor catchments to predict runoffs for ungauged

catchments. The threshold named 'all' means that no catchment was excluded from the donor catchments. However, all catchments, whether poorly or well modeled, were all considered as pseudo-ungauged.

**2.7 Framework establishment**

Firstly, all five parameter regionalization methods were applied at the catchment scale over 2277 catchments by using four conceptual hydrological models. Secondly, the best performed regionalization method was selected as global scale

regionalization scheme (GSRS) for regionalization of global hydrological models at the spatial resolution of $0.5°×0.5°$ grid cell all over the world except for Antarctica and Arctic region. This procedure is based on an assumption that the parameters at



the catchment scale can also be used at the grid cell scale. In other words, the $0.5^{\circ} \times 0.5^{\circ}$ grid cell was treated as a catchment (Beck et al., 2016).

The GSRS was only used to simulate runoff at the grid scale. For calculating watershed streamflow, runoff routing algorithms

are required (Vörösmarty et al. 1989; Döll et al., 2003). There are several large-scale runoff routing algorithms have been developed (Graham et al., 1999; Vörösmarty et al., 2000; Döll and Lehner, 2002). The Network Response Routing (NRF) method was selected in this study to converge the grid runoff to catchment streamflow, since this method transfers high-resolution delay dynamics, instead of networks, to any lower spatial resolution where runoff is generated (Gong et al., 2009; Li et al., 2019). There are 2 parameters for the NRF runoff routing approach and these parameters were calibrated using

catchment observations but not regionalized. Thus the framework was established, that is, using GSRS and conceptual hydrological models to simulate runoff at the grid scale globally and then using NRF method to converge the grid runoff to catchment streamflow. According to the framework, four GHMs were built.

**Table 2 Summary of regionalization methods used in this study**

| Regionalization methods | Averaging options | Weighting approaches | Abbreviation |
|---|---|---|---|
| Global mean (GM) | | | GM |
| Spatial Proximity (SP) | Parameter Averaging | Arithmetic Mean | SPA |
| | | Inverse Distance Weighted | SPI |
| | Output Averaging | Arithmetic Mean | SPA-OUT |
| | | Inverse Distance Weighted | SPI-OUT |
| Physical Similarity (PS) | Parameter Averaging | Arithmetic Mean | PSA |
| | | Inverse Distance Weighted | PSI |
| | Output Averaging | Arithmetic Mean | PSA-OUT |
| | | Inverse Distance Weighted | PSI-OUT |
| Physical Similarity considering distance (PSD) | Parameter Averaging | Arithmetic Mean | PSDA |
| | | Inverse Distance Weighted | PSDI |
| | Output Averaging | Arithmetic Mean | PSDA-OUT |
| | | Inverse Distance Weighted | PSDI-OUT |
| Regression (RE) | Multiple        linear | | RE |

SPA means spatial proximity method with parameter averaging option and arithmetic mean approach; PSI-OUT means

physical similarity method with output averaging option and Inverse Distance Weighted approach.

### 3. Results and discussion

### 3.1 Hydrological model calibration and validation at catchment scale

Figure 2 shows the cumulative density function (CDF) curves of the percentage of catchments with KGE values exceeding the given value (Mittelhammer et al., 2013) for all hydrological models over the calibration and validation periods. The calibration



results of different hydrological models are close to each other. For all hydrological models, the KGE value of more than 60%

catchments are higher than 0.7 for the calibration period and 0.5 for the validation period, respectively. In Table 3, we can see

the number of catchments and its percentage in the total catchments under different thresholds of model efficiency

(>0, >0.5, >0.6, >0.7, >0.8, >0.9) in the calibration period. Generally, the number of catchments under different thresholds are

similar among the models. More detailed spatial distribution of the model efficiency at the calibration period is shown in Fig. 3.

We can see that the KGE values are above 0.8 for most catchments in eastern Canada, the USA, the southern China and along

the Atlantic Coast of Europe. For those regions, the good model performances are probably attributed to the dense precipitation

gauges. Other catchments in the American tropics, the Andes (South America) and the northwest China show low KGE values,

because of the complex topography and the lack of precipitation gauges in these areas. The results show that the distributions

of model efficiency of four hydrological models are similar to each other and indicate that the difference between hydrological

models was negligible in the model calibration and validation, which is in line with previous studies (Beck et al., 2016; Vetter

et al., 2015; Demirel et al., 2015).

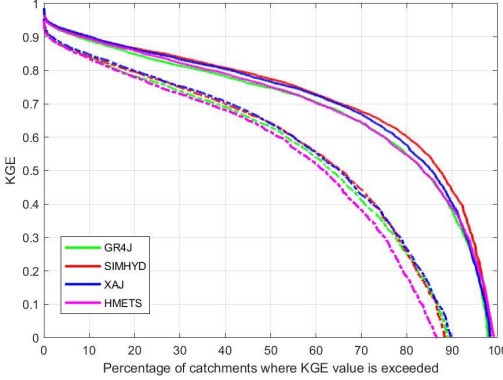

**Figure 2: The performance of hydrological models by split-sample test evaluated by the KGE value. The solid and dash lines show
the performance for the calibration and validation periods respectively.**

**3.2 Parameter regionalization at the catchment scale**


Figure 4 shows the model performance of distance/attributes-based methods using the different number of donor catchments

with different weighting and averaging options under each KGE threshold. It shows that the IDW approach performs better

than the AM approach for all distance/attributes-based regionalization methods and all hydrological models. This is consistent

with the results from other studies (e.g. Arsenault and Brissette, 2014; Samuel et al., 2011; Li et al., 2014). The worse

performance of AM approach probably caused by the large difference of distance or similarity among our studied catchments.

However, the weighting scheme of IDW minimizes the negative impact caused by the farthest distance or the least similar

donors.





Further, the output averaging option outperforms the parameter averaging option globally in the study, which is also consistent

with previous studies (e.g. Arsenault and Brissette, 2014; Samuel et al., 2011; Li et al., 2014; Yang et al., 2018, 2020) at the

regional scale.

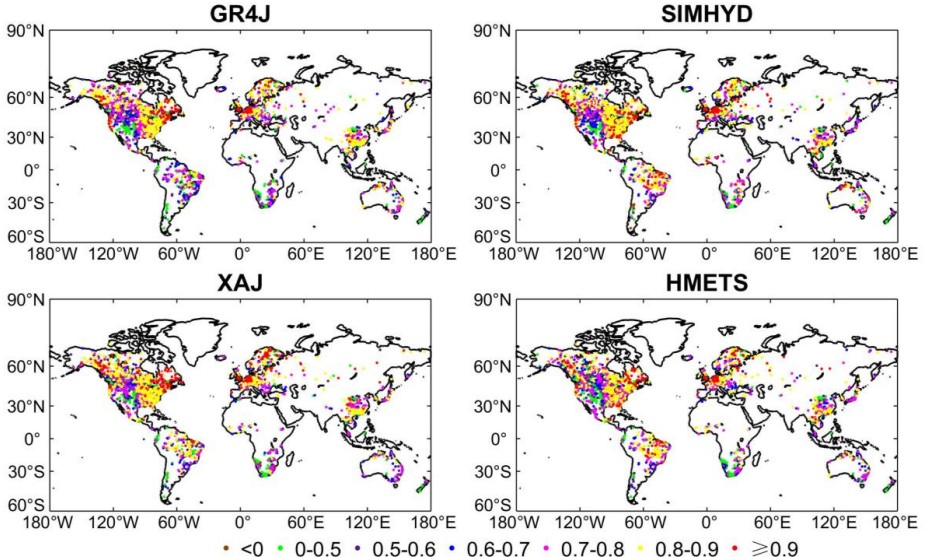

**Figure 3: Spatial distribution of model efficiency (i.e. KGE).**

**Table 3. The numbers and the percentage of donor catchments under different efficiency thresholds for four hydrological models**

|  | all | >0 | >0.5 | >0.6 | >0.7 | >0.8 | >0.9 |
|---|---|---|---|---|---|---|---|
| **GR4J** | 2277 | 2229 | 1902 | 1707 | 1386 | 791 | 170 |
|  | 100 | 98 | 84 | 75 | 61 | 35 | 7 |
| **SIMHYD** | 2277 | 2243 | 1985 | 1826 | 1491 | 965 | 211 |
|  | 100 | 99 | 87 | 80 | 65 | 42 | 9 |
| **XAJ** | 2277 | 2240 | 1947 | 1770 | 1477 | 939 | 236 |
|  | 100 | 98 | 86 | 78 | 65 | 41 | 10 |
| **HMETS** | 2277 | 2258 | 1903 | 1711 | 1375 | 840 | 193 |
|  | 100 | 99 | 84 | 75 | 60 | 37 | 8 |

The hydrological model performance is improved with the increase in the number of donor catchments for all hydrological

models when using the output averaging option. However, for the parameter averaging option, with the increase number of

donor catchments the KGE value becomes smaller for most of the regionalization methods. In addition, the optimal number of

donor catchments for output averaging option always lies between 4 and 6. To balance the effect and the amount of

computation, 5 donor catchments are suggested to use for output averaging method.





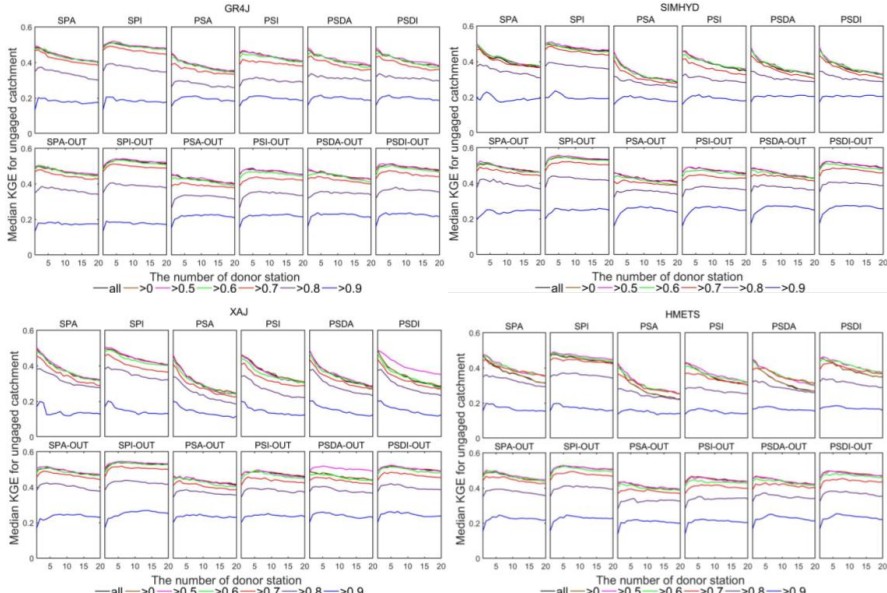

**Figure 4: The median KGE value for distance/attributes-based regionalization methods with an increasing number of donor catchments.**

How will using the poorly calibrated catchments as donor catchments impact the performance in the regionalization method and hydrological modeling, has not been much discussed in the literature. In this study, we compared the performance of regionalization methods under different KGE thresholds for all regionalization methods (including regression method, not shown) and found that the KGE threshold value of 0.5 is the best. Using catchments under the KGE threshold of 0.5 as donor catchments has little effect on the performance of hydrological models for ungauged catchments. However, when using the efficiency threshold up to 0.9, the hydrological model performance is remarkably dropped for ungauged catchments. This is attributed to the limited number and the particular location of the gauged catchments considered (Table 5). When using the threshold of 0.90, only 170 (for GR4J), 211 (for SIMHYD), 236 (for XAJ) and 193 (for HMETS) donor catchments out of the 2277 catchments were available for use. Therefore, the threshold of 0.5 was taken as the best performance threshold and the total number of donor catchments for global regionalization scheme ranges from 1902 (for GR4J) to 1985 (for SIMHYD) for different hydrological models. Thus, when there are limited gauged (donor) catchments around the ungauged catchment, it may be preferable to keep poorly calibrated gauged catchments in regionalization.

The performance of all regionalization approaches under threshold 0.5 is shown in Fig.5. The differences were observed among regionalization methods. For example, the median KGE value of the SPI-OUT was the highest and the value of the GM method was the lowest, which indicated that the SPI-OUT outperforms all other regionalization methods for all hydrological models and the GM method produces the worst results.




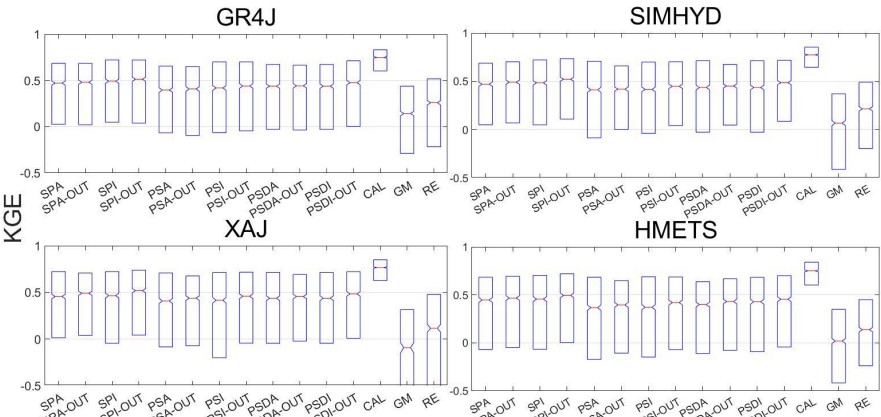

**Figure 5: Comparison of model efficiencies on ungauged catchments using several regionalization schemes (see the detailed information of regionalization methods in Table 3).**

Figure 6 shows the percentage of the best performed regionalization method under different mean distance (between the donor and ungauged catchments). For all hydrological models, when the mean distance between donor and ungauged catchments is smaller than 1500 km, the average proportion of catchments that the SPI-OUT method outperforming others is the largest. This indicates that the greatest gains of the SPI-OUT method in performance were achieved for ungauged catchments with mean distance no more than 1500 km from the donors.

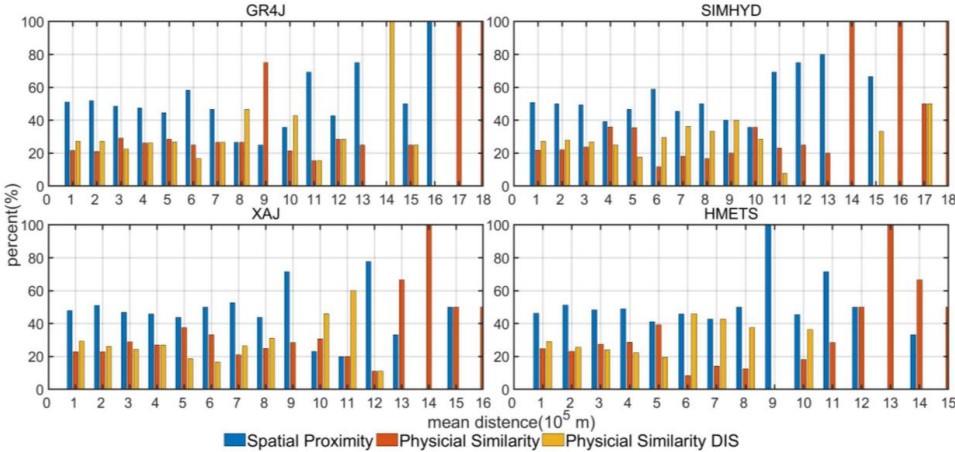

**Figure 6: The proportion of outperformed regionalization method over 2277 catchments with increasing mean distance between the donor and ungauged catchments.**

Overall, the performance of different regionalization schemes were consistent in each hydrological models. That is the SPI-OUT method outperformd the others and the GM method performed the worst. In addition, the differences of regionalization performance among hydrological models were small, which indicated that the difference of regionalization performance caused by model structure is not significant. However, in this study, it is the two parsimonious hydrological models (i.e., GR4J and SIMHYD) that slightly outperforms the other two more complex models (i.e., XAJ and HMETS) in





most of the situations. This phenomenon may be caused by equifinality. Equifinality is defined as having multiple sets of parameters that lead to equally acceptable model performance during the model calibration and validation (Beven and Freer, 2001; Luis et al., 2010). Some studies indicated that hydrological models with large parameter spaces and high parameter interdependence may have more acceptable parameter sets during calibration than the parsimonious hydrological models and consequently have a less chance to be successful for hydrological regionalization (Arsenault and Brissette, 2014; Yang et al.,

2018). In order to illustrate the equifinality of different hydrological models exists in this study, 10 calibration sets were generated instead of a single one during model calibration, considering that 10 sets should be different enough from one another to adequately sample the parameter set uncertainty under equifinality constraints (Arsenault and Brissette, 2014). The 10 calibrated parameter sets for each catchment were only accepted if the KGE value was within 0.01 of the best KGE value for that basin to ensure equifinality was present. The normalization arithmetic was used to calculate the distribution of

model parameters during the 10 calibration. A normalization factor (NF) was calculated by using Eq. (4):

$$NF = \frac{cal_{\max,i} - cal_{\min,i}}{bou_{top,i} - bou_{bottom,i}} \tag{4}$$

where, i represents the ith model parameter; $cal_{\max,i}$ and $cal_{\min,i}$ are the maximum and minimum parameter values among the

10 calibration, respectively; $bou_{top,i}$ and $bou_{bottom,i}$ are the top and bottom limitation of the ith parameter during model

calibration. The closer that NF is to 1, the larger the parameter spaces exist in the 10 calibration.

The NF values of each parameters of four hydrological models were calculated for 2277 catchments. Figure 7 shows the percentage of catchments with NF values exceeding the given value (Mittelhammer et al., 2013) for each parameters of four hydrological models. For GR4J, the parameter equifinality is not as important as for other hydrological model, as indicated that the NF values are relative smaller. For SIMHYD, there are 5 parameters (out of 11) with high NF values. However, for XAJ and HMETS, most parameters have high NF values. This illustrate that four models are all subject to the effects of parameter

equifinality. In particular, the equifinality is more serious for models with more parameters (i.e. XAJ and HMETS) than those with few parameters (i.e. GR4J and SIMHYD).

The parameter equifinality would affect the regionalization results of this study, which needs to be further investigated in the future. In fact, due to the high relationship between regionalization methods and calibrated model parameter sets, equifinality has been recognized as one of the major source of uncertainty exist in regionalization. However, it is hard to quantify its impact

on regionalization performance, since the high uncertainties exist in regionalization. In the past few decades, there are some studies have focused on evaluating the uncertainty caused by equifinality for runoff simulations in ungauged basins (Arsenault and Brissette, 2014). Some have also proposed new regionalization methods to reduce the influence of uncertainty ( Luis et al., 2010). However, further research is needed on issues of uncertainties exist in regionalization, since we are still far from understanding uncertainty lies in every aspect of prediction in ungauged basins (PUB) studies.





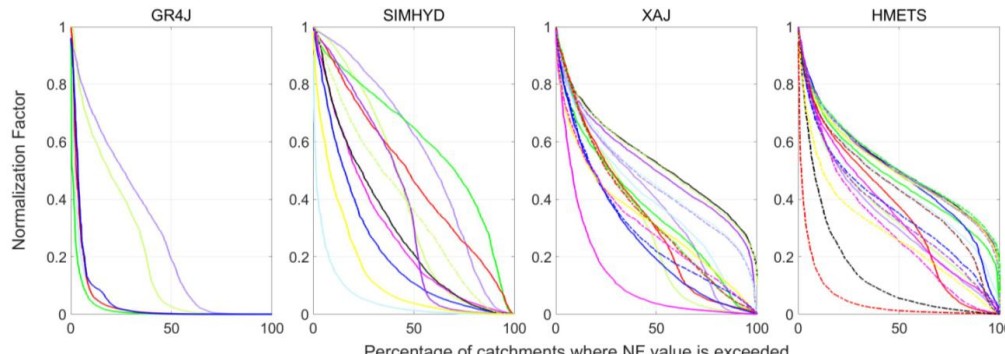

**Figure 7: The CDF curves of different parameters. The different lines represent the different parameters of each hydrological model (no need to specify in this study).**

### 3.3 GHMs regionalization at the grid scale

The GSRS was selected based on the best performed catchment scale regionalization methods (section 3.2). That is, only catchments with KGE value being greater than 0.5 at the calibration period were used as donor catchments. For grid cells with a mean distance less than 1500 km to donors, the calibrated parameter sets of the 5 nearest donor catchments were transferred by using SPI-OUT method. For grid cells with a mean distance larger than 1500 km, the parameters were extracted from PSDI-OUT method. Figure 8 shows the mean distance to the 5 nearest donor catchments. The mean distances were generally <1500 km for most part of the world for all GHMs. While there were still some regions where the mean distance was >1500 km, like southern South America, Southeast Asia, Northeast Africa and Northeast North America.

By using the above parameter regionalization at the grid scale, the runoff can be calculated for all global grid cells. To further quantify the performance of the global regionalization scheme, the NRF runoff routing approach was used to converge runoff of GHMs to streamflow time series for all 2277 catchments. The KGE values are presented in Fig. 9 and Table 4. Even though the KGE values from GHMs were smaller than those obtained by using the best performed catchment scale regionalization methods (i.e. SPI-OUT), they were close to those obtained by using the other catchment scale distance/attributes-based regionalization results. In addition, the performance of the GHMs are better than that of the catchment scale regression and global mean methods. The median KGE value of SIMHYD-G was the largest (0.384) and that of HMETS-G was the least (0.374). The difference between the median KGE value of GHMs and that using the best performed catchment scale regionalization method (i.e., SPI-OUT) were 0.134 (for GR4J), 0.137 (for SIMHYD), 0.137 (for XAJ) and 0.121 (for HMETS), respectively. The difference between the median KGE value of GHMs and that using the calibrated parameters were 0.370 (for GR4J), 0.389 (for SIMHYD), 0.384 (for XAJ) and 0.346 (for HMETS), respectively. The performance of GHMs built using framework was about half of that obtained using calibrated parameters in terms of the median KGE value. This result is consistent with previous study obtained by Beck at al. (2016) using an aggregate objective function (AOF) score as model calibration criterion in 1113 small-to-medium sized catchments at global scale. In fact, the median daily KGE values of four





GHMs are larger than that found from Beck at al. (2016) (0.19 for HBV with regionalization parameters and 0.04 for the ensemble mean of nine state-of-the-art models). However, the performance of GHMs was not as good as that of the regional hydrological model calibration (Widén-Nilsson et al., 2007; Beck et al. 2016).

In fact, GHMs were built for macroscale water resource management and they would not be the first choice for basin scale applications because of its coarse resolution (Sood and Smakhtin, 2015). Whereas, four different GHMs were built effectively

for providing valuable estimates of global water resources and helping to formulate water allocation policies under climate change.

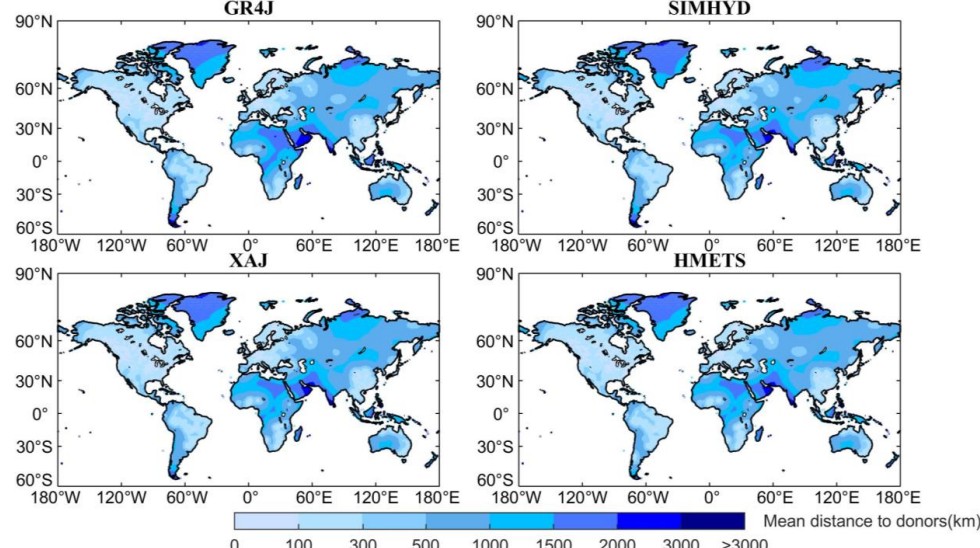

Figure 8: Mean distance to the 5 nearest donor catchments.

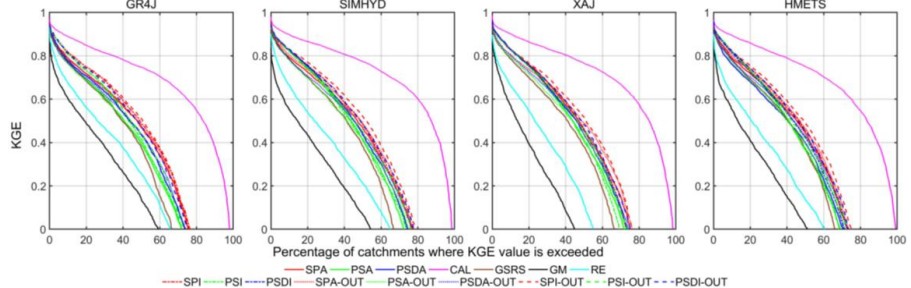

Figure 9: The cumulative density function (CDF) curves of different regionalization methods.

Table 4. The median of KGE values for the tested regionalization methods and hydrological models over all catchments

|        | CAL   | GM     | SPI-OUT | framework |
|--------|-------|--------|---------|-----------|
| GR4J   | 0.748 | 0.141  | 0.512   | 0.378     |
| SIMHYD | 0.774 | 0.068  | 0.522   | 0.384     |
| XAJ    | 0.766 | -0.093 | 0.519   | 0.382     |
| HMETS  | 0.750 | 0.019  | 0.495   | 0.374     |





To further evaluate the effectiveness of the framework in terms of climate regimes, Table 5 summarizes the median KGE value over 2277 catchments for five Köppen-Geiger climate types. Generally, all GHMs perform much worse for arid climate regions than for the other four climate types. Figure 10 showed the spatial distribution of KGE value of GHMs. For most of

Europe and the east coast of North America, KGE is generally above 0.5, and even above 0.7 over many catchments. In contrast, most catchments in southwest Africa and northwest Australian perform much worse. In fact, previous studies indicated that most of the existing GHMs overestimate runoff in arid basins (e.g. TRIP with different land-surface models, Oki et al., 2001; WGHM, Döll et al., 2003; HBV, Beck et al., 2016). Because for arid regions, the high evaporative losses, the highly nonlinear response behavior, and the flashy nature of the streamflow time series make it difficult to simulate streamflow

time series (Pilgrim et al., 1988; Widén-Nilsson et al., 2007; Beck et al., 2016).

The differences among four GHMs are small, indicating that even though uncertainty exists in terms of using difference conceptual models, the proposal framework can be used with any conceptual hydrological model. Overall, the XAJ-G performs the best and the HMET-G performs the worst. Parajka et al. (2013) compared previous studies and found that poorer regionalization performance with more model parameters. However, this is not shown in our study. The results show that

neither the GR4J-G model (who has the least model parameters), nor the HMETS-G model (who has the largest number of model parameters) shows the best performance in global regionalization scheme, which indicates that the performance of regionalization method does not reduce with the increase of number of the model parameters. Similar conclusion has been drawn in Yang et al. (2020) where they studied dependence of regionalization methods on the complexity of hydrological models on 86 independent catchments in Norway. This might because both model structure and adequate complexity are

important (Gan et al., 1997; Orth et al., 2015; Reynolds et al., 2018; Yang et al., 2020). It suggests that hydrologists must strike the right balance between model flexibility and the number of parameters for optimal results.

**Table 5.The median KGE values obtained by GHMs for different climate types**

| Climate Type | GR4J-G | SIMHYD-G | XAJ-G | HMETS-G |
|---|---|---|---|---|
| All (n=2277) | 0.378 | 0.384 | 0.382 | 0.374 |
| A: equatorial (n=293) | 0.192 | 0.283 | 0.273 | 0.269 |
| B: arid (n=247) | -0.310 | -0.340 | -0.316 | -0.332 |
| C: warm temperate (n=717) | 0.449 | 0.450 | 0.454 | 0.452 |
| D: snow (n=970) | 0.467 | 0.465 | 0.464 | 0.467 |
| E: polar (n=50) | 0.483 | 0.465 | 0.517 | 0.440 |




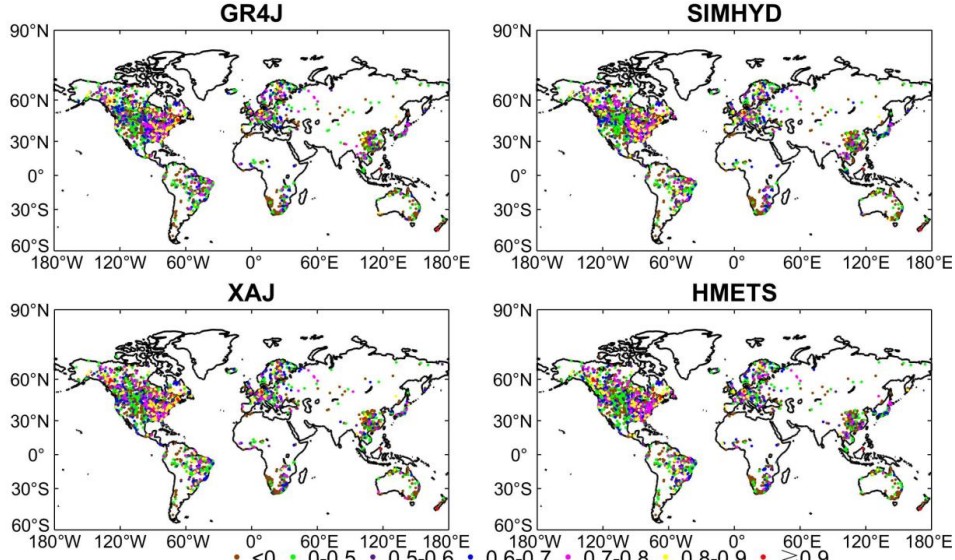

**Figure 10: The distribution of KGE value obtained by GHMs.**

### 3.4 Runoff variation obtained by GHMs

In order to evaluate global water budget of four GHMs, the performance of the GHMs is further evaluated over six large basins, whose drainage area is larger than 40,000 km$^2$ from six continents. Figure 11 shows mean monthly hydrograph of observed and simulated streamflows. Generally, the seasonality of streamflow was well captured by the GHMs. For three out of six catchments (i.e. Beaver river in North America; Passo mariano pinto in South America and Hofkirchen in Europe), streamflows of January and February are underestimated, while streamflows of April and May are overestimated. In addition, a slight overestimation is found in Oceania and African catchments (i.e. Red rock and Hol pads leegte), which may be caused by the bias from GPCC precipitation (Muller Schmied et al., 2016).

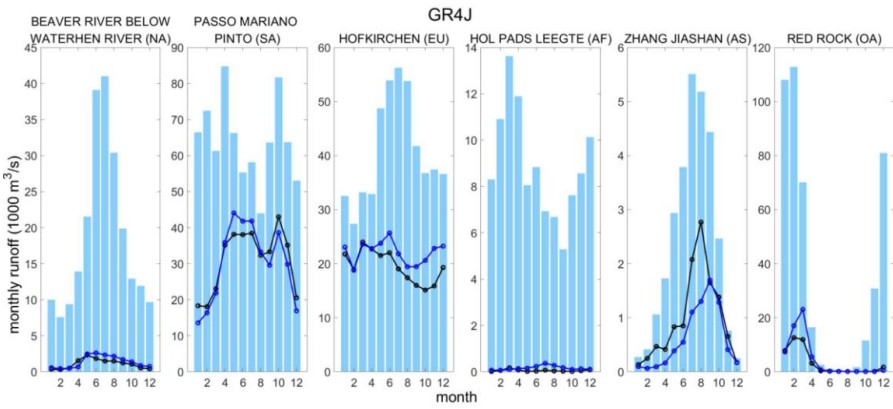

**Figure 11: Comparison of long-term intra-annual variations of simulated and observed runoff at six continents.. Note different ordinate scales.**

### 3.5 Global and continental water resources

Using the established GHMs, the global and continental water resources were calculated and compared to those calculated using other GHMs in literatures (Table 6). The spatial variability of long-term (1982-2015) average annual runoff is presented





in Fig. 12. Generally, four GHMs show similar results at 2277 catchments. However, the estimated global water budget from the GHMs are not always consistent with each other.

The performances of four GHMs were consistent with each other at the catchment scale (see Fig. 10). However, when it comes to the grid cell scale, the differences among 4 GHMs become larger. For example, the simulated continental runoffs from the GR4J-G and HMETS-G are larger than those from the SIMHYD-G and XAJ-G, especially in Asia and Africa. Furthermore, the global long-term annual runoff estimated by GHMs is 48309 km$^3$/yr for GR4J-G, 42539 km$^3$/yr for SIMHYD-G, 42089 km$^3$/yr for XAJ-G and 45157 km$^3$/yr for HMETS-G. The annual runoff differences between four GHMs' are even higher than the value of annual runoff in Africa. This indicates the hydrological model structure differences bring large uncertainty in global water budget estimates.

The six individual simulations of global long term average water resources showed in Table 6 encompass a range between 29500 and 48300 km$^3$/yr. The difference between these individual global water resource simulations is 18800 km$^3$/yr, which is even much higher than the global consumptive water use estimated using Global Water Use Model of WaterGAP 2 (1250 km$^3$/yr in 1995). In fact, it is difficult to compare different GHMs, for there are different time periods, data quality, spatial and temporal resolution and so on of the data used in the building of the GHMs. Our study gives relatively more consistent results among the 4 GHMs due probably to the reason that same data (amount, period) are used. Moreover, the artificial impact on runoff cannot be ignored, because of the regulation, water extraction and re-routing of water to other basins have influence on the runoff estimation. In present study, catchments smaller than 50,000 km$^2$ were used to minimize these impacts. In the future studies, a reliable global precipitation and runoff database which considered anthropogenic influences could be used for improvements.

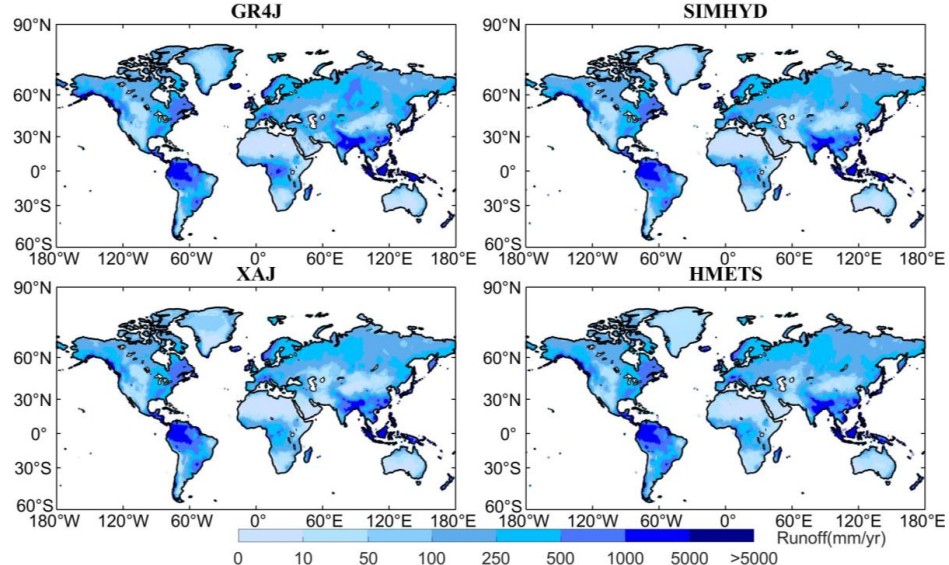

**Figure 12: Long-term average annual total runoff from land and open water fraction of cell (time period 1982–2015), in mm/yr**




**Table 6. Global and continental runoff estimates in km³/year (many data may not be directly comparable because of different continental boundaries and averaging periods)**

| long-term average annual runoff (km³/yr) | K78 | O01 | D03 | GRDC | W07 | GR4J-G | SIMHYD-G | XAJ-G | HMETS-G |
|---|---|---|---|---|---|---|---|---|---|
| Global (except Antarctica) | 44560 | 29485 | 36687 | 40533 | 38605 | **48309** | **42539** | **42089** | **45157** |
| Europe | 2970 | 2191 | 2763 | 3083 | 3669 | **3322** | **3168** | **3219** | **3325** |
| Asia | 14100 | 9385 | 11234 | 13848 | 13611 | **18050** | **14915** | **14635** | **16412** |
| Africa | 4600 | 3616 | 3529 | 3690 | 3738 | **4918** | **3753** | **3764** | **3694** |
| North America[a] | 8180 | 3824 | 5540 | 6294 | 7009 | **7577** | **6871** | **7061** | **7599** |
| South America | 12200 | 8789 | 11382 | 11897 | 9448 | **12425** | **11832** | **11764** | **12369** |
| Oceania[b] | 2510 | 1680 | 2239 | 1722 | 1129 | **2017** | **1999** | **1646** | **1758** |

Bolded columns are from this study

K78, Korzun et al. (1978), Table 157, time period not specified.

O01, Oki et al. (2001), Table 2, land-suface models and TRIP routing model, time period 1987–1988.

D03, Döll et al. (2003), Table 1, model WGHM, time period 1961–1990.

GRDC (2004), time period ''approximately'' 1961–1990.

W07, Widén-Nilsson et al. (2001), Table 2, model WASMOD-M, time period

[a] Includes Greenland, exceptW07 who only simulated a minor part of Greenland.

[b] Oceania is defined as Australia, New Zeeland, Papua New Guinea and some small Islands.

**4. Conclusions**

By conducting a comprehensive evaluation using five regionalization methods and four daily conceptual hydrological models

over a wide range of catchments around the world, the best performed global regionalization method at the catchment scale was found. According to the catchment-scale regionalization results, the optimal GSRS produced parameter maps for all grid cells (0.5°×0.5°) covering the global land area, except for Antarctica and Arctic region, were obtained. On the basis of the GSRS, for grid cells with a mean distance less than 1500 km to donors, the calibrated parameter sets of the 5 nearest donor catchments were transferred by using SPI-OUT method. And for those grid cells with a mean distance larger than 1500 km, the

parameters were extracted from PSDI-OUT method. Thus, the framework was established according to the GSRS, conceptual hydrological models and NRF method. The main conclusions are:

- The SPI-OUT method offers the best results with the largest KGE value using the 0.5 efficiency threshold for all hydrological models, and the optimal number of donor catchments lies between 3 and 6 for the output averaging option.

- The median KGE values of the GHMs were smaller than those obtained by using the best performed catchment scale regionalization methods (SPI-OUT), but were close to those obtained by using the other catchment scale distance/attributes-based regionalization results. They were about half of that obtained using calibrated parameters for all 2277 catchments.



- The continental runoff estimates of GR4J-G and HMETS-G were larger than that of SIMHYD-G and XAJ-G, especially for Asia and Africa. The global long-term average annual runoff estimated by GHMs are 48309 (for GR4J-G), 42539 (for SIMHYD-G), 42089 (for XAJ-G) and 45157 (for HMETS-G) km$^3$/yr.

- The proposal framework can be used with any GHM for estimating global water resources, even though uncertainty exists in terms of using different conceptual models.

**Competing interests.** The author declares that there is no conflict of interest.

**Acknowledgement.** This work was partially supported by the National Key Research and Development Program of China (No. 2017YFA0603704), the National Natural Science Foundation of China (Grant Nos. 51779176, 51539009), the Overseas Expertise Introduction Project for Discipline Innovation (111 Project) funded by Ministry of Education of China and State Administration of Foreign Experts Affairs P.R. China (Grant No. B18037), the Research Council of Norway (FRINATEK Project 274310) and the Thousand Youth Talents Plan from the Organization Department of CCP Central Committee (Wuhan University, China).

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
