# Peer review of "A framework to regionalize conceptual model parameters for global hydrological modeling"

_Hydrology and Earth System Sciences, 2020_

## Referee Comment (RC1) · Anonymous Referee #1 · 12 Jul 2020

This manuscript examines a variety of regionalization approaches applied to regionalize parameters of four catchment-scale conceptual models to global grid cells. The performance of standard regionalization techniques based on spatial proximity, physical similarity and the combination of both is examined for several thousand catchments world-wide and is compared to the performance with at-site calibrated parameters. The combination of best-performing regionalization approaches are used to interpolate parameters from gauged locations to the grid cells word-wide and global water balance components are estimated using four different conceptual hydrological models. The comparison of regionalization methods for global scale hydrological models. However, it is not clear how the framework proposed in this study advances the fidelity of

global hydrological models. The components of the proposed framework are not clear defined making it rather difficult to understand the novelty and the advantages of this work compared with previous studies. Little insights and discussion is provided on the effect of parameter uncertainty on the estimates of global water balance components. Moreover, the introduction of more recent works on model parameter regionalization, especially the work tackling parameter discontinuity for regional and global studies is missing. Some critical assumptions (e.g., on independence among catchment descriptors or that catchments with similar catchment descriptors have similar model parameters) were neither tested nor critically discussed. The reported differences in performance of different regionalization methods and models is minimal. Finally, several missing methodological aspects regarding distance calculation between the catchments, unclear distinction between calibration and evaluation catchments and interpolation to the global grid cells makes it difficult to evaluate the credibility of this study. Therefore, I think a substantial revision of the manuscript is required. Please find my detailed comments attached.

Please also note the supplement to this comment: https://www.hydrol-earth-syst-sci-discuss.net/hess-2020-127/hess-2020-127-RC1supplement.pdf

**Supplement:**

**Review of the manuscript „A framework to regionalize conceptual model parameters for global hydrological modeling"**

This manuscript examines a variety of regionalization approaches applied to regionalize parameters of four catchment-scale conceptual models to global grid cells. The performance of standard regionalization techniques based on spatial proximity, physical similarity and the combination of both is examined for several thousand catchments world-wide and is compared to the performance with at-site calibrated parameters. The combination of best-performing regionalization approaches are used to interpolate parameters from gauged locations to the grid cells word-wide and global water balance components are estimated using four different conceptual hydrological models.

The comparison of regionalization methods for global scale hydrological modeling has an immense importance for reliable estimation of global water resources. However, it is not clear how the framework proposed in this study advances the fidelity of global hydrological models. The components of the proposed framework are not clear defined making it rather difficult to understand the novelty and the advantages of this work compared with previous studies. Little insights and discussion is provided on the effect of parameter uncertainty on the estimates of global water balance components. Moreover, the introduction of more recent works on model parameter regionalization, especially the work tackling parameter discontinuity for regional and global studies is missing. Some critical assumptions (e.g., on independence among catchment descriptors or that catchments with similar catchment descriptors have similar model parameters) were neither tested nor critically discussed. The reported differences in performance of different regionalization methods and models is minimal. Finally, several missing methodological aspects regarding distance calculation between the catchments, unclear distinction between calibration and evaluation catchments and interpolation to the global grid cells makes it difficult to evaluate the credibility of this study. Therefore, I think a substantial revision of the manuscript is required. Below I present my detailed comments.

**General comments**

1. Introduction should clearly define the gaps that currently exist in parameter regionalization for regional and global hydrological models and should clearly state how this study tackles these problems. Currently, the Introduction is rather structured as a listing of performed studies without assessment of their advantages and disadvantages for global scale hydrological modeling making it difficult to understand the novelty of this study. In my opinion this study rather uses well-established techniques with known flaws and merely compares their performance at global scale in terms of single performance metric. A more clear novelty statement should be presented to make it clear how this study solves or advances current issues in global scale modeling and regionalization of model parameters. I miss also the discussion on the issue of model parameter discontinuity for regional and global hydrological modeling (e.g., Samaniego et al., 2017) and how the proposed framework can deal with it.

2. The description and the components of the proposed framework are not clear. It is not clear from the manuscript if the framework is actually a combination of all steps (i.e., different regionalization methods + different catchment scale conceptual models + interpolation to grid cells) or if it is just the selection procedure for regionalization methods. It is not clear which features does make it a framework and not a simple sequence of methodological steps. Moreover, the workflow itself has to be clarified too. Specifically, it is not clear if the grids

that correspond to donor catchments were preserved or regionalized as well. It is not clear which portion of catchments was left for evaluation of global regionalization. Please, also see my specific comments regarding these issued below.

3. Regionalization methods examined here refer to so called two step similarity approaches (Wallner et al., 2013) where donor catchments are independently calibrated and then their parameters are regionalized based on various similarity metrics (e.g., spatial proximity or physical similarity or both). Such approaches were reported to suffer from equifinality problem (Bárdossy, 2007; Götzinger and Bárdossy, 2007). Alternative one step approaches (e.g., Hundecha and Bárdossy, 2004; Samaniego et al., 2010; Mizukami et al., 2017) that were developed to tackle this problem are not discussed in this study. Only a very simplistic uncertainty analysis was performed in this study, but nevertheless exposed large equifinality problem for all 4 models. It was not shown how the detected parameter uncertainty has affected regionalization performance and has propagated to the global water balance estimates. Little discussion was provided on the reliability of final global water balance estimates.

4. Several crucial assumptions were neither tested not critically discussed (e.g., independency of catchment descriptors used in this study; link between physical and hydrological similarity). Moreover, several methodological details are missing or ignored (e.g., catchments affected by human activities are not rigorously filtered out; no indication on spatial discretization used in the models; no indication if the distance between catchments was calculated based on outlets or centroids; no clear distinction between donor and evaluation catchments), making it difficulty to the judge the credibility of this study. Please see my specific comments for details.

5. Writing and structure of the paper needs to be improved. Moreover, please check consistency of tense and plurals/singulars of nouns and verbs.

**Specific comments**

**Line 36-43:** According to this sentence in Line 36-38 land surface scheme models are the most commonly used global hydrological models. In the next two sentences it is stated that due to large biases of these models global hydrological models are developed. These statements are confusing, please clarify.

**Line 44-60:** I miss here an overview of the approaches that tune model parameters of global hydrological models (e.g., WaterGAP by Döll et al., 2003).

**Line 71:** I do not think that regionalization technique proposed in Samaniego et al. 2010 (referred here is Luis et al., 2010) can be attributed to clustering or hydrological classification.

**Line 74:** I would welcome here more insights on why despite numerous studies on comparison of regionalization approaches it is still not clear which of them perform better and how this study contributes to identification an appropriate method.

**Line 76-77:** Accounting for uncertainties in global hydrological modeling is an important issue that indeed was not tackled sufficiently in the previous studies. However, it is not clear how this study accounts for uncertainties? From what I have seen later in the paper, the authors indicated presence of considerable parameter uncertainty in each model and showed that global runoff estimates vary considerable among tested models. But does it really mean that the uncertainty is accounted for in this study?

**Line 96:** Please specify the spatial resolution of the precipitation product.

**Line 101-102 and Table 1:** I urge the authors to carefully examine the assumption about independency among catchment descriptors. I am afraid some of them have strong dependencies (e.g., slope and elevation or aridity index and mean annual potential evaporation). Instead of rather uninformative Table 1 that presents mean, max and min values of catchments descriptors globally (clearly a wide variety is to be expected here due to the global focus of the study), a correlation matrix would instead confirm or oppose the assumption of independency. The second assumption on the existence of well-behaved relationship between catchment descriptors and model parameters is certainly more difficult to prove, but at least a note on possible issues with this assumption has to be stated (see e.g., Odin et al., 2010 or Merz et al., 2020).

**Line 111-119:** Indeed, presence of regulated catchments creates a considerable obstacle for hydrological modeling. Does the selection based only on catchment area is able to filter out the regulated catchments? I suggest authors to include additional criteria that are customary in large-scale hydrological modeling (e.g., remove catchments with large dams, Lehner et al., 2011, Grill et al., 2019; test the closure of water balance, Beck et al., 2016).

**Section 2.3:** The rationale on selection of these four models is not clear to me. It is noted (Line 76-77) that by considering models of different structure and concepts one might account for the model uncertainty. However, from the description provided for each model in this section, it is not clear which differences in their concepts and structures apart from the number of parameters exist. Moreover, I miss here the indication if the models were applied in lumped or in a distributed fashion.

**Line 156-157:** According to the catchment selection rules listed in Line 115-116 catchments with at least 5 years of observations were selected. This would mean around 3.5 years for calibration in these catchments. How many catchments have such short calibration period? Do you think such short calibration period might increase parameter uncertainty?

**Line 167-177:** I would welcome here the rationale on selection of these regionalization methods. Most of the approaches used here are two step similarity approaches, meaning that in the first step the model parameters are identified at gauged locations independently from similarity or spatial proximity of the catchments. In the second step calibrated parameters are regionalized assuming that closer catchments or more physically similar catchments have similar model parameters. Such approaches are not able to account for equifinality of model parameters (Bardossy, 2007; Götzinger and Bardossy, 2007). Therefore, neighboring catchments and even physically similar catchments not necessarily obtain similar parameter sets during calibration (Oudin et al., 2010). The methods used in this study are likely to suffer from a similar problem. This issue has to be addressed here and put into perspective of one step similarity approaches that were specifically developed to target this issue (e.g., Samaniego et al., 2010; Wallner et al., 2013; Mizukami et al., 2017).

**Section 2.5:** Several important details are missing in this section. Was the best performing parameter set regionalized or a mean parameters from 10 best sets mentioned in Lines 310-315? Was spatial proximity defined based on geographical position of the outlet or of the catchment centroid? Was the whole parameter set transferred or each single model parameter was regionalized independently? No details on regression method is provided. It should be specified what kind of regression model was used, which parameter estimation method was used and if all catchment descriptors were used to build the regression model. It should be

also reported if the regression model was built for each individual model parameter or for the whole set.

**Line 182-184:** How accurate is distance calculation on these projected coordinates globally? Why not to simply calculate geographical distances? Does "catchment position" refers to catchment outlet or catchment centroid?

**Line 187:** I think Luis et al. 2010 and Samaniego et al. 2010 is the same study.

**Line 210-211:** What does this sentence mean? Please clarify which catchments were used for the evaluation of regionalization techniques.

**Line 213-218:** It is not clear how exactly the parameters were regionalized to grid cells. Were the calibrated parameters of donor catchment preserved during regionalization? If they were preserved, were they assigned to catchment centroid or catchment outlet? How other grids within donor catchments were assigned?

**Section 2.7:** It is not clear what GSRS states for. Till Line 219 part I had an impression that GSRS is the proposed framework for the selection of an appropriate regionalization technique to transfer parameters of catchment scale conceptual models to the global grid cells. Therefore, the statement in this sentence comes surprising. Please clarify if GSRS is actually the proposed regionalization framework mentioned elsewhere in the manuscript. Please clarify its description and components.

**Line 221-225:** It is not clear to me how the Network Response Routing converges grid cell runoff to catchment streamflow. Please clarify it and indicate what are the two parameters mentioned here. Moreover, please clarify how these two parameters exactly were calibrated. If these parameters were not regionalized how is it possible to obtain catchment streamflow at ungauged locations (i.e., grid cells)? Furthermore, I find merging of the conceptual model parameters with Network Response Routing parameters rather confusing. Please explain why this step was necessary.

**Line 226:** Does this mean that Network Response Routing is the part of framework? So does it mean that the framework is actually GSRS+NRF?

**Line 241-242:** Is precipitation gauge density is the only reason for good performance of hydrological models in these regions? Could these spatial variability in model performance result from inability of selected models to simulate discharge in drier areas compared to wetter areas?

**Line 266-268:** Where these results can be seen?

**Line 273-276:** Why should poorly calibrated donors be considered in further regionalization? Is likely that by using the poorly performing parameter set additional uncertainty will be added to the regionalization. What was the criteria to choose the threshold value?

**Line 285-289 and Figure 5:** Are the difference in the performance of different regionalization methods considerable? I see considerable differences among calibration, global mean and regression method compared to all other regionalization approaches. Apart from that the differences are minimal. Taking into account that it seems that only one best performing parameter set was used for regionalization, I am wondering if these differences would still be visible after accounting for the parametric uncertainties.

**Section 3.2:** I am missing here the discussion on such poor performance of the regression method. Could it be linked to the multicollinearity of catchment descriptors? Was the whole parameter set regionalized with this method or were regression models built for each model parameter individually? Was a global regression model built or was it region-specific?

**Figure 6:** I am not sure that I see in this Figure that 1500 km is an optimal threshold. For GR4J 1300 km and for HMETS and XAJ 1200 km seems to be more appropriate. How exactly was threshold of 1500 km identified? Is this threshold sufficiently robust given that there are large fluctuations in the performance of the methods around this distance (e.g., for GR4J for 1400 km the methods based on both physical similarity and distance outperform all other methods for all catchments, while for 1300 km and 1500 km spatial proximity is the best performing method). Why the x axis in the subplots differ? For some it reaches till $18*10^5$ m for others only till $15*10^5$ m. For all subplots the last bar seems to be cut. Please clarify what "Physical Similarity DIS" in the legend refers to?

**Line 301-304:** The difference in performance between the models is of the same magnitude as between the regionalization methods (apart from global mean and regression method). Therefore, I find the conclusion that model structure has an insignificant effect while the choice of regionalization method is important rather inconsequent.

**Line 310-320:** This is a description of methods and should be moved to the methods Section.

**Line 311-312:** Please clarify which criteria was used to select sufficiently different parameter sets.

**Line 328-329:** This statement requires a citation. If this is a finding of this work consider reformulating this sentence accordingly.

**Line 329-330:** One way to quantify the influence of parametric uncertainty for regionalization results is to perform regionalization using all equifinal parameter sets (i.e., 10 best performing parameter sets in this study) and analyze the differences in regionalization performance and global runoff estimates.

**Line 348-352:** How the deterioration in performance between the regionalization at catchment scale and regionalization at grid cell can be explained? If the same donor catchments were used and the best performing regionalization methods were selected why deterioration occurs at grid cells? Is this deterioration is caused solely by averaging over the grid cell?

**Line 352:** Does "-G" stands for global version of respective hydrological model? Please introduce appropriately this new notation and explain how the global version is different from catchment version. Later (Line 354, 356, Table 4 and elsewhere) "-G" notation is not used anymore, although it seems like the global set up of the models is meant there as well.

**Line 359-361:** The values reported here and in Table 4 seems to be median KGE for all catchments (including donor catchments), while Beck et al. (2016) has reported KGE values for evaluation catchments only. Therefore, this comparison is not fair. Moreover, please consider reporting regionalization performance for evaluation catchments only. Without these results it is difficult to judge the credibility of the proposed framework.

**Line 364-366:** In the later part of this paper the authors refer to the 4 catchment scale conceptual models regionalized to grid cells as global hydrological models (GHMs). The argumentation provided in the review of global hydrological models by Sood and Shmakhtin

(2015) that global hydrological models were built for macroscale water resources management and not for streamflow simulation cannot be justified here, as Sood and Shmakhtin (2015) do not refer to regionalized catchment scale conceptual models but to actual global hydrological models that either are the components of general circulation models or stand-alone hydrological models with very few or no calibrated parameters that run directly at the global scale. Instead, the models derived here were specifically designed to improve streamflow simulation and calibrated to the observed streamflow.

**Figure 9:** Please clarify in the caption if the cumulative curves are showed for all catchments or evaluation catchments only. It seems that y axis is cut at 0, please indicate it in the caption. Clarify what CAL and GSRS stands for in the caption.

**Table 4:** Please clarify what "framework" stands for? Is it an equivalent of "GSRS" from Figure 9?

**Line 383-384:** According to Table 5 SIMHYD-G performs the best. Please clarify the statement in this sentence.

**Line 389-390:** Although I generally agree that the balance between model flexibility and complexity is very important, I would say that the results of this study rather show the opposite. Minor differences in KGE between the models (0.01 of KGE between the best and the worst one; this is exactly the threshold for performance difference that the authors selected to identify equifinal parameters earlier) indicates that the choice of the model played rather negligible role here. Moreover, earlier (Line 300-303) it was stated that model structure played insignificant role for regionalization performance in this study supporting rather the other conclusion of this work that potentially any model can be used in the proposed framework.

**Line 422-424:** A reference is needed for these estimates.

**Line 426-430:** This portion makes me wonder why the available global datasets on anthropogenic influence (see my comment to Line 111-119) were not considered to filter out the affected catchments. Are there any prospects on obtaining reliable global precipitation datasets that can be used in the future studies?

**Line 462:** Usage of "GHM" in this context is confusing, as in this study the GHM were built from catchments-scale conceptual models by combining them with various regionalization approaches. If I understood correctly these steps combined are the framework proposed in this study. Consider using "conceptual model" instead.

**Editorial comments**

**Abbreviations:** The manuscript is oversaturated with abbreviations. Some of these abbreviations are never used after their introduction (e.g., LSS, PUB, AOF), some other abbreviations are only used once or twice (e.g., IDW, AM, CDF). Consider omitting using unnecessary abbreviations in the manuscript and especially in the abstract (e.g., NRF, IDW, KGE).
**Line 15:** Why Network Response Routing is abbreviated as NRF. Should it be NRR instead?
**Line 54:** a priori
**Line 67, 71 and elsewhere:** Samaniego et al., 2010 instead of Luis et al., 2010
**Line 76:** Moreover
**Line 85:** delete "widely"
**Line 383:** "proposed" instead of "proposal"

**Line 384:** I think you mean here that it was not confirmed in this study.
**Line 385:** "that" instead of "who"
**Line 387:** "is not reduced" instead of "does not reduce"
**Line 389:** "in" instead of "on"
**Line 389:** might be
**Line 453:** "highest" instead of "largest"
**Figure 1:** Please clarify if the Figure actually shows the location of catchment outlets or centroids.
**Figure 3:** Please indicate in the caption if model efficiency corresponds to calibrated model parameters here.
**Figure 4:** Why global mean and regression methods are not shown here? Please label the x axis. It would be useful if you will select the same colors for performance classes as in Figure 3.
**Figure 5:** please specify what CAL stands for. If it stands for calibration, consider moving its box to the first position. Please clarify box plot structure in the caption since whiskers and the outliers are not plotted here.
**Table 2:** Why only SPA and SPI-OUT are mentioned in the caption?
**Table 6:** Please indicate if any model was used to provide runoff estimate in the study of Korzun et al. (1978) and GRDC; please indicate which period was modelled in Widén-Nilsson et al. (2001). What does "approximately" means for GRDC time period?
**Figure 11:** Please indicate the scale for precipitation. Consider transforming discharge to mm/day to make it comparable with precipitation. Please remove a black line on the right and the double full stop in the caption.

**References**

Bárdossy, A. (2007). Calibration of hydrological model parameters for ungauged catchments. *Hydrology and Earth System Sciences*, *11*(2), 703–710. https://doi.org/10.5194/hess-11-703-2007

Götzinger, J., & Bárdossy, A. (2007). Comparison of four regionalisation methods for a distributed hydrological model. *Journal of Hydrology*, *333*(2–4), 374–384. https://doi.org/10.1016/j.jhydrol.2006.09.008

Grill, G., Lehner, B., Thieme, M., Geenen, B., Tickner, D., Antonelli, F., … Zarfl, C. (2019). Mapping the world's free-flowing rivers. *Nature*, *569*(7755), 215–221. https://doi.org/10.1038/s41586-019-1111-9

Hundecha, Y., & Bárdossy, A. (2004). Modeling of the effect of land use changes on the runoff generation of a river basin through parameter regionalization of a watershed model. *Journal of Hydrology*, *292*(1–4), 281–295. https://doi.org/10.1016/j.jhydrol.2004.01.002

Lehner, B., Liermann, C. R., Revenga, C., Vörömsmarty, C., Fekete, B., Crouzet, P., … Wisser, D. (2011). High-resolution mapping of the world's reservoirs and dams for sustainable river-flow management. *Frontiers in Ecology and the Environment*, *9*(9), 494–502. https://doi.org/10.1890/100125

Merz, R., Tarasova, L., & Basso, S. (2020). Parameter's Controls of Distributed Catchment Models—How Much Information is in Conventional Catchment Descriptors? *Water Resources Research*, *56*(2), 1–18. https://doi.org/10.1029/2019WR026008

Mizukami, N., Clark, M. P., Newman, A. J., Wood, A. W., Gutmann, E. D., Nijssen, B., … Samaniego, L. (2017). Towards seamless large-domain parameter estimation for hydrologic models. *Water Resources Research*, *53*(9), 8020–8040. https://doi.org/10.1002/2017WR020401

Oudin, L., Kay, A., Andréassian, V., & Perrin, C. (2010). Are seemingly physically similar catchments truly hydrologically similar? *Water Resources Research*, *46*(11), 1–15. https://doi.org/10.1029/2009WR008887

Samaniego, L., Kumar, R., Thober, S., Rakovec, O., Zink, M., Wanders, N., … Attinger, S. (2017). Toward seamless hydrologic predictions across spatial scales. *Hydrology and Earth System Sciences*, *21*(9), 4323–4346. https://doi.org/10.5194/hess-21-4323-2017

Wallner, M., Haberlandt, U., & Dietrich, J. (2013). A one-step similarity approach for the regionalization of hydrological model parameters based on Self-Organizing Maps. *Journal of Hydrology*, *494*, 59–71. https://doi.org/10.1016/j.jhydrol.2013.04.022

---

## Referee Comment (RC2) · Anonymous Referee #2 · 26 Aug 2020

The paper compares regionalization approaches at global scale. I like that the paper tests many different regionalization approaches but, unfortunately, one important approach is missing (more below). In addition, the English should be improved by a native English speaking person.

The "spatial proximity method" may well yield the highest KGE, but it cannot be used at global scale due to the lack of gauges in many regions. As an example, grid cells in the southern Ecuadorian Andes would receive parameters from donor catchments located in the Amazon, which doesn't make any sense. So while this approach may give you the best performance scores, it is not actually a method that should be used at global scale. This should be explicitly mentioned in the abstract and the conclusion.

Even if the "mean distance between the donor catchments and the target catchment is

no more than 1500 km" the actual difference in climate and landscape could be huge.

"The k value of the PCRGLOBWB was determined based on the drainage [...]" This was paraphrased (like many other sentences) from Beck et al. (2016), but is this still the case in PCGLOBWB 2.0 (https://gmd.copernicus.org/articles/11/2429/2018/gmd-11-2429-2018.html)?

Table 1: The aridity index can become extremely high in desert regions (higher than 100) so I'm surprised that this isn't reflected in your mean. Did you apply some sort of mask, or did you cap the values before calculating the mean? This info needs to be in the caption.

Table 1: Which datasets did you use for the different attributes?

Section 2.2: Hydro1K is a very outdated dataset. A newer dataset should be used, maybe HydroSHEDS or MERIT.

Figure 1: Which KG map did you use? These are too many classes to distinguish, and the legend is a mess. Better to condense to the 5 major classes.

Equation 1: This is the old KGE, the new KGE has a slightly better formulation of the variability component. See Kling et al. (2012; https://doi.org/10.1016/j.hydrol.2012.01.011).

Section 2.5: Among the five tested regionalization methods, one important approach is missing, and it is most likely the best approach. It is the approach where the model parameters and the regression equations (relating the catchment attributes to the model parameters) are optimized simultaneously. See for example https://agupubs.onlinelibrary.wiley.com/doi/abs/10.1029/2019JD031485 and https://agupubs.onlinelibrary.wiley.com/doi/full/10.1029/2008wr007327. This approach should definitely be included.

"The regression-based method assumes that a well-behaved relationship exists in the observable catchment characteristics and model parameters" which in reality is almost

never the case due to parameter equifinality and therefore this approach rarely works. Hence my suggestion to test the other regionalization approach.

"The SP method assumes that nearby catchments should have similar behavior for climate and catchment conditions (features) varying uniformly in space." "Nearby" could be several thousand kilometers away so this approach should never be used at global scale.

Another limitation of the study is that lumped catchment attribute and model parameter values are used, despite the often large heterogeneity within catchments. This limitation is addressed in several studies (e.g., Samaniego 2008 and Beck 2020) and should at least be discussed somewhere in the paper.

"The results show that the distributions of model efficiency of four hydrological models are similar to each other and indicate that the difference between hydrological models was negligible in the model calibration and validation, which is in line with previous studies (Beck et al., 2016; Vetter 245et al., 2015; Demirel et al., 2015)." This is definitely not in line with previous studies, which generally found large differences between models and a considerable difference between calibration and validation scores. You actually also obtained substantial differences between calibration and validation scores (figure 2)!

Figures 3 and 10: This figure is impossible to interpret. https://www.climate-lab-book.ac.uk/2016/why-rainbow-colour-scales-can-be-misleading/

Figure 5: The figure is a bit difficult to interpret, maybe use vertical instead of diagonal x-axis labels, and apply some coloring to group similar methods together?

Figure 6: There is no information about the number of catchments representing each bar, so a particular bar could be represented by just 1 catchment. I can't think of a solution right now, but this information should not be hidden.

Figure 8: A non-linear color scale might better.

Table 6: "many data may not be directly comparable because of different continental boundaries and averaging periods." A solution would be to only use estimates representing the same area. Also, why report values from a study >40 years old (Korzun 1978)? Considering adding GSCD estimates (http://www.gloh2o.org/gscd/).

---

## Author Comment (AC1) · 26 Oct 2020

Dear Anonymous Referee #1,

We would like to thank the reviewer for the time taken in reviewing this paper. All comments will be incorporated into the revised manuscript. Please check the attached point-by-point replies. We will make the revisions to the manuscript as suggested.

Please also note the supplement to this comment:
https://hess.copernicus.org/preprints/hess-2020-127/hess-2020-127-AC1-supplement.pdf

127, 2020.

**Supplement:**

Replies to Referee #1

**A framework to regionalize conceptual model parameters for global hydrological modeling**

Wenyan Qi, Jie Chen, Lu Li, Chong-yu Xu, Jingjing Li, Yiheng Xiang and Shaobo Zhang

We would like to thank the reviewer for the constructive comments and advice. We have provided detailed responses to each comment below and will revise the manuscript accordingly. For clarity, comments are given in black, and our responses are given in blue.

This manuscript examines a variety of regionalization approaches applied to regionalize parameters of four catchment-scale conceptual models to global grid cells. The performance of standard regionalization techniques based on spatial proximity, physical similarity and the combination of both is examined for several thousand catchments world-wide and is compared to the performance with at-site calibrated parameters. The combination of best-performing regionalization approaches are used to interpolate parameters from gauged locations to the grid cells word-wide and global water balance components are estimated using four different conceptual hydrological models.

We appreciate the reviewer's evaluation, summary and professional comments on our manuscript. Please find our point-by-point responses below.

The comparison of regionalization methods for global scale hydrological modeling has an immense importance for reliable estimation of global water resources. However, it is not clear how the framework proposed in this study advances the fidelity of global hydrological models. The components of the proposed framework are not clear defined making it rather difficult to understand the novelty and the advantages of this work compared with previous studies.

Reply: We thank the reviewer for his/her rigorous reading and professional comments. Sorry that we failed to describe some parts clear enough in the original manuscript. In response to the reviewer's comments and advice, the following revisions will be made.

To improve the clarity of the proposed framework, a figure (Figure R1) is to be added in the revised manuscript to show the proposed framework in this study. The framework involved the following steps: (1) calibrate catchment hydrological models to obtain parameter information for regionalization, (2) compare different regionalization methods at the catchment scale and select the optimal global Scale Regionalization Scheme (GSRS), (3) parametrize gridded version of hydrological models using GSRS and build GHMs, and (4) calculate catchment streamflow using the runoff routing method.

[Figure]

Figure R1. The schematic diagram of the main steps of the proposed framework.

Previous GHMs (global hydrological models) based on prior parameterizations (expert opinion, hydrologic theory, field data, case studies, or data sets of questionable quality) generally consider model parameters as individual values and set more or less globally uniform parameters, which may result in insufficient streamflow simulations (Beck et al., 2016). Some GHMs based on model parameter regionalization obtain global gridded parameter maps by transferring calibrated parameter sets from the selected donor catchments to grid cells and illustrate the effectiveness of regionalization methods in macro-scale runoff simulation (e.g. Widén-Nilsson et al., 2007, 2009; Beck et al., 2016). However, they simply used one specific scheme to calculate global parameter maps rather than comparing the widely used regionalization approaches at the global scale, failed to select the optimal scheme for the regionalization of GHMs. To overcome these two limitations, this study compared the performance of widely used regionalization methods on global scale by using 4 catchment scale hydrological models and selected the optimal global scale regionalization scheme. The optimal global scale regionalization scheme is then used to generate parameter values for gridded global models. The proposed framework for building GHMs and regionalization schemes can easily be adopted by other models. The main idea is to test and apply the best performed catchment scale regionalization schemes to accurately estimate the grid parameters on global scale for global water resources assessment and use multiple models to account uncertainty.

References:
Beck, H. E., van Dijk, A. I. J. M., de Roo, A., Miralles, D. G., McVicar, T. R., Schellekens, J., and Bruijnzeel, L. A.: Global-scale regionalization of hydrologic model parameters,

Water Resour Res, 52, 3599-3622, 10.1002/2015wr018247, 2016.

Widén-Nilsson, E., Halldin, S., and Xu, C.-Y.: Global water-balance modelling with WASMOD-M: Parameter estimation and regionalisation, J Hydrol, 340, 105-118, 10.1016/j.jhydrol.2007.04.002, 2007.

Little insights and discussion is provided on the effect of parameter uncertainty on the estimates of global water balance components.

Reply: To consider parameter uncertainty, the following analysis is done and is to be added in the revision. According to the normalization factor (NF) values calculated by using ten sets of calibrated hydrological model parameters in the original manuscript, all four models are subjected to the effects of parameter equifinality. To further illustrate the influence of parameter equifinality on regionalization, we randomly selected one of the ten calibrated parameter sets in the catchment scale regionalization under the threshold of 0.5 over 2277 catchments. The regionalization performance of the randomly selected parameter set (Figure R2) is consistent with the results of the original regionalization method (Figure 9-10). A similar conclusion was also drawn by Arsenault and Brissette (2014), i.e., equifinality does not contribute significantly to the overall uncertainty in the applications of regionalization methods. Figure R2 will be added in the revised manuscript.

[Figure]

Figure R2. Comparison of model efficiencies on ungauged catchments using several regionalization schemes between original parameter sets and randomly selected parameter sets. The red and black boxes represent the performance of the original and randomly selected parameter sets, respectively. The blue box represents the calibrated results.

References:

Arsenault, R., and Brissette, F. P.: Continuous streamflow prediction in ungauged basins: The effects of equifinality and parameter set selection on uncertainty in regionalization approaches, Water Resour Res, 50, 6135-6153, 10.1002/2013wr014898, 2014.

Moreover, the introduction of more recent works on model parameter regionalization, especially the work tackling parameter discontinuity for regional and global studies is

missing.

Reply: To enhance the literature review, the recent works on model parameter regionalization as well as the work tackling parameter discontinuity for regional and global studies will be added in the revised manuscript. The introduction part will be rewritten and compacted in the revised manuscript. For more details, please see the response to the general comments 1.

Some critical assumptions (e.g., on independence among catchment descriptors or that catchments with similar catchment descriptors have similar model parameters) were neither tested nor critically discussed.

Reply: Thanks. We will provide more clear clarification in the revision that the catchment descriptors used in this study are primarily based on the comprehensive reviews of regionalization methods made by He et al. (2011) and Razavi and Coulibaly (2013), in which the number of times that catchment descriptors used in other studies were counted after reviewing the regionalization methods. In order, the most often used descriptors are area, slope, percentage of area covered by various terrain types, soil classification, elevation, and drainage density. Based on this information, thirteen catchment descriptors, classified as climate index, terrain characteristics, land use, and soil characteristics, were selected and used in this study (Table 1). In addition to the above mentioned commonly used catchment attributes, "soil clay content" and "water holding capacity" were also used in our study, since some studies (e.g. Garambois et al., 2015) showed their significant impacts on the snow melting process. According to correlation coefficients between catchment descriptors, it is appropriate to keep all these catchment descriptors.

The foundational assumption of the physical similarity method is that catchments with similar attributes should have similar hydrological behaviors (Burn and Boorman 1993; Beck et al., 2016; Yang et al., 2020). Aa one of the most basic and commonly used regionalization method, this method has been successfully used in the regionalization of ungauged catchments all over the world (e.g., North America, Norway, Australia, and France).

A more detailed response can be referred to the specific comments below.

References:
Garambois, P. A., Roux, H., Larnier, K., Labat, D., and Dartus, D.: Parameter regionalization for a process-oriented distributed model dedicated to flash floods, J Hydrol, 525, 383-399, 10.1016/j.jhydrol.2015.03.052, 2015.
He, Y., Bárdossy, A., and Zehe, E.: A review of regionalisation for continuous streamflow simulation, Hydrol Earth Syst Sc, 15, 3539-3553, 10.5194/hess-15-3539-2011, 2011.
Razavi, T., and Coulibaly, P.: Streamflow prediction in ungauged basins: review of regionalization methods, Journal of hydrologic engineering, 18, 958-975, 2013.

The reported differences in performance of different regionalization methods and models is minimal.

Reply: Concerning the relatively small differences in performance of different regionalization methods, we agree with the reviewer that the differences of median Kling-Gupta efficiency (KGE) values between different methods are minor. The same results have been shown in other regionalization studies on catchment scale (Arsenault and Brissette, 2014; Yang et al., 2020). However, according to the cumulative density function (CDF) curves of different regionalization methods, the advantage of SPI-OUT is clear (see Figure 9 in the original manuscript).

Four different models were selected to evaluate whether the best performed method and the effectiveness of the proposed framework are model-independent. The results show that the influence of models on the performance of regionalization is small. However, when it comes to the simulated continental or global runoffs, the difference among the four GHMs can not be ignored.

References:
Arsenault, R., and Brissette, F. P.: Continuous streamflow prediction in ungauged basins: The effects of equifinality and parameter set selection on uncertainty in regionalization approaches, Water Resour Res, 50, 6135-6153, 10.1002/2013wr014898, 2014.
Yang, X., Magnusson, J., Huang, S., Beldring, S., and Xu, C.-Y.: Dependence of regionalization methods on the complexity of hydrological models in multiple climatic regions, J Hydrol, 582, 10.1016/j.jhydrol.2019.124357, 2020.

Finally, several missing methodological aspects regarding distance calculation between the catchments, unclear distinction between calibration and evaluation catchments and interpolation to the global grid cells makes it difficult to evaluate the credibility of this study. Therefore, I think a substantial revision of the manuscript is required. Below I present my detailed comments.

Reply: Sorry that we failed to describe the methodological part clear enough in the original manuscript. The method part mentioned above will be clarified in the response to the specific comments and in the revised manuscript.

General comments
1. Introduction should clearly define the gaps that currently exist in parameter regionalization for regional and global hydrological models and should clearly state how this study tackles these problems. Currently, the Introduction is rather structured as a listing of performed studies without assessment of their advantages and disadvantages for global scale hydrological modeling making it difficult to understand the novelty of this study. In my opinion this study rather uses well-established techniques with known flaws and merely compares their performance at global scale in terms of single performance metric.

Reply: Thanks for the comment, and sorry that we failed to make this clear enough in the introduction of the original manuscript. The introduction will be partly rewritten in the revised manuscript as follows:

*Water resource is one of the most important natural resources that can significantly influence the social and economic development for a region and a country (Parajka et al., 2007; Grill et al., 2019). The management of water resources should be based on a full understanding of the spatial and temporal variation of regional water resources. In particular, problems caused by climate change, increasing water demand due to growing population, water conflicts in multinational river basins, and virtual water trade all reflect the requirements of continental and global-scale hydrological simulations (Döll et al., 2003; Oki and Kanae, 2006; Widén-Nilsson et al., 2007, 2009).*

*Models developed to simulate continental or global water resources can be roughly classified into global hydrological models (GHMs), dynamic global vegetation models (DGVMs), and land surface models (LSMs). Most DGVMs do not include lateral water flows or surface water bodies, and can therefore only be used to assess runoff generation but not streamflow discharge (Döll et al., 2015). The LSM is commonly used as a component of climate models in simulating the energy and water balance at soil, atmosphere, and vegetation interfaces (Haddeland et al., 2011; Bierkens, 2015). However, global climate models are considerably biased in global runoff simulations (Yang and Dickinson, 1996; Sellers et al., 1986; Sood and Smakhtin, 2015). Hence, GHMs focused on the simulation of water resources are developed to simulate (sub-) surface water fluxes and storages. Some of the widely used GHMs include Variable Infiltration Capacity model (VIC, Liang et al., 1994), Water Balance Model–Water Transport Model (WBM-WTM, Vörösmarty et al., 1989), PCRaster GLOBal Water Balance model (PCRGLOBWB, Van Beek and Bierkens 2008; Van Beek et al., 2012), and WASMOD-M (Widén-Nilsson et al., 2007, 2009).*

*Although great progresses have been made in GHMs during the past few decades, the performance of GHMs has still not been sufficiently obtained, partly due to the difficulty of global parameterization. Widén-Nilsson et al. (2007) compared Water and Snow Balance Modeling System Macroscale (WASMOD-M) model and the other five global models and found that high volume error exists in all the models. Beck et al. (2016) compared HBV-SIMREG model and other nine macroscale models and found that the daily runoff Nash-Sutcliffe Efficiency (NSE) values range from -1.67 (PCR-GLOBWB) to -0.02 (HBV-SIMREG). Zhang et al. (2016) evaluated the monthly and annual runoff estimates from 14 macro-scale models in 644 Australian catchments and found that for most models negative median NSE scores were obtained. Similar results obtained by Beck et al. (2017), who compared 10 state-of-the-art hydrological models over 966 catchments around the globe and concluded more effort should be devoted to calibrating and regionalizing the parameters of macro-scale models.*

*The majority of GHMs applied at the continental to global scale tend to rely on a priori parameterizations based on expert opinion, hydrologic theory, field data, case studies, or data sets of questionable quality, which may result in insufficient streamflow simulations (Beck et al., 2016). For example, the parameter values of the WBM-WTM are tuned by an adjustment factor, rather than calibration (Vörösmarty et al., 1989). The parameter values of WGHM were globally uniform or related to land cover and its associate properties, except the runoff coefficient which was tuned against time series of measured annual*

*discharges. In addition, the adjusted calibration factor is regionalized to grid cells outside the calibration basins (Döll et al., 2003). The parameter values of the Macro Probability Distribution Model (Macro-PDM) are set based on literature review or previous model applications and 6 out of 13 parameters are globally uniform (Arnell, 1999, 2003). The base flow recession constant (k) value of the PCRGLOBWB was determined based on the drainage theory and hydrogeologic data, however, many studies have found that there is a weak link between k and current hydrogeologic data sets (van Beek and Bierkens 2008; Peña-Arancibia et al., 2010; Beck et al., 2013).*

*Considering the restriction of the prior parameterizations, some GHMs have been developed based on model parameter regionalization. For example, the Water and Snow Balance Modeling System Macroscale (WASMOD-M) transferred the calibrated parameter sets to grid cells by searching for the most commonly occurring parameter set within a rectangular window and found that regionalized parameters produced better streamflow estimates than spatially uniform parameters (Widén-Nilsson et al., 2007). Beck et al. (2016) transferred the calibrated parameter sets of Hydrologiska Byråns Vattenbalansavdelning (HBV) model from the selected donor catchments to 0.5 °grid cells with the most similar climatic and physiographic characteristics and found that HBV with regionalized parameters outperformed nine state-of-the-art macroscale models. More recently, Beck et al. (2020) produced global parameter maps for the HBV hydrological model covering the entire land surface including ungauged regions by using transfer equations which link model parameters to climate and landscape characteristics and found that the median Kling-Gupta efficiency (KGE) values over 4229 independent validation catchments are comparable to those from previous large catchment sample studies.*

*Additionally, some studies have focused on the regionalization of macro-scale hydrologic models and illustrated the effectiveness of regionalization methods in macro-scale runoff simulation. For example, Troy et al. (2006) interpolated the model parameters of calibrated grid cells to the uncalibrated grid cells across the continental United States and found that this approach was useful for large-scale applications. Livneh and Lettenmaier (2013) tested the regression model which links "zonally representative" parameters to catchment descriptors across the continental United States and found that this approach resulted in improved model performance. Li and Zhang (2017) evaluated two grid-based regionalization approaches (gridded spatial proximity and gridded integrated similarity), and their lumped counterparts over 605 unregulated catchments across Australia and found a marginal difference between the gridded and lumped regionalization approaches. They concluded that rainfall-runoff modeling together with the gridded regionalization approach could be used for macro-scale runoff prediction.*

*Various regionalization methods were proposed in the past few decades (Abdulla and Lettenmaier, 1997; Hundecha and Bárdossy, 2004; Pokhrel et al., 2008; Jin et al., 2009; Samaniego et al., 2010; He et al., 2011; Razavi and Coulibaly, 2013). The widely used regionalization methods in literature can be categorized into the following categories: regression-based approaches (RE) (Xu, 1999, Yang et al., 2018), distance/attributes-based approaches (spatial proximity and physical similarity)(Yang et al., 2020), global mean method (GM) (Jin et al., 2009; He et al., 2011; Razavi and Coulibaly, 2013), scaling*

*relationship approach (Croke and Norton, 2004; Schreider et al., 2002), and simultaneous regionalization method (Hundecha and Bárdossy, 2004; Samaniego et al., 2010; Beck et al., 2020). However, limitations still exist in these methods (Bárdossy, 2007). For example, the regression method and distance/attributes-based approaches are confounded by the equifinality problem of hydrological model parameters (Götzinger and Bárdossy, 2007), and the scaling relationships approach neglects the spatial heterogeneity of catchment characteristics. In addition, the selection of the transfer function and catchment characteristics used in the simultaneous regionalization method still needs to be discussed (Croke and Norton, 2004; Samaniego et al., 2010; Beck et al., 2020). Numerous studies have been made to compare regionalization approaches in different regions (e.g., Oudin et al., 2008; Li et al., 2017; Yang et al., 2019, 2020). However, there is still no clear conclusion on the best-performed regionalization method. In addition, there is a lack of comprehensive evaluation and comparison of regionalization methods at the global scale. Moreover, most of these studies only established one global hydrological model and the uncertainty of GHMs caused by different model structures and concepts is not taken into account by using various hydrological models.*

*Therefore, to complement existing global hydrological models and provide valuable spatial and temporal estimates of global water resources, a framework for building GHMs is proposed by combining the optimal global scale regionalization scheme and different gridded hydrological models. Specifically, the objectives of this study are to:*
*(1) evaluate five most widely-used regionalization methods by using four conceptual hydrological models over 2277 medium-sized catchments around the world to identify the optimal global scale regionalization scheme for parametrizing GHMs;*
*(2) simulate global water resources based on the GHMs established by the proposed framework.*

References:
Abdulla, F. A., and Lettenmaier, D. P.: Development of regional parameter estimation equations for a macroscale hydrologic model, J Hydrol, 197, 230-257, 1997.
Arnell, N. W.: A simple water balance model for the simulation of streamflow over a large geographic domain, J Hydrol, 217, 314-335, 1999.
Arnell, N. W.: Effects of IPCC SRES* emissions scenarios on river runoff: a global perspective, 2003.
Bárdossy, A.: Calibration of hydrological model parameters for ungauged catchments, Hydrol Earth Syst Sc, 11, 703-710, 10.5194/hess-11-703-2007, 2007.
Beck, H. E., Pan, M., Lin, P., Seibert, J., Dijk, A. I. J. M., and Wood, E. F.: Global Fully Distributed Parameter Regionalization Based on Observed Streamflow From 4,229 Headwater Catchments, J Geophys Res (Atmospheres), 125, 10.1029/2019jd031485, 2020.Bierkens, M. F. P.: Global hydrology 2015: State, trends, and directions, Water Resour Res, 51, 4923-4947, 10.1002/2015wr017173, 2015.
Beck, H. E., van Dijk, A. I. J. M., de Roo, A., Dutra, E., Fink, G., Orth, R., and Schellekens, J.: Global evaluation of runoff from 10 state-of-the-art hydrological models, Hydrol Earth Syst Sc, 21, 2881-2903, 10.5194/hess-21-2881-2017, 2017.
Beck, H. E., van Dijk, A. I. J. M., de Roo, A., Miralles, D. G., McVicar, T. R., Schellekens, J., and Bruijnzeel, L. A.: Global-scale regionalization of hydrologic model parameters,

Water Resour Res, 52, 3599-3622, 10.1002/2015wr018247, 2016.

Beck, H. E., van Dijk, A. I. J. M., Miralles, D. G., de Jeu, R. A. M., Sampurno Bruijnzeel, L. A., McVicar, T. R., and Schellekens, J.: Global patterns in base flow index and recession based on streamflow observations from 3394 catchments, Water Resour Res, 49, 7843-7863, 10.1002/2013wr013918, 2013.

Croke, B., and Norton, J.: Regionalisation of rainfall-runoff models, 2004.Döll, P., Kaspar, F., and Lehner, B.: A global hydrological model for deriving water availability indicators: model tuning and validation, J Hydrol, 270, 105-134, 2003.

Döll, P., Douville, H., Güntner, A., Müller Schmied, H., and Wada, Y.: Modelling Freshwater Resources at the Global Scale: Challenges and Prospects, Surv Geophys, 37, 195-221, 10.1007/s10712-015-9343-1, 2015.

Götzinger, J., and Bárdossy, A.: Comparison of four regionalisation methods for a distributed hydrological model, J Hydrol, 333, 374-384, 10.1016/j.jhydrol.2006.09.008, 2007.

Grill, G., Lehner, B., Thieme, M., Geenen, B., Tickner, D., Antonelli, F., Babu, S., Borrelli, P., Cheng, L., Crochetiere, H., Ehalt Macedo, H., Filgueiras, R., Goichot, M., Higgins, J., Hogan, Z., Lip, B., McClain, M. E., Meng, J., Mulligan, M., Nilsson, C., Olden, J. D., Opperman, J. J., Petry, P., Reidy Liermann, C., Saenz, L., Salinas-Rodriguez, S., Schelle, P., Schmitt, R. J. P., Snider, J., Tan, F., Tockner, K., Valdujo, P. H., van Soesbergen, A., and Zarfl, C.: Mapping the world's free-flowing rivers, Nature, 569, 215-221, 10.1038/s41586-019-1111-9, 2019.

Haddeland, I., Clark, D. B., Franssen, W., Ludwig, F., Voß, F., Arnell, N. W., Bertrand, N., Best, M., Folwell, S., Gerten, D., Gomes, S., Gosling, S. N., Hagemann, S., Hanasaki, N., Harding, R., Heinke, J., Kabat, P., Koirala, S., Oki, T., Polcher, J., Stacke, T., Viterbo, P., Weedon, G. P., and Yeh, P.: Multimodel Estimate of the Global Terrestrial Water Balance: Setup and First Results, J Hydrometeorol, 12, 869-884, 10.1175/2011jhm1324.1, 2011.Oki, T., and Kanae, S.: Global Hydrological Cycles and World Water Resources, 313, 1068-1072, 10.1126/science.1128845 %J Science, 2006.

He, Y., Bárdossy, A., and Zehe, E.: A review of regionalisation for continuous streamflow simulation, Hydrol Earth Syst Sc, 15, 3539-3553, 10.5194/hess-15-3539-2011, 2011.

Jin, X., Xu, C-Y., Zhang, Q., and Chen, Y. D.: Regionalization study of a conceptual hydrological model in Dongjiang basin, south China, Quatern Int, 208, 129-137, 2009.

Li, H., and Zhang, Y.: Regionalising rainfall-runoff modelling for predicting daily runoff: Comparing gridded spatial proximity and gridded integrated similarity approaches against their lumped counterparts, J Hydrol, 550, 279-293, 10.1016/j.jhydrol.2017.05.015, 2017.Liang, X., Lettenmaier, D. P., Wood, E. F., and Burges, S. J.: A simple hydrologically based model of land surface water and energy fluxes for general circulation models, J Geophys Res (Atmospheres), 99, 14415-14428, 1994.

Livneh, B., and Lettenmaier, D. P.: Regional parameter estimation for the unified land model, Water Resour Res, 49, 100-114, 2013.

Oudin, L., Kay, A., Andréassian, V., and Perrin, C.: Are seemingly physically similar catchments truly hydrologically similar? Water Resour Res, 46, 10.1029/2009wr008887, 2010.

Parajka, J., Blöschl, G., and Merz, R.: Regional calibration of catchment models: Potential

for ungauged catchments, Water Resour Res, 43, 10.1029/2006wr005271, 2007.

Peña-Arancibia, J., Van Dijk, A., Mulligan, M., and Bruijnzeel, L. A.: The role of climatic and terrain attributes in estimating baseflow recession in tropical catchments, Hydrol Earth Syst Sc, 14, 2193, 2010Samaniego, L., Kumar, R., and Attinger, S.: Multiscale parameter regionalization of a grid-based hydrologic model at the mesoscale, Water Resour Res, 46, 10.1029/2008wr007327, 2010.

Razavi, T., and Coulibaly, P.: Streamflow prediction in ungauged basins: review of regionalization methods, Journal of hydrologic engineering, 18, 958-975, 2013.

Samaniego, L., Kumar, R., and Attinger, S.: Multiscale parameter regionalization of a grid-based hydrologic model at the mesoscale, Water Resour Res, 46, 10.1029/2008wr007327, 2010.

Samaniego, L., Kumar, R., Thober, S., Rakovec, O., Zink, M., Wanders, N., Eisner, S., Müller Schmied, H., Sutanudjaja, E. H., Warrach-Sagi, K., and Attinger, S.: Toward seamless hydrologic predictions across spatial scales, Hydrol Earth Syst Sc, 21, 4323-4346, 10.5194/hess-21-4323-2017, 2017.

Schreider, S. Y., Jakeman, A. J., Gallant, J., and Merritt, W. S.: Prediction of monthly discharge in ungauged catchments under agricultural land use in the Upper Ping basin, northern Thailand, Math Comput Simulat, 59, 19-33, 2002.

Schreider, S. Y., Jakeman, A. J., Gallant, J., and Merritt, W. S.: Prediction of monthly discharge in ungauged catchments under agricultural land use in the Upper Ping basin, northern Thailand, Math Comput Simulat, 59, 19-33, 2002.

Sellers, P. J., Mintz, Y., Sud, Y. C., and Dalcher, A. J. J. o. A. S.: A Simple Biosphere Model (SiB) for Use Within General-Circulation Models, 43, 505-531, 1986.

Sood, A., and Smakhtin, V.: Global hydrological models: a review, Hydrolog Sci J, 60, 549-565, 10.1080/02626667.2014.950580, 2015.

Troy, T. J., Wood, E. F., and Sheffield, J.: An efficient calibration method for continental‐scale land surface modeling, Water Resour Res, 44, 2008.

Van Beek, L. P., Eikelboom, T., van Vliet, M. T., and Bierkens, M. F.: A physically based model of global freshwater surface temperature, Water Resour Res, 48, 2012.

Van Beek, L., and Bierkens, M.: The Global Hydrological Model PCR-GLOBWB: Conceptualization, Parameterization and Verification Department of Physical Geography, Utrecht University, Utrecht, the Netherlands, 2008.

Vörösmarty, C. J., Moore III, B., Grace, A. L., Gildea, M. P., Melillo, J. M., Peterson, B. J., Rastetter, E. B., and Steudler, P. A.: Continental scale models of water balance and fluvial transport: an application to South America, Global Biogeochem Cy, 3, 241-265, 1989.Widén-Nilsson, E., Halldin, S., and Xu, C.-y.: Global water-balance modelling with WASMOD-M: Parameter estimation and regionalisation, J Hydrol, 340, 105-118, 10.1016/j.jhydrol.2007.04.002, 2007.

Widén-Nilsson, E., Gong, L., Halldin, S., and Xu, C-Y.: Model performance and parameter behavior for varying time aggregations and evaluation criteria in the WASMOD-M global water balance model, Water Resour Res, 45, 10.1029/2007wr006695, 2009.

Xu, C.-Y.: Estimation of parameters of a conceptual water balance model for ungauged catchments, Water Resour Manag, 13, 353-368, 1999.

Yang, X., Magnusson, J., and Xu, C.-Y.: Transferability of regionalization methods under changing climate, J Hydrol, 568, 67-81, 10.1016/j.jhydrol.2018.10.030, 2019.

Yang, X., Magnusson, J., Huang, S., Beldring, S., and Xu, C.-Y.: Dependence of

regionalization methods on the complexity of hydrological models in multiple climatic regions, J Hydrol, 582, 10.1016/j.jhydrol.2019.124357, 2020.

Yang, X., Magnusson, J., Rizzi, J., and Xu, C.-Y.: Runoff prediction in ungauged catchments in Norway: comparison of regionalization approaches, Hydro Res, 49, 487-505, 10.2166/nh.2017.071, 2018.

Yang, Z. L., and Dickinson, R. E.: Description of the Biosphere-Atmosphere Transfer Scheme (BATS) for the Soil Moisture Workshop and evaluation of its performance, Global Planet Change, 13, 117-134, 1996.

Zhang, Y., Zheng, H., Chiew, F. H., Arancibia, J. P., and Zhou, X.: Evaluating regional and global hydrological models against streamflow and evapotranspiration measurements, J Hydrometeorol, 17, 995-1010, 2016.

A more clear novelty statement should be presented to make it clear how this study solves or advances current issues in global scale modeling and regionalization of model parameters. I miss also the discussion on the issue of model parameter discontinuity for regional and global hydrological modeling (e.g., Samaniego et al., 2017) and how the proposed framework can deal with it.

Reply: We agree with the reviewer that one evaluation criterion is not sufficient. In the revised manuscript, the other two metrics: NSE and accuracy of volume estimates (AVE) defined as one minus volume error will be added.

GHMs based on model parameter regionalization obtain global gridded parameter maps by transferring calibrated parameter sets from the selected donor catchments to grid cells and illustrate the effectiveness of regionalization methods in macro-scale runoff simulation (e.g. Widén-Nilsson et al., 2007, 2009; Beck et al., 2016). However, they simply used one specific scheme to calculate global parameter maps rather than comparing the widely used regionalization approaches at the global scale, failed to select the optimal scheme for the regionalization of GHMs. In addition, there is a lack of comprehensive evaluation and comparison of regionalization methods at the global scale. Therefore, this study compared the performance of widely used regionalization methods on global scale by using 4 catchment scale hydrological models and selected the optimal global scale regionalization scheme. The optimal global scale regionalization scheme is then used to generate parameter values for gridded global models. Compared with other previous GHMs, GHMs built by the proposed framework show reasonable efficiency in global water resource estimation.

Besides, the issue of model parameter discontinuity for regional and global hydrological modeling will be discussed in the revised manuscript as follows:

*Based on the best performed regionalization method (section 3.2), the GSRS used to calculate global hydrological parameters and establish GHM was selected. In other words, the gridded parameters of four GHMs at a 0.5 °spatial resolution were calculated by using distance/attributes-based approaches (SPI-OUT and PSDI-OUT) for all global land grids. This study indicates that GHMs built by the proposed framework show reasonable efficiency in global water resource estimation and are comparable to other previous GHMs in catchment streamflow simulation (Widén-Nilsson et al., 2007; Beck et al., 2016, 2020;*

*Arheimer et al., 2019).*

*In addition, the median KGE values of GHMs decrease compared to the results on catchment scale. According to the GSRS, the parameter sets calibrated for lumped catchment scale hydrological models can transfer to land grids around the globe. Therefore, different parameter sets over various grids in one catchment and the convergence from grid runoff to catchment streamflow may result in the deterioration of the performance.*

*There are some limitations for the techniques used in this study. Firstly, the parameter sets of global grids do not have a functional relationship with their physiographic characteristics (Samaniego et al., 2017). Some regionalization methods have been developed to solve the problem on catchment scale, e.g. the simultaneous regionalization method (Hundecha and Bárdossy, 2004; Wallner et al., 2013; Samaniego et al., 2010, 2017; Beck et al., 2020). However, the high degree of variability in meteorological and hydrological variables, as well as different simplifications of hydrological processes in hydrological models make it difficult to effectively select the catchment attributes on global scale which can properly represent the catchment similarity and the proper transfer functions of parameters (Mizukami et al., 2017). In addition, regionalization methods used in this study have been successfully applied all over the world and their effectivenesses have been confirmed on catchment scale. Therefore, under the limitation of various conditions on global scale, using the spatial proximity method is reasonable, as it performs the best in this study.*

*Secondly, the regionalization methods are confounded by parameter equifinality problem (Oudin et al., 2010), as all hydrological models are subjected to parameter equifinality. However, according to Figure R2, similar results were obtained between the randomly selected parameter set and the original regionalization method, which indicates that the influence of equifinality problem on regionalization performance is limited. A similar conclusion has also been drawn in previous studies (e.g. Arsenault and Brissette, 2014).*

*Thirdly, the parameter discontinuity exists between catchment-scale and grid-scale regionalization. Since the spatial resolution of GHMs built in this study is 0.5°, the spatial discontinuities (e.g., calibration imprints circumscribing river basin boundaries) may affect the model performance when using the GHMs to simulate the catchment streamflow (Mizukami et al., 2017; Samaniego et al., 2017). The multiscale parameter regionalization (MPR) technique proposed by Samaniego et al. (2010) which links the field scale (observations) with the catchment scale can provide seamless parameters for LSMs/HMs parameterizations. Even though the MPR technique can significantly improve the consistency of simulated evapotranspiration fields across scales, the efficiency in the simulated flow was only slightly improved (Samaniego et al., 2017). Especially, the difficulty in the selection of proper transfer functions and the high computational cost in performing the MPR regionalization method makes it difficult to use globally. Moreover, since a large number of hydrological models are available for catchment-scale applications, the main objective of this study is to build different GHMs to complement existing global water-balance models and provide valuable spatial and temporal estimates of global water resources, rather than improving catchment streamflow simulation*

*efficiency. Therefore, the parameter discontinuity between catchment-scale and grid-scale regionalization was not considered in this study.*

References:

Arheimer, B., Pimentel, R., Isberg, K., Crochemore, L., and Pineda, L.: Global catchment modelling using World-Wide HYPE (WWH), open data, and stepwise parameter estimation, Hydrol Earth Syst Sc, 24, 535-559, 2020.

Arsenault, R., and Brissette, F. P.: Continuous streamflow prediction in ungauged basins: The effects of equifinality and parameter set selection on uncertainty in regionalization approaches, Water Resour Res, 50, 6135-6153, 10.1002/2013wr014898, 2014.

Beck, H. E., Pan, M., Lin, P., Seibert, J., Dijk, A. I. J. M., and Wood, E. F.: Global Fully Distributed Parameter Regionalization Based on Observed Streamflow From 4,229 Headwater Catchments, J Geophys Res (Atmospheres), 125, 10.1029/2019jd031485, 2020.

Beck, H. E., van Dijk, A. I. J. M., de Roo, A., Miralles, D. G., McVicar, T. R., Schellekens, J., and Bruijnzeel, L. A.: Global-scale regionalization of hydrologic model parameters, Water Resour Res, 52, 3599-3622, 10.1002/2015wr018247, 2016.

Hundecha, Y., and Bárdossy, A.: Modeling of the effect of land use changes on the runoff generation of a river basin through parameter regionalization of a watershed model, J Hydrol, 292, 281-295, 2004.

Mizukami, N., Clark, M. P., Newman, A. J., Wood, A. W., Gutmann, E. D., Nijssen, B., Rakovec, O., and Samaniego, L.: Towards seamless large-domain parameter estimation for hydrologic models, Water Resour Res, 53, 8020-8040, 10.1002/2017wr020401, 2017.

Oudin, L., Kay, A., Andréassian, V., and Perrin, C.: Are seemingly physically similar catchments truly hydrologically similar? Water Resour Res, 46, 10.1029/2009wr008887, 2010.

Samaniego, L., Kumar, R., and Attinger, S.: Multiscale parameter regionalization of a grid-based hydrologic model at the mesoscale, Water Resour Res, 46, 10.1029/2008wr007327, 2010.

Samaniego, L., Kumar, R., Thober, S., Rakovec, O., Zink, M., Wanders, N., Eisner, S., Müller Schmied, H., Sutanudjaja, E. H., Warrach-Sagi, K., and Attinger, S.: Toward seamless hydrologic predictions across spatial scales, Hydrol Earth Syst Sc, 21, 4323-4346, 10.5194/hess-21-4323-2017, 2017.

Wallner, M., Haberlandt, U., and Dietrich, J.: A one-step similarity approach for the regionalization of hydrological model parameters based on Self-Organizing Maps, J Hydrol, 494, 59-71, 10.1016/j.jhydrol.2013.04.022, 2013.

Widén-Nilsson, E., Gong, L., Halldin, S., and Xu, C-Y.: Model performance and parameter behavior for varying time aggregations and evaluation criteria in the WASMOD-M global water balance model, Water Resour Res, 45, 10.1029/2007wr006695, 2009.

2. The description and the components of the proposed framework are not clear. It is not clear from the manuscript if the framework is actually a combination of all steps (i.e., different regionalization methods + different catchment scale conceptual models + interpolation to grid cells) or if it is just the selection procedure for regionalization methods.

It is not clear which features does make it a framework and not a simple sequence of methodological steps. Moreover, the workflow itself has to be clarified too. Specifically, it is not clear if the grids that correspond to donor catchments were preserved or regionalized as well. It is not clear which portion of catchments was left for evaluation of global regionalization. Please, also see my specific comments regarding these issued below.

Reply: Thanks for the comments and sorry for the lack of clarity in this part of the manuscript. The framework involved the following steps: (1) calibrate catchment hydrological models to obtain parameter information for regionalization, (2) compare different regionalization methods at the catchment scale and select the optimal global Scale Regionalization Scheme (GSRS), (3) parametrize gridded version of hydrological models using GSRS and build GHMs, and (4) calculate catchment streamflow using the runoff routing method. According to the proposed framework, the parameter maps were built using GSRS and the runoff time series of $0.5°$ grid cells all over the world except for Antarctica and the Arctic region were calculated. Then, the Network-response Routing Function (NRF) was selected to converge the grid runoff to catchment streamflow. Therefore, the framework provides a possibility for building GHMs with any hydrological model. In addition, even though this study uses specific regionalization schemes and routing methods, the other methods can be used in the framework for the further improvement of the performance of GHMs.

In addition, we evaluated the performance of the GHMs around the 2277 catchments and compared the performance of GHMs with the catchment regionalization results and calibration results, since all of the grids were regionalized.

These responses will be added to the revised manuscript.

3. Regionalization methods examined here refer to so called two step similarity approaches (Wallner et al., 2013) where donor catchments are independently calibrated and then their parameters are regionalized based on various similarity metrics (e.g., spatial proximity or physical similarity or both). Such approaches were reported to suffer from equifinality problem (Bárdossy, 2007; Götzinger and Bárdossy, 2007). Alternative one step approaches (e.g., Hundecha and Bárdossy, 2004; Samaniego et al., 2010; Mizukami et al., 2017) that were developed to tackle this problem are not discussed in this study. Only a very simplistic uncertainty analysis was performed in this study, but nevertheless exposed large equifinality problem for all 4 models. It was not shown how the detected parameter uncertainty has affected regionalization performance and has propagated to the global water balance estimates. Little discussion was provided on the reliability of final global water balance estimates.

Reply: Thanks for the referee's suggestion. The one step regionalization approach will be discussed in the introduction of the revised manuscript. For more details, please see the response to the general comment 1.

In addition, following reviewer's suggestion, the investigation of equifinality problem of regionalization methods has been done and will be added in the revised manuscript. As

described in the response to general comment 1, in the investigation, we randomly choose one of the ten calibrated parameters in the catchment scale regionalization process under the threshold of 0.5 over 2277 catchments and compare the regionalization performance with the original method (please see in the Figure R2). The results show that the regionalization performance of the randomly selected parameters is consistent with the performance of the original regionalization method (Figures 9-10), which means that equifinality does not contribute significantly to the overall uncertainty witnessed throughout the regionalization methods applications. Similar conclusions were also drawn in Arsenault and Brissette (2014).

As for the performance of global water balance estimates, the lack of the exact data on global water resources makes it difficult to illustrate the effectiveness of the global and continental water resources calculated by GHMs. Therefore, to demonstrate the performance of GHMs from various aspects, we first compared the mean monthly hydrograph of observed and simulated streamflows from 6 large catchments (>40,000 km$^2$) and found that GHMs can effectively capture the seasonality of catchment streamflow. Then, the NRF method was selected to converge the grid runoff to catchment streamflow and the KGE between simulated and observed streamflow was calculated to quantify the performance of streamflow simulations. Finally, we evaluated the performance of GHMs in simulating global long term average water resources by comparing the other GHMs. The results show that the performance of the GHMs built by the proposed framework is comparable to other previous GHMs.

These points will be added to the revised manuscript.

4. Several crucial assumptions were neither tested not critically discussed (e.g., independency of catchment descriptors used in this study; link between physical and hydrological similarity). Moreover, several methodological details are missing or ignored (e.g., catchments affected by human activities are not rigorously filtered out; no indication on spatial discretization used in the models; no indication if the distance between catchments was calculated based on outlets or centroids; no clear distinction between donor and evaluation catchments), making it difficult to the judge the credibility of this study. Please see my specific comments for details.

Reply: Thanks for the comments and suggestions. We will provide more clear clarification in the revision that the catchment descriptors used in this study are primarily based on the paper of He et al. (2011) and Razavi and Coulibaly (2013), which counted the number of times that catchment descriptors were used after reviewing the regionalization methods. Based on this information, thirteen catchment descriptors were selected and used in regionalization methods in this study (Table 1).

The inadequate consideration of the influence caused by regulation is one of the major limitations in the previous global modeling studies, as well as in this study. This is because explicitly considering the impact caused by regulated catchments is a challenge in global hydrological modeling due to the lack of global information to an adequate accuracy. The interference of reservoir storage, regulation, and artificial utilization of water resources

may affect the performance of nature catchment streamflow simulation (Lehner et al., 2011). However, the impact may be negligible in simulating the long-term average annual runoff and global and continental water resources. Following reviewer's advice, in the revised manuscript, we will add the discussion of the influence of the regulated catchments and provide perspective for future studies.

As for the information about the spatial discretization used in the models, we will also better clarify that lumped models were calibrated and used for the comparison of regionalization methods and the selection of the global scale regionalization scheme. And for the global hydrological model building, gridded versions (0.5°) of these four models were used. Similar procedure was used in other global hydrological modeling studies (e.g. *Widén-Nilsson et al., 2007, 2009; Beck et al., 2016).*

The catchment position used in this study represents the geographical position of the catchment outlet. The distance between catchments was calculated based on catchment outlets.

In addition, we evaluated the performance of the GHMs around the 2277 catchments and compared the performance of GHMs with the catchment regionalization results and calibration results, since all of the grids were regionalized.

For more details, please see the answer to the specific comments of Line 101-102, Line 111-119, Section 2.5 and Line 359-361.

References:

Beck, H. E., van Dijk, A. I. J. M., de Roo, A., Miralles, D. G., McVicar, T. R., Schellekens, J., and Bruijnzeel, L. A.: Global-scale regionalization of hydrologic model parameters, Water Resour Res, 52, 3599-3622, 10.1002/2015wr018247, 2016.

He, Y., Bárdossy, A., and Zehe, E.: A review of regionalization for continuous streamflow simulation, Hydrol Earth Syst Sc, 15, 3539-3553, 10.5194/hess-15-3539-2011, 2011.

Lehner, B., Liermann, C. R., Revenga, C., Vörösmarty, C., Fekete, B., Crouzet, P., Döll, P., Endejan, M., Frenken, K., Magome, J., Nilsson, C., Robertson, J. C., Rödel, R., Sindorf, N., and Wisser, D.: High‐resolution mapping of the world's reservoirs and dams for sustainable river‐flow management, Frontiers in Ecology and the Environment, 9, 494-502, 10.1890/100125, 2011.

Widén-Nilsson, E., Halldin, S., and Xu, C.-Y.: Global water-balance modelling with WASMOD-M: Parameter estimation and regionalisation, J Hydrol, 340, 105-118, 10.1016/j.jhydrol.2007.04.002, 2007.

Widén-Nilsson, E., Gong, L., Halldin, S., and Xu, C. –Y.: Model performance and parameter behavior for varying time aggregations and evaluation criteria in the WASMOD-M global water balance model, Water Resour Res, 45, 10.1029/2007wr006695, 2009.

5. Writing and structure of the paper needs to be improved. Moreover, please check consistency of tense and plurals/singulars of nouns and verbs.

Reply: Thanks for the referee's suggestion. We will thoroughly check the grammar errors in the manuscript.

Specific comments
Line 36-43: According to this sentence in Line 36-38 land surface scheme models are the most commonly used global hydrological models. In the next two sentences it is stated that due to large biases of these models global hydrological models are developed. These statements are confusing, please clarify.

Reply: Sorry for the confusion. This sentence will be changed to "The LSM is commonly used as a component of climate models in simulating the energy and water balance at soil, atmosphere, and vegetation interfaces (Haddeland et al., 2011; Bierkens, 2015). However, global climate models are considerably biased in global runoff simulations (Yang and Dickinson, 1996; Sellers et al., 1986; Sood and Smakhtin, 2015)." in the revised manuscript.

References:

Bierkens, M. F. P.: Global hydrology 2015: State, trends, and directions, Water Resour Res, 51, 4923-4947, 10.1002/2015wr017173, 2015.
Haddeland, I., Clark, D. B., Franssen, W., Ludwig, F., Voß, F., Arnell, N. W., Bertrand, N., Best, M., Folwell, S., Gerten, D., Gomes, S., Gosling, S. N., Hagemann, S., Hanasaki, N., Harding, R., Heinke, J., Kabat, P., Koirala, S., Oki, T., Polcher, J., Stacke, T., Viterbo, P., Weedon, G. P., and Yeh, P.: Multimodel Estimate of the Global Terrestrial Water Balance: Setup and First Results, J Hydrometeorol, 12, 869-884, 10.1175/2011jhm1324.1, 2011.
Sellers, P. J., Mintz, Y., Sud, Y. C., and Dalcher, A. J. o. A. S.: A Simple Biosphere Model (SiB) for Use Within General-Circulation Models, 43, 505-531, 1986.
Sood, A., and Smakhtin, V.: Global hydrological models: a review, Hydrolog Sci J, 60, 549-565, 10.1080/02626667.2014.950580, 2015.
Yang, Z. L., and Dickinson, R. E.: Description of the Biosphere-Atmosphere Transfer Scheme (BATS) for the Soil Moisture Workshop and evaluation of its performance, Global Planet Change, 13, 117-134, 1996.

Line 44-60: I miss here an overview of the approaches that tune model parameters of global hydrological models (e.g., WaterGAP by Döll et al., 2003).

Reply: Thanks. We have clarified that the parameter values of WGHM (Döll et al., 2003) were globally uniform or related to land cover and its associate properties, except the runoff coefficient which was tuned against time series of the measured annual river discharges. In addition, the adjusted calibration factor was regionalized to grid cells outside the calibration basins (Döll et al., 2003). This information will be added in the revised manuscript.

Line 71: I do not think that regionalization technique proposed in Samaniego et al. 2010 (referred here is Luis et al., 2010) can be attributed to clustering or hydrological classification.

Reply: Sorry for the confusion. The regionalization method proposed in Samaniego et al. (2010) can be classified as a simultaneous regionalization method, in which parameter regionalization is carried out through simultaneous calibration of transfer-function parameters by assuming prior relationships between basin predictors. This will be modified in the revised manuscript.

Line 74: I would welcome here more insights on why despite numerous studies on comparison of regionalization approaches it is still not clear which of them perform better and how this study contributes to identification an appropriate method.

Reply: We will add a discussion on this statement that despite numerous studies that have been conducted to compare the performance of regionalization approaches, no consistent conclusion has been drawn. This was mainly due to the following reasons: (1) different hydrological models with various structures and concepts used in different research may affect the performance of the regionalization approaches; (2) there are high diversity and heterogeneity in different study regions; and (3) the subjective choices made by the authors, such as the number of donor catchments and the use of poorly modeled catchments as donors (Oudin et al., 2008).

References:
Oudin, L., Andréassian, V., Perrin, C., Michel, C., and Le Moine, N.: Spatial proximity, physical similarity, regression and ungaged catchments: A comparison of regionalization approaches based on 913 French catchments, Water Resour Res, 44, 10.1029/2007wr006240, 2008.

Line 76-77: Accounting for uncertainties in global hydrological modeling is an important issue that indeed was not tackled sufficiently in the previous studies. However, it is not clear how this study accounts for uncertainties? From what I have seen later in the paper, the authors indicated presence of considerable parameter uncertainty in each model and showed that global runoff estimates vary considerable among tested models. But does it really mean that the uncertainty is accounted for in this study?

Reply: We agree with the reviewer that the uncertainty is insufficiently accounted in this study as well. We tried to emphasize that different hydrological models with different structures and concepts may cause a large difference in global water resources modeling. To solve this problem, four different and widely used hydrological models were used and compared. The results show that the performance of hydrological models at the catchment scale was consistent with each other. However, the difference caused by different hydrological model structures cannot be ignored in the global water resources estimation, since these GHMs were built by the same framework and driven by the same meteorological data. This will be added in the discussion of the revised manuscript.

Line 96: Please specify the spatial resolution of the precipitation product.

Reply: The spatial resolution of precipitation data is 0.5 ° as the other meteorological data. This information will be added in the revised manuscript.

Line 101-102 and Table 1: I urge the authors to carefully examine the assumption about independency among catchment descriptors. I am afraid some of them have strong dependencies (e.g., slope and elevation or aridity index and mean annual potential evaporation). Instead of rather uninformative Table 1 that presents mean, max and min values of catchments descriptors globally (clearly a wide variety is to be expected here due to the global focus of the study), a correlation matrix would instead confirm or oppose the assumption of independency. The second assumption on the existence of well-behaved relationship between catchment descriptors and model parameters is certainly more difficult to prove, but at least a note on possible issues with this assumption has to be stated (see e.g., Odin et al., 2010 or Merz et al., 2020).

Reply: Thanks for the referee's suggestion. The catchment descriptors used in this study are primarily based on the paper of He et al. (2011) and Razavi and Coulibaly (2013). In addition, "soil clay content" and "water holding capacity" are also used, since the significant influence of these descriptors in the snow melting process was reported previously. We calculated the correlation coefficients between catchment descriptors to investigate the interdependencies of the selected catchment descriptors (please see below in the Figure R3). The results show the largest correlation coefficient between two catchment descriptors came from topsoil clay fraction and subsoil clay fraction (0.87), followed by water holding capacity and mean slope (-0.46). The topsoil clay fraction and subsoil clay fraction represent the percentage of clay in the topsoil (0–30 cm) and subsoil (30-100 cm), respectively. Both of them can significantly affect the snow melting process and soil infiltration capability and cannot be removed (Garambois et al., 2015, Mamedov et al., 2001; Räisänen et al., 2015).

[Figure]

Figure R3. Correlation coefficients between catchment descriptors

All the above will be added in the revised manuscript.

References:

He, Y., Bárdossy, A., and Zehe, E.: A review of regionalisation for continuous streamflow simulation, Hydrol Earth Syst Sc, 15, 3539-3553, 10.5194/hess-15-3539-2011, 2011.
Garambois, P. A., Roux, H., Larnier, K., Labat, D., and Dartus, D.: Parameter

regionalization for a process-oriented distributed model dedicated to flash floods, J Hydrol, 525, 383-399, 10.1016/j.jhydrol.2015.03.052, 2015.

Mamedov, A., Shainberg, I., and Levy, G.: Irrigation with effluents: effects of prewetting rate and clay content on runoff and soil loss, J Environ Qual, 30, 2149-2156, 2001.

Räsänen, P., Kokhanovsky, A., Guyot, G., Jourdan, O., and Nousiainen, T.: Parameterization of single-scattering properties of snow, 2015.

Line 111-119: Indeed, presence of regulated catchments creates a considerable obstacle for hydrological modeling. Does the selection based only on catchment area is able to filter out the regulated catchments? I suggest authors to include additional criteria that are customary in large-scale hydrological modeling (e.g., remove catchments with large dams, Lehner et al., 2011, Grill et al., 2019; test the closure of water balance, Beck et al., 2016).

Reply: Yes, it is not sufficient to effectively filter out the regulated catchments by only considering the size of the catchment. We will add a discussion that the inadequate consideration of the influence caused by regulation is one of the limitations in this study. However, effectively considering the impact caused by regulated catchments is a challenge in global hydrological modeling. The effectiveness of other simple criteria for filtering out regulated catchments remains to be verified. In addition, the interference of reservoir storage, regulation, and artificial utilization of water resources may affect the performance of nature catchment streamflow simulation. However, the impact is negligible with the long-term average annual runoff and global and continental water resources estimation. Since the focus of this study is on average water resources, the influence of the regulated catchments was not considered. However, we will discuss this limitation in the revised manuscript.

Section 2.3: The rationale on selection of these four models is not clear to me. It is noted (Line 76-77) that by considering models of different structure and concepts one might account for the model uncertainty. However, from the description provided for each model in this section, it is not clear which differences in their concepts and structures apart from the number of parameters exist. Moreover, I miss here the indication if the models were applied in lumped or in a distributed fashion.

Reply: There are a plethora of hydrological models available. Considering the scope of the project, computationally-demanding models had to be excluded in the regionalization method comparison from the start. We chose 4 widely used lumped models for water resources management, and these 4 models have different complexity and different number of free parameters to explore this aspect of the project.

Considering the length limit of the paper, only a brief description of model structures and parameters was given in the original manuscript. The hydrological models used in this study are varied in the simplification of the natural hydrological processes, especially in the process of vertical water balance and horizontal transport. For example, for GR4J, the transformation of rain to flow is carried out utilizing two reservoirs and a routing production (Boumenni et al., 2017). SIMHYD contains three storages for interception loss, soil moisture and groundwater, and the river runoff is composed of surface flow, interflow

and baseflow (Chiew et al., 2002; Chiew 2010). In XAJ, the non-uniform vertical distribution of soil is taken into account, whereby the three-layer calculation algorithm is applied to compute total evapotranspiration (Chen et al., 2019). HMETS uses two connected reservoirs representing unsaturated (i.e. vadose) and saturated (i.e. phreatic) zones, and takes all the exchanges made between the surface, vadose and saturated zones for the vertical water balance into consideration (Martel et al., 2017; Shen et al., 2018).

For more detail, please see Figure R4.

[Figure]

Figure R4. The structure of the four hydrological models in this study.

All the information above will be added in the revised manuscript.

References:

Boumenni, H., Bachnou, A., and Alaa, N. E.: The rainfall-runoff model GR4J optimization of parameter by genetic algorithms and Gauss-Newton method: application for the watershed Ourika (High Atlas, Morocco), Arabian Journal of Geosciences, 10, 10.1007/s12517-017-3086-x, 2017.

Chen, Y., Shi, P., Qu, S., Ji, X., Zhao, L., Gou, J., and Mou, S.: Integrating XAJ Model with GIUH Based on Nash Model for Rainfall-Runoff Modelling, Water, 11, 10.3390/w11040772, 2019.

Chiew, F. H., Peel, M. C., and Western, A. W.: Application and testing of the simple rainfall-runoff model SIMHYD, Mathematical models of small watershed hydrology and applications, 335-367, 2002.

Chiew, F. H.: Lumped Conceptual Rainfall‐Runoff Models and Simple Water Balance Methods: Overview and Applications in Ungauged and Data Limited Regions, Geography Compass, 4, 206-225, 2010.

Martel, J.-L., Demeester, K., Brissette, F. P., Arsenault, R., and Poulin, A.: HMET: a simple and efficient hydrology model for teaching hydrological modeling, flow forecasting and climate change impacts, The International journal of engineering education, 33, 1307-1316, 2017.

Shen, M., Chen, J., Zhuan, M., Chen, H., Xu, C.-Y., and Xiong, L.: Estimating uncertainty

and its temporal variation related to global climate models in quantifying climate change impacts on hydrology, J Hydrol, 556, 10-24, 10.1016/j.jhydrol.2017.11.004, 2018.

Line 156-157: According to the catchment selection rules listed in Line 115-116 catchments with at least 5 years of observations were selected. This would mean around 3.5 years for calibration in these catchments. How many catchments have such short calibration period? Do you think such short calibration period might increase parameter uncertainty?

Reply: We counted the observed discharge lengths of 2277 catchments. The results show that only one catchment has five years of observations and no more than 10% of 2277 catchments (219) have less than ten years of observations. We acknowledge that such a short calibration period will have uncertainties, while considering 90% of the catchment has over 10 years data and only one catchment with five years observation, we believe that the uncertainty is limited which will not impact the main conclusions in this study. We will add the clarification in the revised manuscript.

Line 167-177: I would welcome here the rationale on selection of these regionalization methods. Most of the approaches used here are two step similarity approaches, meaning that in the first step the model parameters are identified at gauged locations independently from similarity or spatial proximity of the catchments. In the second step calibrated parameters are regionalized assuming that closer catchments or more physically similar catchments have similar model parameters. Such approaches are not able to account for equifinality of model parameters (Bardossy, 2007; Götzinger and Bardossy, 2007). Therefore, neighboring catchments and even physically similar catchments not necessarily obtain similar parameter sets during calibration (Oudin et al., 2010). The methods used in this study are likely to suffer from a similar problem. This issue has to be addressed here and put into perspective of one step similarity approaches that were specifically developed to target this issue (e.g., Samaniego et al., 2010; Wallner et al., 2013; Mizukami et al., 2017).

Reply: Thanks for the referee's suggestion. We will add the discussion of the limitations of the regionalization methods used in this study and the impact of parameter equifinality on the performance of regionalization. More detailed responses can be found in the response to the general comments and Figure R2. In addition, the one step similarity approach will be introduced and discussed in the revised manuscript. More details can be found in the response to general comment 1.

Section 2.5: Several important details are missing in this section. Was the best performing parameter set regionalized or a mean parameters from 10 best sets mentioned in Lines 310-315? Was spatial proximity defined based on geographical position of the outlet or of the catchment centroid? Was the whole parameter set transferred or each single model parameter was regionalized independently? No details on regression method is provided. It should be specified what kind of regression model was used, which parameter estimation method was used and if all catchment descriptors were used to build the regression model.

It should be also reported if the regression model was built for each individual model parameter or for the whole set.

Reply: Sorry for the confusion in that section. Please find our responses as follows.
The parameter sets used in the regionalization is the best performing parameter set (with the largest median KGE values).
The spatial proximity is defined based on the geographical position of the outlet.
The whole parameter set was transferred to regionalization.
The multiple linear regression method was used in this study, in which the relationships among model parameters and the selected thirteen catchment descriptors were established using multiple linear regression and these functions were used to estimate model parameters for ungauged catchments.
The regression equation was built for each model parameter.

All these will be specified in the revised manuscript.

Line 182-184: How accurate is distance calculation on these projected coordinates globally? Why not to simply calculate geographical distances? Does "catchment position" refers to catchment outlet or catchment centroid?

Reply: First of all, we would like to apologize for the typos made in this part. In the program code and the formula used to calculate the distance between two catchment outlets is the Haversine formula (Abebe et al., 2019), rather than the Euclidean distance as mentioned in the original manuscript. Haversine formula determines the great-circle distance between two points on a sphere given their longitudes and latitudes. As the Earth is nearly spherical, the Haversine formula provides a good approximation of the distance between two points of the Earth surface, with a less than 0.3% error on average. The formula is given as follow:

$$D_{td} = 2 \times r \times \sin^{-1}\left(\sqrt{\sin^2\left(\frac{\phi_t - \phi_d}{2}\right) + \cos(\phi_t) \times \cos(\phi_d) \times \sin^2\left(\frac{\lambda_t - \lambda_d}{2}\right)}\right)$$

where r is the average radius of the Earth (i.e. 6378.137 km), t and d represent the target and donor catchments, respectively; $\phi_t$, $\phi_d$ and $\lambda_t$, $\lambda_d$ are catchment outlet latitude values and longitude values of the target and donor catchments (in radians).
This formula is proposed based on an assumption of a spherical earth. However, there is small bias in using a spherical model since the earth is not quite a sphere which typically below 0.3% (http://www.movable-type.co.uk/scripts/latlong.html).
The catchment position used in this study represents the geographical position of the catchment outlet.

All these above will be added in the revised manuscript.

References:

Abebe, M. A., Tekli, J., Getahun, F., Chbeir, R., and Tekli, G.: Generic metadata representation framework for social-based event detection, description, and linkage, Knowl-based Syst, 188, 10.1016/j.knosys.2019.06.025, 2020.

Line 187: I think Luis et al. 2010 and Samaniego et al. 2010 is the same study.

Reply: Thank you. This will be corrected in the revised manuscript.

Line 210-211: What does this sentence mean? Please clarify which catchments were used for the evaluation of regionalization techniques.

Reply: This sentence means that we treat each of the 2,277 catchments as the pseudo-ungauged catchments (no matter whether it is used as donor catchments) to evaluate the performance of regionalization schemes under each threshold. In addition, the threshold named 'all' means that all 2277 catchments can be used as donor catchments, whether poorly or well modeled. The reason why we set different threshold is that it is not clear in the literature whether it is more important to keep more donor catchments with the poorly calibrated result or to only choose well calibrated catchments. Therefore, we defined different thresholds of model efficiency (all, >0, >0.5, >0.6, >0.7, >0.8, >0.9) for the calibration period and tested the regionalization performance under different thresholds, to find the most appropriate donor catchments for global regionalization. This is considered as one of the innovation points of the study.

We evaluated the performance of GHMs over 2277 catchments and compared the performance of GHMs with the regionalization results and calibration results. This information will be added in the revised manuscript.

Line 213-218: It is not clear how exactly the parameters were regionalized to grid cells. Were the calibrated parameters of donor catchment preserved during regionalization? If they were preserved, were they assigned to catchment centroid or catchment outlet? How other grids within donor catchments were assigned?

Reply: Sorry for the confusion. All grids were regionalized in this study including grids corresponding to donor catchments. Only catchments with KGE value being greater than 0.5 at the calibration period were used as donor catchments.

The spatial proximity between catchments is defined based on the geographical position of the outlet. However, when calculating the distance between the grid and catchment, we use the center of the grid and the outlet of the catchment. For grids with a mean distance less than 1500 km to donors, the calibrated parameter sets of the 5 nearest donor catchments were transferred by using the SPI-OUT method. For grids with a mean distance larger than 1500 km, the parameters were derived from the PSDI-OUT method. All this information will be added.

Section 2.7: It is not clear what GSRS states for. Till Line 219 part I had an impression that GSRS is the proposed framework for the selection of an appropriate regionalization technique to transfer parameters of catchment scale conceptual models to the global grid cells. Therefore, the statement in this sentence comes surprising. Please clarify if GSRS is actually the proposed regionalization framework mentioned elsewhere in the manuscript. Please clarify its description and components.

Reply: Sorry for the confusion. Global scale regionalization scheme (GSRS) stands for the selection of an appropriate regionalization technique to transfer parameters from catchment scale to grid scale. The framework mentioned in the manuscript represents the combination of four main steps (Figure R1). The concept of the framework will be clarified in the revised manuscript at first time when it appears.

Line 221-225: It is not clear to me how the Network Response Routing converges grid cell runoff to catchment streamflow. Please clarify it and indicate what are the two parameters mentioned here. Moreover, please clarify how these two parameters exactly were calibrated. If these parameters were not regionalized how is it possible to obtain catchment streamflow at ungauged locations (i.e., grid cells)? Furthermore, I find merging of the conceptual model parameters with Network Response Routing parameters rather confusing. Please explain why this step was necessary.

Reply: Sorry for the unclear statement. The improved Network-response Routing Function (NRF) (Gong et al., 2009; Li et al., 2020) was used in this study as a flow routing method. The formula of this method is given as follow:

$$vi = v_{45} * (\tan c_i)^b$$

where $vi$ is the wave velocity of the grid, and $c_i$ is the slope of the grid. $v_{45}$ is the wave velocity in the grid with slope of 45°, and $b$ is a parameter that reflects how sensitive the wave velocity is to the slope.

The values of these parameters for calibration are [4,5,6,7,8,9,10] for $v_{45}$, and [0.2,0.3,0.4,0.5,0.6] for $b$, respectively. They were chosen based on computer capability limitations and the physical meaning of each parameter. Therefore, there are 35 routing parameter value sets, composed of five kinds of parameter $b$ and seven kinds of parameter $v_{45}$.

In this study, the enumeration method was used to calibrate these two routing parameters. Each routing parameter set was used to converge the grid runoff to catchment streamflow, and then the KGE was calculated based on the simulated streamflow and the observed streamflow for all catchments.

Since two routing parameters are calibrated rather than regionalized, it is hard to obtain streamflow time series for ungauged catchments. One way to obtain the streamflow for ungauged catchments is to simulate 35 sets of streamflows by using all 35 routing parameter value sets and then calculate their average (usually named as VBmean method). To evaluate the performance of this method, we calculated the KGE, NSE and AVE values of four GHMs in daily streamflow simulation over all 2277 catchment. Table R1 shows the median KGE, NSE and AVE values of 2277 catchments, which indicates that the performance of GHM VBmean is smaller than GHM. However, when it comes to AVE, the difference between these two methods becomes small. This result indicates that the VBmean method is proper to be used in the water balance analysis for ungauged catchments. However, effective estimation of streamflow time series and extreme flows in ungauged catchments still needs to be further studied. For example, using up-to-date

aggregation methods that can converge grid runoff to catchment or novel runoff routing method which do not need calibration to effectively simulate streamflow for ungauged catchments.

Table 4 in the original manuscript will be replaced by Table R1.

Table R1 The median values for the tested regionalization methods and GHMs over all catchments

|  | KGE | | | | |
|---|---|---|---|---|---|
|  | CAL | GM | SPI-OUT | GHM | GHM VBmean |
| GR4J | 0.748 | 0.141 | 0.542 | 0.378 | 0.308 |
| SIMHYD | 0.774 | 0.068 | 0.558 | 0.384 | 0.330 |
| XAJ | 0.766 | -0.093 | 0.546 | 0.382 | 0.328 |
| HMETS | 0.750 | 0.019 | 0.529 | 0.374 | 0.310 |
|  | NSE | | | | |
|  | CAL | GM | SPI-OUT | GHM | GHM VBmean |
| GR4J | 0.533 | -0.052 | 0.386 | 0.235 | 0.223 |
| SIMHYD | 0.566 | -0.249 | 0.400 | 0.258 | 0.246 |
| XAJ | 0.568 | -0.857 | 0.400 | 0.264 | 0.243 |
| HMETS | 0.528 | -0.420 | 0.380 | 0.284 | 0.249 |
|  | AVE | | | | |
|  | CAL | GM | SPI-OUT | GHM | GHM VBmean |
| GR4J | 0.977 | 0.633 | 0.842 | 0.682 | 0.678 |
| SIMHYD | 0.985 | 0.516 | 0.840 | 0.686 | 0.686 |
| XAJ | 0.979 | 0.500 | 0.843 | 0.673 | 0.667 |
| HMETS | 0.982 | 0.618 | 0.833 | 0.679 | 0.677 |

GHMs are developed to simulate (sub-) surface water fluxes and storages. However, it is difficult to judge the effectiveness of the global and continental water resources calculated by GHMs built, since exact data on global water resources are not available. Therefore, we selected the NRF method to converge the grid runoff to catchment streamflow to quantify the performance of the GHMs at the catchment scale. Previous studies have shown that NRF is an efficient routing method especially for large scale models (Gong et al., 2009)

All these will be added and discussed in the revised manuscript.

References:

Gong, L., Widén-Nilsson, E., Halldin, S., and Xu, C. Y.: Large-scale runoff routing with an aggregated network-response function, J Hydrol, 368, 237-250, 10.1016/j.jhydrol.2009.02.007, 2009.

Li, J., Zhao, H., Zhang, J., Chen, H., and Guo, S.: An improved routing algorithm for a large-scale distributed hydrological model with consideration of underlying surface impact, Hydro Res, 10.2166/nh.2020.170, 2020.

Line 226: Does this mean that Network Response Routing is the part of framework? So does it mean that the framework is actually GSRS+NRF?

Reply: Yes. Global scale regionalization scheme (GSRS) stands for the selection of an appropriate regionalization technique to transfer parameters from catchment scale to grid scale. The framework mentioned in the manuscript represents the combination of four main steps (Figure R1). The concept of the framework will be clarified in the revised manuscript at first time when it appears.

Line 241-242: Is precipitation gauge density is the only reason for good performance of hydrological models in these regions? Could these spatial variability in model performance result from inability of selected models to simulate discharge in drier areas compared to wetter areas?

Reply: Thanks for the questions. Besides of the poor precipitation gauge density, failure to use the most suitable model is another reason for the relatively poor performance of hydrological models in arid and semi-arid regions. Another fact is that most of the hydrological models show poorer simulation results in drier areas compared to wetter areas (Arheimer et al. 2020; Beck et al., 2020). It is difficult to estimate the streamflow in drier areas for the complicated streamflow generation processes. For example, both infiltration-excess and storage-excess runoff processes exist in most catchments in these areas (Beck et al., 2016; Ghebrehiwot and Kozlov, 2019). Therefore, it is of great significance to improve the performance of hydrological models and regionalization methods for ungauged basins in arid regions (Ghebrehiwot and Kozlov, 2019). This will be added in the revised manuscript.

References:
Arheimer, B., Pimentel, R., Isberg, K., Crochemore, L., and Pineda, L.: Global catchment modelling using World-Wide HYPE (WWH), open data, and stepwise parameter estimation, Hydrol Earth Syst Sc, 24, 535-559, 2020.
Beck, H. E., van Dijk, A. I. J. M., de Roo, A., Miralles, D. G., McVicar, T. R., Schellekens, J., and Bruijnzeel, L. A.: Global-scale regionalization of hydrologic model parameters, Water Resour Res, 52, 3599-3622, 10.1002/2015wr018247, 2016.
Beck, H. E., Pan, M., Lin, P., Seibert, J., Dijk, A. I. J. M. V., and Wood, E. F.: Global Fully Distributed Parameter Regionalization Based on Observed Streamflow From 4,229 Headwater Catchments, J Geophys Res (Atmospheres), 125, 2020.
Ghebrehiwot, A. A., and Kozlov, D. V.: Hydrological modeling for ungauged basins of arid and semi-arid regions: review, Vestnik MGSU, 1023-1036, 10.22227/1997-0935.2019.8.1023-1036, 2019.

Line 266-268: Where these results can be seen?

Reply: Figure 4 shows that the performance of regionalization is improved from one to multiple donor catchments and the optimal number is slightly different in regionalization methods and models. However, the optimal number is around five for the output averaging option. After reaching the optimal number, the differences between donor catchment

numbers become small. To further select the efficient number of the donor catchment in the global scale regionalization scheme, we calculated the differences of median KGE values between the best donor catchment numbers and five donors, which shows no more than 0.03 for each situation. Therefore, to balance the effect and the amount of computation, 5 donor catchments are suggested and selected to use for the output averaging method. This will be added and discussed in the revised manuscript.

Line 273-276: Why should poorly calibrated donors be considered in further regionalization? Is likely that by using the poorly performing parameter set additional uncertainty will be added to the regionalization. What was the criteria to choose the threshold value?

Reply: Poorly modeled catchments may yield higher uncertain model parameter values, but they may add some diversities for modeling ungauged catchments as well (Oudin et al., 2008). Therefore, it is worth evaluating whether it is necessary to consider poorly modeled catchments in regionalization. To balance the effectiveness and computation, we set a series of thresholds of "all", ">0" and a linear scale from 0.5 to 0.9 to choose the optimal threshold for the global scale regionalization scheme. This will be explained in the revised manuscript.

References:
Oudin, L., Andréassian, V., Perrin, C., Michel, C., and Le Moine, N.: Spatial proximity, physical similarity, regression and ungauged catchments: A comparison of regionalization approaches based on 913 French catchments, Water Resour Res, 44, 10.1029/2007wr006240, 2008.

Line 285-289 and Figure 5: Are the difference in the performance of different regionalization methods considerable? I see considerable differences among calibration, global mean and regression method compared to all other regionalization approaches. Apart from that the differences are minimal. Taking into account that it seems that only one best performing parameter set was used for regionalization, I am wondering if these differences would still be visible after accounting for the parametric uncertainties.

Reply: Indeed, the differences in performance are small between distance/attributes-based regionalization methods (spatial proximity, physical similarity, and physical similarity method considering distance). The same results have been shown in other regionalization studies on catchment scale (Arsenault and Brissette, 2014; Yang et al., 2020). According to previous studies (Oudin et al., 2008), it is not possible to decide which approach (spatial proximity, physical similarity) is the most appropriate one when the streaming network density is lower than 60 stations per 100,000 $km^2$. As we used four hydrological models, this result confirmed that the SPI-OUT method slightly performs better than the others.

To illustrate the influence of parameter equifinality on the regionalization, we randomly choose one of the ten calibrated parameters in the catchment scale regionalization process under the threshold of 0.5 over 2277 catchments. The regionalization performance of the randomly selected parameters is consistent with the results of the original regionalization

method (see Fig. R1). The differences are still visible for the randomly selected parameters. In the revised manuscript, Table 4 will be replaced by Table R1 (the red color shows the revised part in the Table R1), in which, the threshold of 0.5 is used instead of 0.7.

All above will be added in the revised manuscript.

References:
Arsenault, R., and Brissette, F. P.: Continuous streamflow prediction in ungauged basins: The effects of equifinality and parameter set selection on uncertainty in regionalization approaches, Water Resour Res, 50, 6135-6153, 10.1002/2013wr014898, 2014.
Oudin, L., Andréassian, V., Perrin, C., Michel, C., and Le Moine, N.: Spatial proximity, physical similarity, regression and ungaged catchments: A comparison of regionalization approaches based on 913 French catchments, Water Resour Res, 44, 10.1029/2007wr006240, 2008.
Yang, X., Magnusson, J., Huang, S., Beldring, S., and Xu, C.-Y.: Dependence of regionalization methods on the complexity of hydrological models in multiple climatic regions, J Hydrol, 582, 10.1016/j.jhydrol.2019.124357, 2020.

Section 3.2: I am missing here the discussion on such poor performance of the regression method. Could it be linked to the multicollinearity of catchment descriptors? Was the whole parameter set regionalized with this method or were regression models built for each model parameter individually? Was a global regression model built or was it region-specific?

Reply: Sorry for the lack of sufficient discussion on the performance of regression. Regression models were built for each model parameter individually, and they are globally consistent rather than region-specific. One of the reasons for their poor performance might be because the parameters of these complex models are not well correlated to the catchment characteristics (Arsenault and Brissette, 2014; Yang et al., 2018). As a result, most of their parameters are estimated with poor confidence. This information will be discussed in the revised manuscript.

References:
Arsenault, R., and Brissette, F. P.: Continuous streamflow prediction in ungauged basins: The effects of equifinality and parameter set selection on uncertainty in regionalization approaches, Water Resour Res, 50, 6135-6153, 10.1002/2013wr014898, 2014.
Yang, X., Magnusson, J., Rizzi, J., and Xu, C. Y.: Runoff prediction in ungauged catchments in Norway: comparison of regionalization approaches. Hydrology Research, 49(2), 487-505. doi:10.2166/nh.2017.071, 2018.

Figure 6: I am not sure that I see in this Figure that 1500 km is an optimal threshold. For GR4J 1300 km and for HMETS and XAJ 1200 km seems to be more appropriate. How exactly was threshold of 1500 km identified? Is this threshold sufficiently robust given that there are large fluctuations in the performance of the methods around this distance (e.g., for GR4J for 1400 km the methods based on both physical similarity and distance

outperform all other methods for all catchments, while for 1300 km and 1500 km spatial proximity is the best performing method). Why the x axis in the subplots differ? For some it reaches till $18*10^5$ m for others only till $15*10^5$ m. For all subplots the last bar seems to be cut. Please clarify what "Physical Similarity DIS" in the legend refers to?

Reply: We agree with the reviewer that only a few catchments have a mean distance large than 1500 km, which may result in some fluctuations in the performance of the methods. With the increased mean distance between donor and ungauged catchments, the advantage of the SPI-OUT method is reduced, and it is not an easy task to decide which method outperforms others. However, SPI-OUT generally performs better than or comparable to other regionalization methods in different distances. According to Beck et al. (2016, 2020), the regionalization approach provides slightly less (but still substantial) benefit in poorly gauged regions, and a significant decrease in KGE values can be seen when the mean distance is more than 5000 km. This phenomenon may be more pronounced for the spatial proximity method according to the foundational assumption of this method. Therefore, we took 1500 km as the threshold to decrease the influence of the increasing distance. It is noted that this choice of 1500 km is subject to change with the increase or decrease of density of the catchments.

The value of the x-axis represents the mean distance to the 5 nearest donors under the threshold of 0.5 for all 2277 catchments. As the threshold increases, the number of catchments available as donors reduces. For example, under the threshold of 0.5, there are 1902 (GR4J), 1985 (SIMHYD), 1947 (XAJ), and 1903 (HMETS) catchments that can be used as donor catchments. Therefore, the range of the mean distance to the 5 nearest donors over 2277 catchments is varying for different models.

Besides, "Physical Similarity DIS" means physical similarity method considering distance (PSD). We will add the clarification in the revised manuscript.

Figure 6 will be redrawn and all above will be added and discussed in the revised manuscript.

References:
Beck, H. E., van Dijk, A. I. J. M., de Roo, A., Miralles, D. G., McVicar, T. R., Schellekens, J., and Bruijnzeel, L. A.: Global-scale regionalization of hydrologic model parameters, Water Resour Res, 52, 3599-3622, 10.1002/2015wr018247, 2016.
Beck, H. E., Pan, M., Lin, P., Seibert, J., Dijk, A. I. J. M., and Wood, E. F.: Global Fully Distributed Parameter Regionalization Based on Observed Streamflow From 4,229 Headwater Catchments, J Geophys Res (Atmospheres), 125, 10.1029/2019jd031485, 2020.

Line 301-304: The difference in performance between the models is of the same magnitude as between the regionalization methods (apart from global mean and regression method). Therefore, I find the conclusion that model structure has an insignificant effect while the choice of regionalization method is important rather inconsequent.

Reply: It is difficult to decide which approach is the most appropriate since highly varying hydroclimatic characteristics exist all over the world. In our study, a comprehensive comparison of different regionalization methods in 2277 catchments globally indicates that SPI-OUT performs better than other methods in general. According to the cumulative density function (CDF) curves of different regionalization methods, the advantage of this approach is obvious (please see in the Figure R5), although the difference of median KGE among various methods is small.

Four different hydrological models were selected to evaluate whether the best performed method and the effectiveness of the proposed framework are model-independent. The models used in this study all perform reasonable results around the world and have been widely used in regionalization studies. According to the results, the influence of models on the performance of simulations is smaller than the regionalization methods. However, when it comes to the simulated continental or global runoffs, the difference among the four GHMs becomes larger.

All the above information will be added in the revised manuscript.

[Figure]

Figure R5. The cumulative density function (CDF) curves of different regionalization methods. "CAL" and "GHM" represent the results of calibration and global hydrological modeling, respectively. "GHM VBmean" represents the results of global hydrological modeling using the VBmean method (please note that the y-axis label starts from 0).

Line 310-320: This is a description of methods and should be moved to the methods Section.

Reply: Thanks. This will be corrected in the revised manuscript.

Line 311-312: Please clarify which criteria was used to select sufficiently different parameter sets.

Reply: Equifinality is defined as "multiple sets of parameters that lead to equally acceptable model performance during the model calibration and validation" (Beven and Freer, 2001; Samaniego et al., 2010). In this study, to consider the parameter equifinality of different hydrological models, all hydrological models were calibrated ten times with different

initial random seeds. In fact, the difference between ten parameter sets is small and has slight influence on regionalization performance. This will be added in the revised manuscript.

References:

Beven, K., and Freer, J.: Equifinality, data assimilation, and uncertainty estimation in mechanistic modeling of complex environmental systems using the GLUE methodology, J Hydrol, 249, 11-29, 2001.

Samaniego, L., Bárdossy, A., and Kumar, R.: Streamflow prediction in ungauged catchments using copula‐based dissimilarity measures, Water Resour Res, 46, 2010.

Line 328-329: This statement requires a citation. If this is a finding of this work consider reformulating this sentence accordingly.

Reply: Thanks for the suggestion. References will be added in the revised manuscript.

References:

Arsenault, R., and Brissette, F. P.: Continuous streamflow prediction in ungauged basins: The effects of equifinality and parameter set selection on uncertainty in regionalization approaches, Water Resour Res, 50, 6135-6153, 10.1002/2013wr014898, 2014.

Beck, H. E., Pan, M., Lin, P., Seibert, J., Dijk, A. I. J. M., and Wood, E. F.: Global Fully Distributed Parameter Regionalization Based on Observed Streamflow From 4,229 Headwater Catchments, J Geophys Res (Atmospheres), 125, 10.1029/2019jd031485, 2020.

Beven, K., and Freer, J.: Equifinality, data assimilation, and uncertainty estimation in mechanistic modelling of complex environmental systems using the GLUE methodology, J Hydrol, 249, 11-29, 2001.

Beven, K.: A manifesto for the equifinality thesis, J Hydrol, 320, 18-36, 10.1016/j.jhydrol.2005.07.007, 2006.

Kokkonen, T. S., Jakeman, A. J., Young, P. C., and Koivusalo, H. J.: Predicting daily flows in ungauged catchments: model regionalization from catchment descriptors at the Coweeta Hydrologic Laboratory, North Carolina, Hydrol Process, 17, 2219-2238, 10.1002/hyp.1329, 2003.

Xia, Y., Mitchell, K., Ek, M., Sheffield, J., Cosgrove, B., Wood, E., Luo, L., Alonge, C., Wei, H., and Meng, J.: Continental‐scale water and energy flux analysis and validation for the North American Land Data Assimilation System project phase 2 (NLDAS‐2): 1. Intercomparison and application of model products, J Geophys Res (Atmospheres), 117, 2012.

Line 329-330: One way to quantify the influence of parametric uncertainty for regionalization results is to perform regionalization using all equifinal parameter sets (i.e., 10 best performing parameter sets in this study) and analyze the differences in regionalization performance and global runoff estimates.

Reply: Thanks for the suggestion. Regionalization using all equifinal parameter sets is very

expensive in the computational cost in the study regarding a list of different regionalization methods together with 2277 catchments globally, although it is a good method to quantify the influence of parametric uncertainty. To balance the computation and the exploration of the influence of parameter equifinality on regionalization, we randomly choose one of the ten calibrated parameters in the catchment scale regionalization process under the threshold of 0.5 over 2277 catchments. The results show that the regionalization performance of the randomly selected parameter sets is consistent with the results of the original regionalization method (please see Fig. R1). This result indicates that the equifinality does not significantly affect the performance of regionalization methods. This will be added in the revised manuscript.

Line 348-352: How the deterioration in performance between the regionalization at catchment scale and regionalization at grid cell can be explained? If the same donor catchments were used and the best performing regionalization methods were selected why deterioration occurs at grid cells? Is this deterioration is caused solely by averaging over the grid cell?

Reply: The calibration and regionalization of hydrological models at the catchment scale were based on lumped models. However, for the building of global hydrological models, gridded versions of these four models were used. In addition, to evaluate the performance of GHMs at the catchment scale, we added the NRF method to converge the grid runoff to catchment streamflow. Therefore, the different parameter sets of different grid cells in one catchment may cause the deterioration of the hydrological model performance. In addition, the converge of the grid runoff to catchment streamflow using the NRF method and the low resolution may be the other reasons. Since the spatial resolution of GHMs built in this study is 0.5 °, the spatial discontinuities may affect the model performance when using the GHMs to simulate the catchment streamflow (Mizukami et al., 2017; Samaniego et al., 2017). This information will be added in the revised manuscript.

References:
Mizukami, N., Clark, M. P., Newman, A. J., Wood, A. W., Gutmann, E. D., Nijssen, B., Rakovec, O., and Samaniego, L.: Towards seamless large-domain parameter estimation for hydrologic models, Water Resour Res, 53, 8020-8040, 10.1002/2017wr020401, 2017.
Samaniego, L., Kumar, R., Thober, S., Rakovec, O., Zink, M., Wanders, N., Eisner, S., Müller Schmied, H., Sutanudjaja, E. H., Warrach-Sagi, K., and Attinger, S.: Toward seamless hydrologic predictions across spatial scales, Hydrol Earth Syst Sc, 21, 4323-4346, 10.5194/hess-21-4323-2017, 2017.

Line 352: Does "-G" stands for global version of respective hydrological model? Please introduce appropriately this new notation and explain how the global version is different from catchment version. Later (Line 354, 356, Table 4 and elsewhere) "-G" notation is not used anymore, although it seems like the global set up of the models is meant there as well.

Reply: Yes, the "-G" represents the global version of the respective hydrological model. According to the parameter maps built using GSRS, the runoff time series of 0.5 °grid cells

all over the world except for Antarctica and the Arctic region were calculated. Then, the NRF method was used to converge the grid runoff to catchment streamflow. Therefore, the global hydrological models (-G") were built using the gridded version of these four lumped catchment scale models.

Sorry for the unclear statement. In this section, to quantify the performance of GHMs, we calculated the difference of performance between the median KGE value of GHMs and that using the best performed regionalization method (i.e., SPI-OUT) of original lumped models, as well as the difference of performance between the median KGE value of GHMs and calibrated results of original lumped models. Therefore, we use "GR4J" rather than "GR4J-G" to represent the difference between these two versions. This will be specified in the revised manuscript.

Line 359-361: The values reported here and in Table 4 seems to be median KGE for all catchments (including donor catchments), while Beck et al. (2016) has reported KGE values for 2277 catchments only. Therefore, this comparison is not fair. Moreover, please consider reporting regionalization performance for evaluation catchments only. Without these results it is difficult to judge the credibility of the proposed framework.

Reply: Thanks for the suggestion. We agree with the reviewer this is not a fair comparison, since we calculated the median KGE values over 2277 catchments which include donor catchments. Therefore, to further illustrate the effectiveness of GHMs, we compared our results with some recent researches:

*In probably the most similar previous global regionalization study, Arheimer et al. (2020) produced global parameter maps for the HYPE hydrological model by using a stepwise approach for groups of parameters regulating specific processes and catchment characteristics in representative gauged catchments. They obtained a median monthly KGE of 0.40 for 2475 gauges used in parameter estimation and 0.39 for 2863 independent validation stations. Beck et al. (2020) produced parameter maps (0.05 resolution) for HBV using a regionalization approach that involves the optimization of transfer equations linking model parameters to climate and landscape characteristics. They used 4229 catchments and eight predictors to optimize the transfer equation and calculate the global parameter maps and they obtained a median daily KGE of 0.46 for these 4229 stations. Our median daily KGE values are comparable to Beck et al. (2020) and Arheimer et al. (2020).*

All these will be added in the revised manuscript.

References:

Arheimer, B., Pimentel, R., Isberg, K., Crochemore, L., and Pineda, L.: Global catchment modelling using World-Wide HYPE (WWH), open data, and stepwise parameter estimation, Hydrol Earth Syst Sc, 24, 535-559, 2020.
Beck, H. E., van Dijk, A. I. J. M., de Roo, A., Miralles, D. G., McVicar, T. R., Schellekens, J., and Bruijnzeel, L. A.: Global-scale regionalization of hydrologic model parameters, Water Resour Res, 52, 3599-3622, 10.1002/2015wr018247, 2016.

Beck, H. E., Pan, M., Lin, P., Seibert, J., Dijk, A. I. J. M. V., and Wood, E. F.: Global Fully Distributed Parameter Regionalization Based on Observed Streamflow From 4,229 Headwater Catchments, J Geophys Res (Atmospheres), 125, 2020.

Xia, Y., Mitchell, K., Ek, M., Sheffield, J., Cosgrove, B., Wood, E., Luo, L., Alonge, C., Wei, H., and Meng, J.: Continental‐scale water and energy flux analysis and validation for the North American Land Data Assimilation System project phase 2 (NLDAS‐2): 1. Intercomparison and application of model products, J Geophys Res (Atmospheres), 117, 2012.

Yamazaki, D., Ikeshima, D., Sosa, J., Bates, P. D., Allen, G., and Pavelsky, T.: MERIT Hydro: A high‐resolution global hydrography map based on latest topography datasets, Water Resour Res, 10.1029/2019wr024873, 2019.

Zink, M., Kumar, R., Cuntz, M., and Samaniego, L.: A high-resolution dataset of water fluxes and states for Germany accounting for parametric uncertainty, Hydrology and Earth System ences Discussions, 21, 1-29, 2016.

Line 364-366: In the later part of this paper the authors refer to the 4 catchment scale conceptual models regionalized to grid cells as global hydrological models (GHMs). The argumentation provided in the review of global hydrological models by Sood and Shmakhtin (2015) that global hydrological models were built for macroscale water resources management and not for streamflow simulation cannot be justified here, as Sood and Shmakhtin (2015) do not refer to regionalized catchment scale conceptual models but to actual global hydrological models that either are the components of general circulation models or stand-alone hydrological models with very few or no calibrated parameters that run directly at the global scale. Instead, the models derived here were specifically designed to improve streamflow simulation and calibrated to the observed streamflow.

Reply: Sorry for the unclear statement. The main objective of this study is to simulate global water resources based on the GHMs established by the proposed framework to complement existing global water-balance models and provide valuable spatial and temporal estimates of global water resources. However, the lack of exact data on global water resources makes it difficult to illustrate the effectiveness of the global and continental water resources calculated by GHMs. Therefore, we added the NRF method in our framework to converge the grid runoff to catchment streamflow and quantify the performance of the GHMs from the catchment scale. All the information above will be added in the revised manuscript.

Figure 9: Please clarify in the caption if the cumulative curves are showed for all catchments or evaluation catchments only. It seems that y axis is cut at 0, please indicate it in the caption. Clarify what CAL and GSRS stands for in the caption.

Reply: Thanks for the advice. The cumulative curves are shown for all 2277 catchments since the grids corresponding to the donor catchments were regionalized by the global scale regionalization scheme. The "CAL" represents the calibration results and the "GSRS" represents the GHMs" results. To make it clear, here we use the "GHM" to replace "GSRS". Figure 9 in the original manuscript will be replaced by Figure R5.

All the information above will be added in the revised manuscript.

Table 4: Please clarify what "framework" stands for? Is it an equivalent of "GSRS" from Figure 9?

Reply: Sorry for the confusion. The "framework" in table 4 and "GSRS" from Figure 9 both represent the results of GHMs. Therefore, to make it easy to understand, we unified using "GHM" to represent the results of the global hydrological model and "GHM VBmean" to represent the results of the global hydrological model using VBmethod (please see Table R1 and Figure R5).

Line 383-384: According to Table 5 SIMHYD-G performs the best. Please clarify the statement in this sentence.

Reply: Sorry for the unclear statement. The median KGE value of SIMHYD-G over 2277 catchments is bigger than XAJ-G. However, the median KGE values of XAJ-G in different climate regions are better than or equal to that of SIMHYD-G, especially in arid and polar regions. Therefore, we consider XAJ-G as the best performed model for its good and stable performance among different climate regions. In general, the performance differences of different GHMs at catchment scale are small, which means that the effectiveness of the proposed framework is model-independent. However, when it comes to the simulated continental or global runoffs, the difference among the four GHMs becomes larger. Therefore, the sentence "Overall, the XAJ-G performs the best and the HMET-G performs the worst." will be deleted and this part will be rewritten.

Line 389-390: Although I generally agree that the balance between model flexibility and complexity is very important, I would say that the results of this study rather show the opposite. Minor differences in KGE between the models (0.01 of KGE between the best and the worst one; this is exactly the threshold for performance difference that the authors selected to identify equifinal parameters earlier) indicates that the choice of the model played rather negligible role here. Moreover, earlier (Line 300-303) it was stated that model structure played insignificant role for regionalization performance in this study supporting rather the other conclusion of this work that potentially any model can be used in the proposed framework.

Reply: Thanks for the comments. In this section, we tried to emphasize that there is no need to specifically choose simple or complicated models for GHMs building by using regionalization. Therefore, the sentence "Overall, the XAJ-G performs the best and the HMET-G performs the worst." will be deleted and this part will be rewritten.

Line 422-424: A reference is needed for these estimates.

Reply: Thanks for the suggestion. The reference will be added in this part in the revised manuscript.

Line 426-430: This portion makes me wonder why the available global datasets on

anthropogenic influence (see my comment to Line 111-119) were not considered to filter out the affected catchments. Are there any prospects on obtaining reliable global precipitation datasets that can be used in the future studies?

Reply: According to the previous studies, the insufficient consideration of regulated catchments may affect the simulation results of catchment streamflow, especially for extreme events. However, the influence on the long-term average annual runoff and global and continental water resources estimation may be negligible. Therefore, we simply consider the area of the catchment to filter out the regulated catchments rather than consider global dam data sets.

However, inadequate consideration of the influence caused by regulation is one of the major limitations of this study. Therefore, further studies are still needed to take the influence caused by regulation into consideration to improve the performance of GHMs on the catchment streamflow simulation.

A lot of global gridded precipitation datasets were developed in recent years (Weedon et al., 2014; Schneider et al., 2015) and these datasets differ in terms of data sources (radar, gauge, satellite, analysis, or reanalysis, or combinations thereof), spatial resolution and temporal span. Many studies addressed the importance of precipitation datasets choice for applications (Beck et al., 2017). With the development of remote sensing technology, the satellite-based precipitation datasets, as well as the merging of multiple satellite and reanalysis precipitation datasets have great potential for simulation of global water resources. For example, the Multi-source weighted-ensemble precipitation (MSWEP) based on weighted averaging of several satellites, gauges, and reanalysis products are proved to show superior performance in numerous precipitation dataset evaluation studies (e.g., Beck et al., 2017, 2019), which can be used in future studies to evaluate the performance of the GHMs driven by different precipitation datasets.

Reference:

Beck, H. E., Vergopolan, N., Pan, M., Levizzani, V., Van Dijk, A. I., Weedon, G. P., Brocca, L., Pappenberger, F., Huffman, G. J., and Wood, E. F.: Global-scale evaluation of 22 precipitation datasets using gauge observations and hydrological modeling, Hydrol Earth Syst Sc, 21, 6201-6217, 2017.

Beck, H. E., Pan, M., Roy, T., Weedon, G. P., Pappenberger, F., Van Dijk, A. I., Huffman, G. J., Adler, R. F., and Wood, E. F.: Daily evaluation of 26 precipitation datasets using Stage-IV gauge-radar data for the CONUS, Hydrol Earth Syst Sc, 23, 207-224, 2019.

Schneider, U., Becker, A., Finger, P., Meyer-Christoffer, A., Ziese, M., and Rudolf, B.: GPCC"s new land surface precipitation climatology based on quality-controlled in situ data and its role in quantifying the global water cycle, Theor Appl Climatol, 115, 15-40, 2014.

Weedon, G. P., Balsamo, G., Bellouin, N., Gomes, S., Best, M. J., and Viterbo, P.: The WFDEI meteorological forcing data set: WATCH Forcing Data methodology applied to ERA‐Interim reanalysis data, Water Resour Res, 50, 7505-7514, 2014.

Line 462: Usage of "GHM" in this context is confusing, as in this study the GHM were

built from catchments-scale conceptual models by combining them with various regionalization approaches. If I understood correctly these steps combined are the framework proposed in this study. Consider using "conceptual model" instead.

Reply: Sorry for the unclear statement. The lumped models were used for the comparison of regionalization methods and the selection of the global scale regionalization scheme. For building the global hydrological model, gridded versions of these four models were used as global scale. This will be clarified in the revised manuscript.

Editorial comments

Abbreviations: The manuscript is oversaturated with abbreviations. Some of these abbreviations are never used after their introduction (e.g., LSS, PUB, AOF), some other abbreviations are only used once or twice (e.g., IDW, AM, CDF). Consider omitting using unnecessary abbreviations in the manuscript and especially in the abstract (e.g., NRF, IDW, KGE).

Reply: Thanks for the advice. We will revise carefully the whole manuscript and remove the unnecessary abbreviations in it.

Line 15: Why Network Response Routing is abbreviated as NRF. Should it be NRR instead?

Reply: Sorry for the confusion. The flow routing method used in this study is the improved Network-response Routing Function (NRF) (Gong et al., 2009; Li et al., 2020). The full name of the routing method will be corrected in the revised manuscript.

References:
Gong, L., Widén-Nilsson, E., Halldin, S., and Xu, C.: Large-scale runoff routing with an aggregated time-delay-histogram method, 2007.
Li, J., Zhao, H., Zhang, J., Chen, H., and Guo, S.: An improved routing algorithm for a large scale distributed the hydrological model with consideration of underlying surface impact, Hydro Res, 2020.

Line 54: a priori

Reply: Thanks. This will be corrected in the revised manuscript.

Line 67, 71 and elsewhere: Samaniego et al., 2010 instead of Luis et al., 2010

Reply: Thanks. This will be corrected in the revised manuscript.

Line 76: Moreover

Reply: Thanks. This will be corrected in the revised manuscript.

Line 85: delete "widely"

Reply: Thanks. This word will be deleted in the revised manuscript.

Line 383: "proposed" instead of "proposal"

Reply: Thanks. This will be corrected in the revised manuscript.

Line 384: I think you mean here that it was not confirmed in this study.

Reply: Sorry for the unclear statement. This sentence will be changed to "However, this is not confirmed in this study" in the revised manuscript.

Line 385: "that" instead of "who"

Reply: Thanks. This will be corrected in the revised manuscript.

Line 387: "is not reduced" instead of "does not reduce"

Reply: Thanks. This will be corrected in the revised manuscript.

Line 389: "in" instead of "on"

Reply: Thanks. This will be corrected in the revised manuscript.

Line 389: might be

Reply: Thanks. This will be corrected in the revised manuscript.

Line 453: "highest" instead of "largest"

Reply: Thanks. This will be corrected in the revised manuscript.

Figure 1: Please clarify if the Figure actually shows the location of catchment outlets or centroids.

Reply: Sorry for the confusion. The locations of catchments in this figure represent the catchment outlet. This will be illustrated in the revised manuscript.

Figure 3: Please indicate in the caption if model efficiency corresponds to calibrated model parameters here.

Reply: Sorry for the unclear statement. The spatial distribution of model efficiency in Figure 3 represents the calibrated KGE values over 2277 catchments. This will be clarified in the revised manuscript.

Figure 4: Why global mean and regression methods are not shown here? Please label the x axis. It would be useful if you will select the same colors for performance classes as in

Figure 3.

Reply: Here we evaluated the influence of the number of donor catchments as well as different weighting and averaging options on the regionalization performance of distance/attributes-based methods. The global mean and regression methods used in this study do not have this problem, therefore we did not show these two methods in figure 4. However, we compared the performance of different regionalization methods in figure 5.

The x-axis label will be added in the revised manuscript. And the color will be unified in the revised manuscript.

Figure 5: please specify what CAL stands for. If it stands for calibration, consider moving its box to the first position. Please clarify box plot structure in the caption since whiskers and the outliers are not plotted here.

Reply: Thanks for the comment. "CAL" represents the calibration results and this will be clarified in the revised manuscript. In addition, the box of calibration will be moved to the first position and the structure of the box plot will be clarified in the revised manuscript.

Table 2: Why only SPA and SPI-OUT are mentioned in the caption?

Reply: Sorry for the confusion. We want to emphasize the meaning of the abbreviation of different regionalization methods by showing these two examples. This will be changed to "For example, SPA means spatial proximity method with parameter averaging option and arithmetic mean approach; PSI-OUT means physical similarity method with output averaging option and Inverse Distance Weighted approach".

Table 6: Please indicate if any model was used to provide runoff estimate in the study of Korzun et al. (1978) and GRDC; please indicate which period was modelled in Widén-Nilsson et al. (2001). What does "approximately" means for GRDC time period?

Reply: The long-term annual runoff of Korzun et al. (1978) was derived from a comprehensive ensemble of a vast amount of data around the world. The long term annual runoff of GRDC was derived from observed river discharges and the simulation period was 1961-1990. In addition, the runoff simulation period modeled in Widén-Nilsson was 1915-2000. We will add the information in the revised manuscript.

References:

Korzun, V.I., Sokolow, A.A., Budyko, M.I., Voskresensky, K.P., Kalininin, G.P., Konoplyanstev, A.A., Korotkevich, E.S., Kuzin, P.S., Lvovich, M.I. (Ed.),: World Water Balance and Water Resources of the Earth. USSR National Committee for the International Hydrological Decade. English translation. Studies and Reports in Hydrology No. 25, Paris, UNESCO. pp. 663, 1978.

Widén-Nilsson, E., Halldin, S., and Xu, C.-y.: Global water-balance modelling with WASMOD-M: Parameter estimation and regionalisation, J Hydrol, 340, 105-118, 10.1016/j.jhydrol.2007.04.002, 2007.

Figure 11: Please indicate the scale for precipitation. Consider transforming discharge to mm/day to make it comparable with precipitation. Please remove a black line on the right and the double full stop in the caption.

Reply: Thank you. All these will be corrected in the revised manuscript.

References
Bárdossy, A. (2007). Calibration of hydrological model parameters for ungauged catchments. Hydrology and Earth System Sciences, 11(2), 703–710. https://doi.org/10.5194/hess-11-703-2007

Götzinger, J., and Bárdossy, A. (2007). Comparison of four regionalisation methods for a distributed hydrological model. Journal of Hydrology, 333(2–4), 374–384. https://doi.org/10.1016/j.jhydrol.2006.09.008

Grill, G., Lehner, B., Thieme, M., Geenen, B., Tickner, D., Antonelli, F., and Zarfl, C. (2019). Mapping the world's free-flowing rivers. Nature, 569(7755), 215–221. https://doi.org/10.1038/s41586-019-1111-9

Hundecha, Y., and Bárdossy, A. (2004). Modeling of the effect of land use changes on the runoff generation of a river basin through parameter regionalization of a watershed model. Journal of Hydrology, 292(1–4), 281–295. https://doi.org/10.1016/j.jhydrol.2004.01.002

Lehner, B., Liermann, C. R., Revenga, C., Vörösmarty, C., Fekete, B., Crouzet, P., and Wisser, D. (2011). High-resolution mapping of the world's reservoirs and dams for sustainable river-flow management. Frontiers in Ecology and the Environment, 9(9), 494–502. https://doi.org/10.1890/100125

Merz, R., Tarasova, L., and Basso, S. (2020). Parameter's Controls of Distributed Catchment Models—How Much Information is in Conventional Catchment Descriptors? Water Resources Research, 56(2), 1–18. https://doi.org/10.1029/2019WR026008

Mizukami, N., Clark, M. P., Newman, A. J., Wood, A. W., Gutmann, E. D., Nijssen, B., and Samaniego, L. (2017). Towards seamless large-domain parameter estimation for hydrologic models. Water Resources Research, 53(9), 8020–8040. https://doi.org/10.1002/2017WR020401

Oudin, L., Kay, A., Andréassian, V., and Perrin, C. (2010). Are seemingly physically similar catchments truly hydrologically similar? Water Resources Research, 46(11), 1–15. https://doi.org/10.1029/2009WR008887

Samaniego, L., Kumar, R., Thober, S., Rakovec, O., Zink, M., Wanders, N., Attinger, S. (2017). Toward seamless hydrologic predictions across spatial scales. Hydrology and Earth System Sciences, 21(9), 4323–4346. https://doi.org/10.5194/hess-21-4323-2017

Wallner, M., Haberlandt, U., and Dietrich, J. (2013). A one-step similarity approach for the regionalization of hydrological model parameters based on Self-Organizing Maps. Journal of Hydrology, 494, 59–71. https://doi.org/10.1016/j.jhydrol.2013.04.022

---

## Author Comment (AC2) · 26 Oct 2020

Dear Anonymous Referee #2,

We would like to thank the reviewer for the time taken in reviewing this paper. All comments will be incorporated into the revised manuscript. Please check the attached point-by-point replies. We will make the revisions to the manuscript as suggested.

Please also note the supplement to this comment: https://hess.copernicus.org/preprints/hess-2020-127/hess-2020-127-AC2-supplement.pdf
* * *
[Figure]

127, 2020.

**Supplement:**

Reply: We thank the reviewer for the constructive comments and advice. We have provided detailed responses to each comment below and will revise the manuscript accordingly. For clarity, comments are given in black, and our responses are given in blue.

The paper compares regionalization approaches at global scale. I like that the paper tests many different regionalization approaches but, unfortunately, one important approach is missing (more below). In addition, the English should be improved by a native English speaking person.

Reply: We appreciate that the reviewer is in favor of our study. All the comments and suggestions have been replied below and will be addressed in the revised manuscript. The revised version of the manuscript will undergo a language check by a native English speaking person. We believe reviewer's professional comments will greatly improve the quality of the manuscript.

The "spatial proximity method" may well yield the highest KGE, but it cannot be used at global scale due to the lack of gauges in many regions. As an example, grid cells in the southern Ecuadorian Andes would receive parameters from donor catchments located in the Amazon, which doesn't make any sense. So while this approach may give you the best performance scores, it is not actually a method that should be used at global scale. This should be explicitly mentioned in the abstract and the conclusion.
Even if the "mean distance between the donor catchments and the target catchment is no more than 1500 km" the actual difference in climate and landscape could be huge.

Reply: Thanks. This is indeed one of the limitations of the spatial proximity method. Beck et al. (2016, 2020) demonstrated that regionalization approaches provide slightly less (but still substantial) benefit in poorly gauged regions. According to the results shown in Figure 6, the advantage of the SPI-OUT method is reduced with the increased mean distance between donor and ungauged catchments, and it is not an easy task to decide which method outperforms others. However, when the mean distance is less than 1500 km, SPI-OUT generally performs better than or comparable to other regionalization methods in this study. Therefore, we took 1500 km as our threshold to get the optimal results of this method and decrease the influence of the increasing distance.

In addition, the high degree of variability in meteorological and hydrological variables, as well as the sparse meteorological and hydrological stations both make it difficult to effectively simulate runoff in poorly gauged regions. The selection of the catchment attributes which can properly represent the catchment similarity is also more difficult in these regions (Merz et al., 2020).

It is true that lack of gauges or too few gauges in many regions is a common problem for all other regionalization methods at global scale. This problem is even more pronounced for other methods than spatial proximity, for example regression method and physical similarity method, where no control about the distance for the donor catchments. Compared with other methods, spatial proximity is the only method that tries to choose the nearest catchments to the target catchments. Therefore, under the limitation of various conditions, using the spatial proximity method may be reasonable as it performs the best.

Following reviewer's comment and advice, above discussions will be added in the revised version.

References:

Beck, H. E., van Dijk, A. I. J. M., de Roo, A., Miralles, D. G., McVicar, T. R., Schellekens, J., and Bruijnzeel, L. A.: Global-scale regionalization of hydrologic model parameters, Water Resour Res, 52, 3599-3622, 10.1002/2015wr018247, 2016.
Beck, H. E., Pan, M., Lin, P., Seibert, J., Dijk, A. I. J. M., and Wood, E. F.: Global Fully Distributed Parameter Regionalization Based on Observed Streamflow From 4,229 Headwater Catchments, J Geophys Res (Atmospheres), 125, 10.1029/2019jd031485, 2020.
Merz, R., Tarasova, L., and Basso, S.: Parameter's Controls of Distributed Catchment Models—How Much Information is in Conventional Catchment Descriptors?, Water Resour Res, 56, 10.1029/2019wr026008, 2020.

"The k value of the PCRGLOBWB was determined based on the drainage [...]" This was paraphrased (like many other sentences) from Beck et al. (2016), but is this still the case in PCGLOBWB 2.0 (https://gmd.copernicus.org/articles/11/2429/2018/gmd-11-2429-2018.html)?

Reply: Yes, the k value in PCGLOBWB 2.0 is calculated following the drainage theory of Kraijenhoff van de Leur (1958) based on drainage network density and aquifer properties as that in PCGLOBWB 1.0. In addition, the PCGLOBWB 2.0 raises the possibility that coupling the MODFLOW to calculate groundwater heads and flow paths. We will add the clarification in the revised manuscript.

References:

Van De Leur, D. K.: A study of non-steady groundwater flow with special reference to a reservoir coefficient, De Ingenieur, 70, B87-B94, 1958.

Table 1: The aridity index can become extremely high in desert regions (higher than 100)

so I'm surprised that this isn't reflected in your mean. Did you apply some sort of mask, or did you cap the values before calculating the mean? This info needs to be in the caption.

Reply: Sorry for the confusion. We did not do some pretreatment before calculating the mean value for the aridity index. Table 1 shows the descriptors of each catchment. Only 247 catchments belong to the arid climate according to the Köppen-Geiger climate classification (1976-2000) and most of them are in the semiarid steppe regions. Therefore, the mean value of the aridity index is not that high. A more clear clarification will be added in the revised manuscript.

References:
Kottek, M., Grieser, J., Beck, C., Rudolf, B., and Rubel, F.: World Map of the Köppen-Geiger climate classification updated, Meteorol Z, 15, 259-263, 10.1127/0941-2948/2006/0130, 2006.

Table 1: Which datasets did you use for the different attributes?

Reply: Aridity and potential evaporation were derived from Global Aridity and PET Database (Zomer et al., 2008). Terrain characteristics were derived from Hydro1K Database (USGS, 1996). Land use was driven from GlobCover Land Cover Maps (Bichero et al., 2011). The soil index was obtained from the Harmonized World Soil Database (FAO and ISRIC, 2012). All the information above will be added in the revised manuscript.

References:
Bichero, P., Defourny, P., Brockmann, C., Schouten,L., Vancutsem, C., Huc, M., Bontemps, S., Leroy, M., Achard, F., and Herold, M.: Globcover 2009-products description and validation report, URL: http://ionia1. esrin. esa. int/docs/GLOBCOVER2009 Validation Report 2, vol. 2, 2011.
FAO, I., and ISRIC, I.: JRC: Harmonized world soil database (version 1.1), FAO, Rome, Italy and IIASA, Laxenburg, Austria, 2012.
USGS (US Geological Survey): HYDRO1K Elevation Derivative Database. Earth Resources Observation and Science (EROS) Data Center (EDC), Sioux Falls, South Dakota, USA. Available at http:// edc.usgs.gov/products/elevation/gtopo30/hydro/index.html, 1996.
Zomer, R. J., Trabucco, A., Bossio, D. A., and Verchot, L. V.: Climate change mitigation: A spatial analysis of global land suitability for clean development mechanism afforestation and reforestation, Agriculture, ecosystems & environment, 126, 67-80, 2008.

Section 2.2: Hydro1K is a very outdated dataset. A newer dataset should be used, maybe HydroSHEDS or MERIT.

Reply: Thanks for the suggestion. In recent years, many high-resolution topographical data sets that are potentially helpful in producing more accurate hydrography maps have been released (e.g., HydroSHEDS, MERIT, and ASTER GDEM), which are useful for regional modeling studies. However, the quality of Hydro1K has been confirmed by many previous

studies. In the last few years, Hydro1K has been widely used and has become the most commonly used global ancillary files for topographic index values. Therefore, due to the relatively low resolution used in this study (0.5 °), the efficiency of Hydro1K and a large amount of calculation for changing the topography data sets, we chose to use Hydro1K in this study and the use of updated datasets will be presented as a perspective in the revised manuscript. We will add it in the discussion of the revised manuscript.

References:
Gong, L., Halldin, S., and Xu, C. Y.: Global-scale river routing-an efficient time-delay algorithm based on HydroSHEDS high-resolution hydrography, Hydrol Process, 25, 1114-1128, 10.1002/hyp.7795, 2011.
Karlsson, J. M., and Arnberg, W.: Quality analysis of SRTM and HYDRO1K: a case study of flood inundation in Mozambique, Int J Remote Sens, 32, 267-285, 10.1080/01431160903464112, 2011.
Marthews, T. R., Dadson, S. J., Lehner, B., Abele, S., and Gedney, N.: High-resolution global topographic index values for use in large-scale hydrological modelling, Hydrol Earth Syst Sc, 19, 91-104, 10.5194/hess-19-91-2015, 2015.
Yamazaki, D., Ikeshima, D., Sosa, J., Bates, P. D., Allen, G., and Pavelsky, T.: MERIT Hydro: A high‐resolution global hydrography map based on latest topography datasets, Water Resour Res, 10.1029/2019wr024873, 2019.

Figure 1: Which KG map did you use? These are too many classes to distinguish, and the legend is a mess. Better to condense to the 5 major classes.

Reply: Thanks for the suggestion. The World Map of Köppen-Geiger climate classification (1976-2000) used in this study is from Kottek et al. (2006). The climate classification of each 2277 catchment in Figure 1 will be changed to the 5 major classes in the revised manuscript.

References:
Kottek, M., Grieser, J., Beck, C., Rudolf, B., and Rubel, F.: World Map of the Köppen-Geiger climate classification updated, Meteorol Z, 15, 259-263, 10.1127/0941-2948/2006/0130, 2006.

Equation 1: This is the old KGE, the new KGE has a slightly better formulation of the variability component. See Kling et al. (2012; https://doi.org/10.1016/j.hydrol.2012.01.011).

Reply: Thanks for the suggestion. The new version of KGE will be added as one of the criteria in the revised manuscript.

Section 2.5: Among the five tested regionalization methods, one important approach is missing, and it is most likely the best approach. It is the approach where the model parameters and the regression equations (relating the catchment attributes to the model parameters) are optimized simultaneously. See for example https://agupubs.onlinelibrary.wiley.com/doi/abs/10.1029/2019JD031485 and

https://agupubs.onlinelibrary.wiley.com/doi/full/10.1029/2008wr007327. This approach should definitely be included.

Reply: Thanks for the suggestion. The simultaneous regionalization method mentioned by the reviewer, in which parameter regionalization is carried out through simultaneous calibration of transfer function parameters by assuming prior relationships between basin predictors, was proposed to solve the parameter equifinality problem and was widely used in recent years (Hundecha and Bárdossy, 2004; Samaniego et al., 2010, 2017; Beck et al., 2020). However, the high degree of variability in meteorological and hydrological variables, as well as different simplifications of hydrological processes in hydrological models make it difficult to effectively select the catchment attributes and the proper transfer functions of parameters (Mizukami et al., 2017). Especially, the high computational cost in performing this regionalization method makes it difficult to use in this study.

In addition, the regionalization methods used in this study are the most commonly used, which are less data and computation demanding. Although they do have some limitations, these methods have been successfully used in the regionalization of ungauged catchments all over the world. These methods also show reasonable performance in regionalization in this study. In addition, the GHMs built by the proposed framework show reasonable efficiency in global water resource estimation and are comparable to other previous GHMs in catchment streamflow simulation (Widén-Nilsson et al., 2007; Beck et al., 2016; Arheimer et al., 2020). Therefore, we would like to conduct an even more comprehensive comparative study for all the methods on well selected data rich regions in the future study.

References:
Arheimer, B., Pimentel, R., Isberg, K., Crochemore, L., and Pineda, L.: Global catchment modelling using World-Wide HYPE (WWH), open data, and stepwise parameter estimation, Hydrol Earth Syst Sc, 24, 535-559, 2020.
Beck, H. E., van Dijk, A. I. J. M., de Roo, A., Miralles, D. G., McVicar, T. R., Schellekens, J., and Bruijnzeel, L. A.: Global-scale regionalization of hydrologic model parameters, Water Resour Res, 52, 3599-3622, 10.1002/2015wr018247, 2016.
Beck, H. E., Pan, M., Lin, P., Seibert, J., Dijk, A. I. J. M., and Wood, E. F.: Global Fully Distributed Parameter Regionalization Based on Observed Streamflow From 4,229 Headwater Catchments, J Geophys Res (Atmospheres), 125, 10.1029/2019jd031485, 2020.
Hundecha, Y., and Bárdossy, A.: Modeling of the effect of land use changes on the runoff generation of a river basin through parameter regionalization of a watershed model, J Hydrol, 292, 281-295, 2004.
Mizukami, N., Clark, M. P., Newman, A. J., Wood, A. W., Gutmann, E. D., Nijssen, B., Rakovec, O., and Samaniego, L.: Towards seamless large-domain parameter estimation for hydrologic models, Water Resour Res, 53, 8020-8040, 10.1002/2017wr020401, 2017.
Samaniego, L., Kumar, R., and Attinger, S.: Multiscale parameter regionalization of a grid-based hydrologic model at the mesoscale, Water Resour Res, 46, 10.1029/2008wr007327, 2010.
Samaniego, L., Kumar, R., Thober, S., Rakovec, O., Zink, M., Wanders, N., Eisner, S.,

M üller Schmied, H., Sutanudjaja, E. H., Warrach-Sagi, K., and Attinger, S.: Toward seamless hydrologic predictions across spatial scales, Hydrol Earth Syst Sc, 21, 4323-4346, 10.5194/hess-21-4323-2017, 2017.

Wid én-Nilsson, E., Gong, L., Halldin, S., and Xu, C. Y.: Model performance and parameter behavior for varying time aggregations and evaluation criteria in the WASMOD-M global water balance model, Water Resour Res, 45, 10.1029/2007wr006695, 2009.

"The regression-based method assumes that a well-behaved relationship exists in the observable catchment characteristics and model parameters" which in reality is almost never the case due to parameter equifinality and therefore this approach rarely works. Hence my suggestion to test the other regionalization approach.

Reply: Thanks for the suggestion. We agree that the regression-based method suffers most of the equifinality problem since this problem makes us cannot find the so called "true parameter values". Therefore, other methods are more favorable. The regression method will be deleted in the revised manuscript. In addition, we will adjust the contents and results in the revised manuscript.

References:

Arsenault, R., and Brissette, F. P.: Continuous streamflow prediction in ungauged basins: The effects of equifinality and parameter set selection on uncertainty in regionalization approaches, Water Resour Res, 50, 6135-6153, 10.1002/2013wr014898, 2014.

"The SP method assumes that nearby catchments should have similar behavior for climate and catchment conditions (features) varying uniformly in space." "Nearby" could be several thousand kilometers away so this approach should never be used at global scale.

Reply: Sorry that we failed to explain it clear enough in the original manuscript. As we explained in the reply to the first comment above, lack of gauges or very few gauges in many regions is a common problem for all other regionalization methods at global scale. This problem is even much more pronounced for other methods than spatial proximity, for example regression method and physical similarity method, where no control about the distance for the donor catchments. Compared with other methods, spatial proximity is the only method that tries to choose the nearest catchments to the target catchments. Studies have shown that this is likely the most suitable method at global scale since other methods, among other problems, can choose donor catchment with distance much longer than the SP method. Other studies also show that spatial proximity method chose donor catchment, on average with shorter distance than other methods. We will add clarification and discussion to this important issue in the revised manuscript.

Another limitation of the study is that lumped catchment attribute and model parameter values are used, despite the often large heterogeneity within catchments. This limitation is addressed in several studies (e.g., Samaniego 2008 and Beck 2020) and should at least be discussed somewhere in the paper.

Reply: Thanks for the suggestion. Lumped models consider catchment as a whole and thus ignore the within-catchment heterogeneity in landscape and climate (Beck et al., 2020; Samaniego et al., 2010). However, due to the large number of basins and the large amount of calculations, the use of distributed models in the regionalization methods comparison and global scale regionalization scheme selection may not be appropriate. Therefore, four frequently used conceptual hydrological models in regionalization studies were selected to compare the performance of regionalization methods and to find the optimal global scale regionalization scheme in this study. We will add this in the discussion of the revised manuscript.

References:
Beck, H. E., Pan, M., Lin, P., Seibert, J., Dijk, A. I. J. M. V., and Wood, E. F.: Global Fully Distributed Parameter Regionalization Based on Observed Streamflow From 4,229 Headwater Catchments, J Geophys Res (Atmospheres), 125, 2020.
Samaniego, L., Kumar, R., and Attinger, S.: Multiscale parameter regionalization of a grid-based hydrologic model at the mesoscale, Water Resour Res, 46, 10.1029/2008wr007327, 2010.

"The results show that the distributions of model efficiency of four hydrological models are similar to each other and indicate that the difference between hydrological models was negligible in the model calibration and validation, which is in line with previous studies (Beck et al., 2016; Vetter 245et al., 2015; Demirel et al., 2015)." This is definitely not in line with previous studies, which generally found large differences between models and a considerable difference between calibration and validation scores. You actually also obtained substantial differences between calibration and validation scores (figure 2)!

Reply: Sorry that we failed to make it clear enough in the previous version. Here we would like to illustrate that from the calibration to the validation period, the differences of KGE from hydrological models are similar. The sentence will be changed to "The results show that the distributions of model efficiency of four hydrological models are similar to each other. In particular, the difference of KGE among hydrological models is similar from the calibration to the validation periods." This sentence will be corrected in the revised manuscript.

Figures 3 and 10: This figure is impossible to interpret. https://www.climate-labbook. ac.uk/2016/why-rainbow-colour-scales-can-be-misleading/

Reply: These two figures will be revised in the revised manuscript.

Figure 5: The figure is a bit difficult to interpret, maybe use vertical instead of diagonal x-axis labels, and apply some coloring to group similar methods together?

Reply: Thanks for the suggestion. The figure will be revised.

Figure 6: There is no information about the number of catchments representing each bar, so a particular bar could be represented by just 1 catchment. I can't think of a solution right

now, but this information should not be hidden.

Reply: Thanks for the referee's suggestion. This figure will be redrawn and the information will be added in the revised manuscript.

Figure 8: A non-linear color scale might better.

Reply: Thanks. This will be revised.

Table 6: "many data may not be directly comparable because of different continental boundaries and averaging periods." A solution would be to only use estimates representing the same area. Also, why report values from a study >40 years old (Korzun 1978)? Considering adding GSCD estimates (http://www.gloh2o.org/gscd/).

Reply: Thanks for the suggestion. This is the optimal way to compare the results of the global water resource simulated in this study. However, it is hard to realize because of the difficulty in obtaining fully coherent data. We tried to emphasize that the comparison of the discharge from the literature may not seem intuitive for the different continental boundaries and averaging periods used in different studies. Therefore, the high uncertainty exists in global and continent mean annual discharge simulation should not be ignored and more efforts should be made to improve the efficiency of global hydrological modeling.

Besides, as one of the initial reports that summarize the water balance for the globe, the results of Korzun et al. (1978) have played important roles in global water management and provided a base for global water balance research (Jones et al., 1979; Döll et al., 2003; Oki et al., 2006; Fasullo et al., 2007). Therefore, taking this report into account is of great use in making this comparison complete.

The GSCD estimates will be added in the comparison.

All the information above will be added in the revised manuscript.

References:
Döll, P., Kaspar, F., and Lehner, B.: A global hydrological model for deriving water availability indicators: model tuning and validation, J Hydrol, 270, 105-134, 2003.
Fasullo, J., Dai, A., Qian, T., Smith, L., and Trenberth, K. E.: Estimates of the Global Water Budget and Its Annual Cycle Using Observational and Model Data, J Hydrometeorol, 8, 758-769, 10.1175/jhm600.1, 2007.
Jones, J. R., Beall, R. M., and Giusti, E. V.: International cooperation in water resources, GeoJournal, 3, 481-487, 1979.
Korzun, V.I., Sokolow, A.A., Budyko, M.I., Voskresensky, K.P., Kalininin, G.P., Konoplyanstev, A.A., Korotkevich, E.S., Kuzin, P.S., Lvovich, M.I. (Ed.),: World Water Balance and Water Resources of the Earth. USSR National Committee for the International Hydrological Decade. English translation. Studies and Reports in Hydrology No. 25, Paris, UNESCO. pp. 663, 1978.
Oki, T., and Kanae, S.: Global hydrological cycles and world water resources, Science, 313,

1068-1072, 2006.